

# The representation of alkalinity and the carbonate pump from CMIP5 to CMIP6 ESMs and implications for the ocean carbon cycle

Alban Planchat[1], Lester Kwiatkowski[2], Laurent Bopp[1], Olivier Torres[1], James R. Christian[3], Momme Butenschön[4], Tomas Lovato[4], Roland Séférian[5], Matthew A. Chamberlain[6], Olivier Aumont[2], Michio Watanabe[7], Akitomo Yamamoto[7], Andrew Yool[8], Tatiana Ilyina[9], Hiroyuki Tsujino[10], Kristen M. Krumhardt[11], Jörg Schwinger[12], Jerry Tjiputra[12], John P. Dunne[13], Charles Stock[13]

[1] LMD-IPSL, CNRS, Ecole Normale Supérieure/PSL Res. Univ, Ecole Polytechnique, Sorbonne Université, Paris, 75005, France
[2] LOCEAN Laboratory, Sorbonne Université-CNRS-IRD-MNHN, Paris, 75005, France
[3] Canadian Centre for Climate Modelling and Analysis, Victoria, BC, Canada
[4] Ocean Modeling and Data Assimilation Division, Fondazione Centro Euro-Mediterraneo sui Cambiamenti Climatici (CMCC), Bologna, Italy
[5] CNRM (Université de Toulouse, Météo-France, CNRS), Toulouse, France
[6] CSIRO Oceans and Atmosphere, Hobart, TAS, Australia
[7] Atmosphere and Ocean Research Institute, the University of Tokyo, Chiba, Japan
[8] National Oceanography Centre, Southampton, SO13 3ZH, UK
[9] Max Planck Institute for Meteorology, Bundesstraße 53, 20146 Hamburg, Germany
[10] JMA Meteorological Research Institute, Tsukuba, Ibaraki, Japan
[11] Climate and Global Dynamics, National Center for Atmospheric Research, Boulder, Colorado, U.S.A.
[12] NORCE Climate & Environment, Bjerknes Centre for Climate Research, Bergen, Norway
[13] NOAA/OAR Geophysical Fluid Dynamics Laboratory, Princeton NJ, USA

*Correspondence to*: Alban Planchat (alban.planchat@lmd.ipsl.fr)

**Abstract.**

Ocean alkalinity is critical to the uptake of atmospheric carbon in surface waters and provides buffering capacity towards associated acidification. However, unlike dissolved inorganic carbon (DIC), alkalinity is not directly impacted by anthropogenic carbon emissions. Within the context of projections of future ocean carbon uptake and potential ecosystem impacts, especially through Coupled Model Intercomparison Projects (CMIPs), the representation of alkalinity and the main driver of its distribution in the ocean interior, the calcium carbonate cycle, have often been overlooked. Here we track the changes from CMIP5 to CMIP6 with respect to the Earth system model (ESM) representation of alkalinity and the carbonate pump which depletes the surface ocean in alkalinity through biological production of calcium carbonate, and releases it at depth through export and dissolution. We report a significant improvement in the representation of alkalinity in CMIP6 ESMs relative to those in CMIP5. This improvement can be explained in part by an increase in calcium carbonate ($CaCO_3$) production for some ESMs, which redistributes alkalinity at the surface and strengthens its vertical gradient in the water column. We were able to constrain a PIC export estimate of 51-70 Tmol $yr^{-1}$ at 100 m for the ESMs to match the observed vertical gradient of alkalinity. Biases in the vertical profile of DIC have also significantly decreased, especially with the enhancement of the carbonate pump, but the representation of the saturation horizons has slightly worsened in contrast. Reviewing the representation of the $CaCO_3$ cycle across CMIP5/6, we find a substantial range of parameterizations. While all biogeochemical models currently represent pelagic calcification, they do so implicitly, and they do not represent benthic calcification. In addition, most models simulate marine calcite but not aragonite. In CMIP6 certain model groups have increased the complexity of simulated $CaCO_3$ production, sinking, dissolution and sedimentation. However, this is insufficient to explain the overall improvement in the alkalinity representation, which is therefore likely a result of improved marine biogeochemistry model tuning or *ad hoc* parameterizations. We find differences in the way ocean alkalinity is initialized that lead to offsets of up to 1 % in the global alkalinity inventory of certain models. These initialization biases should be addressed in future CMIPs by adopting accurate unit conversions. Although modelers aim to balance the global alkalinity budget in ESMs in order to limit



drift in ocean carbon uptake under preindustrial conditions, varying assumptions in the closure of the budget have the potential to influence projections of future carbon uptake. For instance, in many models, carbonate production, dissolution and burial are independent of the seawater saturation state, and when considered, the range of sensitivities is substantial. As such, the future impact of ocean acidification on the carbonate pump, and in turn ocean carbon uptake, is potentially underestimated in current ESMs and insufficiently constrained.

## 1    Introduction

The ocean is a major carbon sink, absorbing a quarter of anthropogenic carbon emissions each year (Friedlingstein et al., 2022) limiting atmospheric $CO_2$ growth rate and hence anthropogenic warming. The cumulative ocean carbon sink is estimated at $170 \pm 35$ GtC over 1850-2020 (Friedlingstein et al., 2022), rising to $290 \pm 30$ GtC under the emission scenario SSP1–2.6 and to $520 \pm 40$ GtC under SSP5–8.5 by 2100 (Liddicoat et al., 2021; Canadell et al., 2021). Carbon uptake by the ocean is not

without consequences for marine ecosystems, as it leads to seawater acidification (Doney et al., 2009; Gattuso and Hansson, 2011), which poses a threat to many marine organisms (e.g., Dutkiewicz et al., 2015; Mostofa et al., 2016), particularly calcifying species (Ilyina et al., 2009; Ridgwell et al., 2009; Lohbeck et al., 2012; Meyer and Riebesell, 2015). In surface waters, the global average pH has already decreased by about 0.1 unit since the beginning of the industrial era (Bindoff et al., 2019). Depending on future emission scenarios, projected acidification would result in global-mean surface ocean pH

decreasing by 0.16 to 0.44 in 2080-2099 compared to preindustrial values (Kwiatkowski et al., 2020).

The absorption of anthropogenic carbon emissions by the ocean is primarily controlled by the increasing atmospheric $CO_2$ concentration and the resulting gradient of $CO_2$ partial pressure ($pCO_2$) across the air-sea interface. Yet, in the surface ocean, $pCO_2$ is controlled by the total amount of dissolved inorganic carbon (DIC) in seawater, but also by sea surface temperature, salinity and alkalinity that control $CO_2$ solubility and the partitioning of DIC between dissolved $CO_2$, bicarbonate,

and carbonate ions. Total alkalinity (Alk), defined as the excess of proton acceptors over proton donors (Dickson, 1981; essentially the sum of the carbonate, borate, water, phosphoric, silicic and fluoride alkalinity components), is a central concept in ocean sciences. Despite multiple definitions that have undoubtedly led to some confusion (Dickson, 1992; Zeebe and Wolf-Gladrow, 2001; Middelburg et al., 2020), Alk has remained a key quantity for studying the ocean carbon cycle primarily because it is (i) measurable, (ii) conservative, and (iii) used to solve the ocean $CO_2$ system. (i) Alk has been extensively

measured by titration methods (Thompson and Anderson, 1940) since the pioneering work of Tornøe (1880) and Dittmar (1884). Today, the Global Ocean Data Analysis Project (GLODAP) compiles Alk measurements from more than 1.3 million water samples collected on almost 1,000 cruises covering the global ocean (Lauvset et al., 2021). (ii) Alk is conservative, i.e., unchanged with respect to modifications of temperature and pressure and conserved during mixing of water masses of different properties. It is thus used in oceanic models of the carbon cycle as a prognostic variable (Zeebe and Wolf-Gladrow, 2001;

Wolf-Gladrow et al., 2007). (iii) knowing Alk in combination with any of the variables DIC, $pCO2$, or $[H^+]$ (Dickson et al., 2007) allows one to compute the entire ocean $CO_2$ system – i.e., the respective concentrations of $CO_2$, $HCO_3^-$, $CO_3^{2-}$, as well as pH and dissolved inorganic carbon (DIC).

Alk is dependent on multiple physical and biogeochemical processes, the interpretation of which is not always straightforward. At the ocean surface, it is mainly affected by freshwater fluxes (precipitation, evaporation, sea-ice formation

or melting and riverine discharge) through dilution or concentration. As a result, the surface distribution of Alk shows a strong salinity dependence (Friis et al., 2003) with higher surface Alk values in regions of net evaporation (e.g., subtropical gyres) and lower surface Alk in regions of net precipitation (e.g., near the Equator). In the ocean interior, the Alk distribution is mainly driven by the biological pump, associated to the consumption of DIC and Alk at the ocean surface through biological production, and the remineralization or dissolution of the biogenic material at depth after sinking (Hain et al., 2014). It is

predominately the carbonate pump, also called the hard tissue pump, that drives the Alk distribution in the water column through (1) biotic calcification in the upper ocean, (2) sinking of biogenic calcium carbonate particles, (3) dissolution, and (4) the burial of part of this particulate inorganic carbon (PIC) at the seafloor (Fig. 1). Fig. 1Calcification acts as a biological Alk sink in the upper ocean, while dissolution at depth acts as a source. In contrast, the soft tissue pump – associated with the production, export and remineralization of organic matter – has only a limited influence on the vertical distribution of Alk

through the consumption and release of nutrients, essentially nitrates (Sarmiento and Gruber, 2006; Wolf-Gladrow et al., 2007). By affecting the balance of proton acceptors over proton donors, nitrogen reactions (nitrification, $N_2$-fixation and



denitrification) can also affect Alk in the water column (Wolf-Gladrow et al., 2007). On centennial timescales, the global ocean inventory of Alk is thought to be roughly in steady state (Revelle and Suess, 1957) – estimated at about 3.56 Pmol – but potential variations are difficult to estimate due to the influence of processes at the ocean boundaries (Middelburg et al., 2020;
see also (0) in Fig. 1). In particular, in addition to freshwater fluxes at the ocean surface, rivers act as an Alk source due to the natural weathering of silicate and carbonate minerals on land, as well as sediment mobilization at the seafloor (Middelburg et al., 2020).

Preliminary work by Revelle and Suess (1957) to determine how carbon dioxide is partitioned between the atmosphere and the ocean initiated a sustained series of modelling efforts to represent the ocean carbon cycle and its coupling
with increasing atmospheric $CO_2$. Early modelling studies, using either box models (Siegenthaler and Sarmiento, 1993) or ocean general circulation models (Maier-Reimer and Hasselmann, 1987; Sarmiento et al., 1992), assumed a spatially homogeneous surface Alk to calculate ocean carbon uptake. Sarmiento et al. (1992), for example, used a constant surface Alk of 2,300 µeq kg$^{-1}$ (where eq refers to molar equivalent since Alk is a charge balance), but recognized that their approach neglected that surface Alk is not constant, and that inclusion of variable Alk would require a model with biology. Some later
studies updated the uniform Alk approach by imposing a local surface Alk that varies proportionally with salinity (e.g., the Princeton solubility model, involved in the first phase of the Ocean Carbon Cycle Model Intercomparison Project, OCMIP-1, Sarmiento et al., 2000). In 1990, the pioneering work of Bacastow and Maier-Reimer (1990) introduced an explicit representation of Alk and calcium carbonate cycling in a three-dimensional ocean general circulation model. In this approach, Alk is included as a three-dimensional state variable and calcium carbonate ($CaCO_3$) formation in the surface ocean is related
to the rate of particulate organic carbon (POC) production with a spatially and temporally constant rain ratio – defined as the ratio between the export of PIC and POC. The downward flux of $CaCO_3$ is assumed to decrease exponentially with depth and all $CaCO_3$ reaching the seafloor dissolves there instantaneously. In a later publication, this approach was updated by Maier-Reimer (1993), with the description of the HAMOCC3 biogeochemical model, in which Alk is also represented as a three-dimensional state variable but with a time and space variable rain ratio, reduced in low-temperature regions, a fixed penetration
depth of 2 km for $CaCO_3$, and explicit interactions with the sediment. In the second phase of OCMIP (OCMIP-2, Doney et al., 2004; Orr et al., 2005), the 13 modelling groups adopted a common biogeochemical framework and followed the approach of Yamanaka and Tajika (1996), after Bacastow and Maier-Reimer (1990), with explicit Alk and a spatially-homogeneous rain ratio. Later work on modelling the global ocean carbon cycle continued in this direction, with an explicit representation of Alk, augmented by the implicit incorporation of aragonite in addition to calcite (Gangstø et al., 2008; Dunne et al., 2013) and
the recent representation of calcifying plankton functional groups (Buitenhuis et al., 2019; Krumhardt et al., 2019).

The development of marine biogeochemical models that resolve the carbonate pump, and consequently better represent the distribution of Alk in the ocean, has furthered our understanding of the evolution of the carbonate pump, and possible feedback on ocean carbon uptake and acidification (e.g., Gehlen et al., 2007; Gangstø et al., 2011; Yool et al., 2013a). Representing Alk and $CaCO_3$ cycling in ESMs requires marine biogeochemistry modelers to balance the complexity required
to evaluate specific processes alongside computational efficiency, within the wider context of representing the Earth system, and particularly the carbon cycle response to anthropogenic emissions.

The objectives of this study are (1) to document how the processes affecting Alk are represented in the latest generation of Earth system models (ESMs), and (2) to evaluate the Alk distribution simulated by each of these models against observations. To do this, we use the latest generation of ESMs (from the 6th phase of the Coupled Model Intercomparison
Project, CMIP6) and compare these models to those from the previous phase (5th phase, CMIP5).





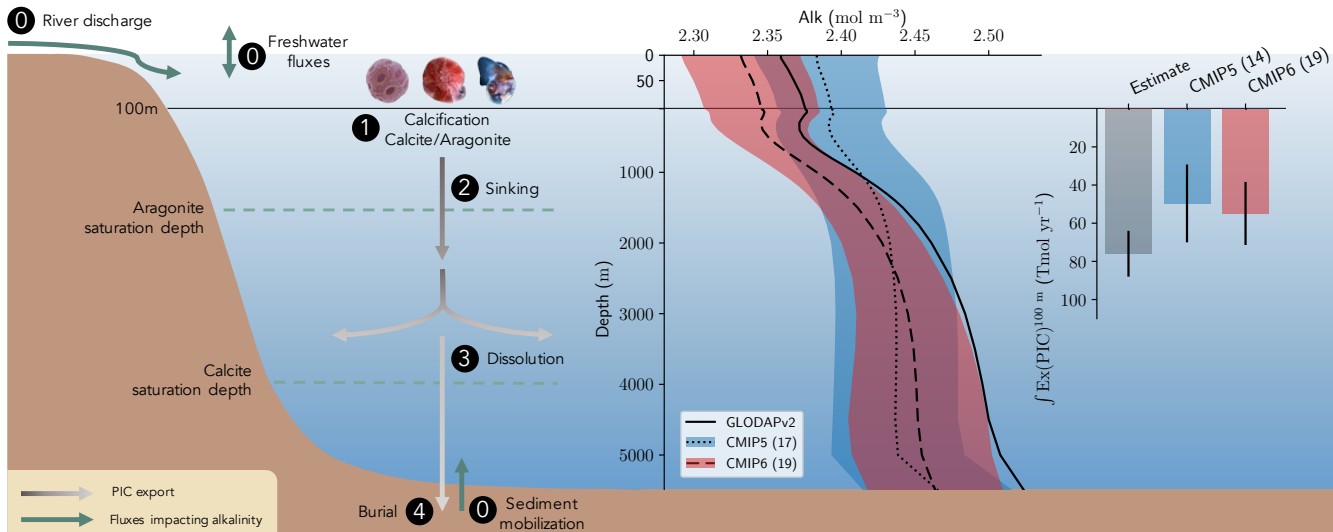

**Fig. 1: Schematic illustration of the processes affecting alkalinity (Alk) at the ocean surface and of the key steps of the carbonate pump addressed in this study. For the CMIP5 and CMIP6 ensemble means, an overview of the Alk profile and a bar chart of the**

**total particulate inorganic carbon (PIC) export at 100 m is also presented, both with their associated standard deviation. Also shown are observations from GLODAPv2 for the Alk profile and an estimate for the PIC export from Sulpis et al. (2021) – at 300 m.**

## 2 Methodology

### 2.1 CMIP ESMs and their marine biogeochemical models

#### 2.1.1 CMIP ESMs

We assess 36 ESMs from 13 different climate modelling centers (CCCma, CMCC, CNRM-CERFACS, CSIRO, HAMMOZ-Consortium, IPSL, MIROC, MOHC, MPI-M, MRI, NCAR, NCC and NOAA-GFDL), which took part in the 5[th] and/or 6[th] phases of the Climate Model Intercomparison Project (CMIP5, Taylor et al., 2012; CMIP6, Eyring et al., 2016). We only consider ESMs for which Alk is not prescribed but determined by physical and biogeochemical processes represented in the

models (e.g., calcification and dissolution, or also primary production and remineralization). This leads us to include 17 ESMs for CMIP5 and 19 for CMIP6 (Table 1). We use "[ESM_name] (CMIP[number])" to refer to a given ESM, indicating if it was used for CMIP5 or CMIP6, but we also refer to modelling centers or to the specificities of the marine biogeochemical components of the given ESMs.

In total, this ESM intercomparison encompasses 15 marine biogeochemical models (CMOC, CanOE, BFM, PISCES,

WOMBAT, OECO, diat-HadOCC, MEDUSA, HAMOCC, NPZD-MRI, BEC, MARBL, TOPAZ, BLING, COBALT) with different versions and/or configurations depending on the CMIP and the modelling group. All the ESMs considered in this study represent the carbonate pump, with the exception of CMCC-CESM (CMIP5), which we include in our analysis to highlight the effect of implementing such a pump. NASA-GISS ESMs were not included since Alk is prescribed there.

**Table 1: Ensemble of ESMs processed in this ESM intercomparison and some of their characteristics.**

| Group | CMIP | ESM | MBG | Experiments (variant label) |
|---|---|---|---|---|
| CCCma | CMIP5 | CanESM2 | CMOC | Historical+RCP4.5, piControl (r1i1p1) |





| | | | | |
|---|---|---|---|---|
| | CMIP6 | CanESM5 | CMOC | Historical, piControl (r1i1p2f1) |
| | CMIP6 | CanESM5-CanOE | CanOE | Historical, piControl (r1i1p2f1) |
| CMCC | CMIP5 | CMCC-CESM | BFM4 | Historical+RCP8.5, piControl (r1i1p1) |
| | CMIP6 | CMCC-ESM2 | BFM5.2 | Historical, piControl (r1i1p1f1) |
| CNRM-CERFACS | CMIP5 | CNRM-ESM1 | PISCESv1 | Historical+RCP4.5, piControl (r1i1p5) |
| | CMIP6 | CNRM-ESM2-1 | PISCESv2-gas | Historical, piControl (r1i1p1f2) |
| CSIRO | CMIP6 | ACCESS-ESM1-5 | WOMBAT | Historical, piControl (r1i1p1f1) |
| HAMMOZ-Consortium | CMIP6 | MPI-ESM-1-2-HAM | HAMOCC6 | Historical, piControl (r1i1p1f1) |
| IPSL | CMIP5 | IPSL-CM5A-LR | PISCESv1 | Historical+RCP4.5, piControl (r1i1p1) |
| | CMIP5 | IPSL-CM5A-MR | PISCESv1 | Historical+RCP4.5, piControl (r1i1p1) |
| | CMIP5 | IPSL-CM5B-LR | PISCESv1 | Historical+RCP4.5, piControl (r1i1p1) |
| | CMIP6 | IPSL-CM6A-LR | PISCESv2 | Historical, piControl (r1i1p1f1) |
| MIROC | CMIP5 | MIROC-ESM | OECO1 | Historical+RCP4.5, piControl (r1i1p1) |
| | CMIP5 | MIROC-ESM-CHEM | OECO1 | Historical+RCP4.5, piControl (r1i1p1) |
| | CMIP6 | MIROC-ES2L | OECO2 | Historical, piControl (r1i1p1f2) |
| MOHC | CMIP5 | HadGEM2-CC | diat-HadOCC | Historical+RCP4.5, piControl (r1i1p1) |
| | CMIP5 | HadGEM2-ES | diat-HadOCC | Historical+RCP4.5, piControl (r1i1p1) |
| | CMIP6 | UKESM1-0-LL | MEDUSA-2.1 | Historical, piControl (r1i1p1f2) |
| MPI-M | CMIP5 | MPI-ESM-LR | HAMOCC5.2 | Historical+RCP4.5, piControl (r1i1p1) |
| | CMIP5 | MPI-ESM-MR | HAMOCC5.2 | Historical+RCP4.5, piControl (r1i1p1) |
| | CMIP6 | MPI-ESM1-2-LR | HAMOCC6 | Historical, piControl (r1i1p1f1) |
| | CMIP6 | MPI-ESM1-2-HR | HAMOCC6 | Historical, piControl (r1i1p1f1) |
| MRI | CMIP5 | MRI-ESM1 | NPZD-MRI | Historical+RCP8.5 (r1i1p1) |
| | CMIP6 | MRI-ESM2-0 | NPZD-MRI | Historical, piControl (r1i2p1f1) |
| NCAR | CMIP5 | CESM1-BGC | BEC | Historical+RCP4.5, piControl (r1i1p1) |
| | CMIP6 | CESM2 | MARBL | Historical (r10i1p1f1), piControl (r1i1p1f1) |
| | CMIP6 | CESM2-WACCM-FV2 | MARBL | Historical, piControl (r1i1p1f1) |
| | CMIP6 | CESM2-FV2 | MARBL | Historical, piControl (r1i1p1f1) |
| | CMIP6 | CESM2-WACCM | MARBL | Historical, piControl (r1i1p1f1) |
| NCC | CMIP5 | NorESM1-ME | HAMOCC5.1 | Historical+RCP4.5, piControl (r1i1p1) |
| | CMIP6 | NorESM2-LM | iHAMOCC | Historical, piControl (r1i1p1f1) |
| NOAA-GFDL | CMIP5 | GFDL-ESM2G | TOPAZ2 | Historical+RCP4.5, piControl (r1i1p1) |
| | CMIP5 | GFDL-ESM2M | TOPAZ2 | Historical+RCP4.5, piControl (r1i1p1) |
| | CMIP6 | GFDL-CM4 | BLINGv2 | Historical, piControl (r1i1p1f1) |
| | CMIP6 | GFDL-ESM4 | COBALTv2 | Historical, piControl (r1i1p1f1) |

### 2.1.2 Review of the marine biogeochemical models

We review the key properties of marine biogeochemical models simulating Alk, seeking to share an in-depth overview of the representation of the Alk tracer in these models, as well as its main interior driver, the carbonate pump. Specifically, we have
collected a wide range of information from the different groups, both for CMIP5 and CMIP6, regarding the protocols followed (e.g., spin-up, initialization), the model boundary conditions (e.g., river discharge, Alk restoration) and the biological complexity and representation of explicit or implicit mechanisms (e.g., $CaCO_3$ production, dissolution, sedimentation). In addition, we also report the nitrogen reactions taken into account, or not, by the different models, but we do not explore their effects on the Alk distribution which can be complex under low oxygen conditions (e.g., Stock et al., 2020). Finally, it should



be noted that we were unable to collect comparable information for all models, notably the HAMMOZ-Consortium group, which is assumed to be identical to MPI-M for CMIP6 in terms of marine biogeochemistry modelling (Table 1).

## 2.2 ESM data and processing

### 2.2.1 ESM data

For the different ESMs, we systematically processed the CMIP piControl and Historical experiments. For CMIP5, where the
historical simulation covers only the time period up to 2005, we concatenated the Historical experiments with RCP4.5 (or RCP8.5 if not available) from 2005 to 2014 to allow for data averaging over 1992-2012, for consistency with observations (see Sect. 2.3). Finally, we analyse only one ensemble member per ESM (Table 1), such that we do not address the role of internal variability in the emergence of climate-related changes on the key marine biogeochemistry variables we consider in this analysis.
The following variables were processed when available: (i) two-dimensional (2D) variables: 'epc100' (sinking flux of organic matter at 100 m, in mol m$^{-2}$ s$^{-1}$), 'epcalc100' (sinking flux of calcite at 100 m, in mol m$^{-2}$ s$^{-1}$), 'eparag100' (sinking flux of aragonite at 100 m, in mol m$^{-2}$ s$^{-1}$); (ii) three-dimensional (3D) variables: 'talk' (total alkalinity, in mol m$^{-3}$), 'dissic' (dissolved inorganic carbon, in mol.m$^{-3}$), 'no3' (nitrate concentration, in mol m$^{-3}$), 'po4' (phosphate concentration, in mol m$^{-3}$), 'so' (salinity, in g kg$^{-1}$), 'thetao' (potential temperature, in K for CMIP5 and °C for CMIP6). In addition, potential
density was calculated from 'so' and 'thetao'. Export values at 100 m were extracted from 3D export fields when 2D exports at 100 m were not provided using 'expc' (sinking flux of organic matter, in mol m$^{-2}$ s$^{-1}$), 'expcalc' (sinking flux of calcite, in mol m$^{-2}$ s$^{-1}$), and 'exparag' (sinking flux of aragonite, in mol m$^{-2}$ s$^{-1}$). Export data were not available for MIROC-ESM and MIROC-ESM-CHEM (CMIP5).

### 2.2.2 Background processing

To facilitate the ESM intercomparison, we used Climate Data Operator (CDO) functions to regrid ESM outputs and observations. Specifically, we used distance-weighted average remapping 'remapdis' to regrid the data on a regular 1°x1° grid, and linear level interpolation with extrapolation 'intlevelx' to regrid to the World Ocean Atlas (WOA) vertical grid with 33 depth levels up to 5500 m (even though 5500 m is not accessible for some of the CMIP5 ESMs). The mid-point of the uppermost level, which we refer to as the surface hereafter, was set to 5 m, given this is the deepest upper ocean level among
the ESMs considered. The analysis was performed in Python with the use of the Gibbs SeaWater (gsw) oceanographic toolbox for ocean property conversions. We also used *mocsy 2.0* (Orr and Epitalon, 2015) to compute the ocean carbonate system over our averaging period (1992-2012) with the use of (i) Alk, DIC, phosphate (or nitrate divided by a Redfield ratio, $r_{N:P}$=16, if not available), salinity and temperature from ESM outputs, (ii) silicate from GLODAPv2 observations (Olsen et al., 2020) as it is not included in many ESMs, and (iii) the equilibrium constants recommended for best practices (Dickson et al., 2007; Orr
and Epitalon, 2015). Quality control of ESM outputs, led us to: (i) exclude 'po4' for CMCC-CESM (CMIP5) due to anomalously high values and long-term drift, and (ii) disregard the few values given at 5500 m for MIROC ESMs in CMIP5 likely used for exchanges at the seafloor. Finally, each model is weighted in the calculation of CMIP5 and CMIP6 statistical values (mean, standard deviation, quantiles, and linear regressions) such that each modelling group has the same total contribution.

### 2.2.3 Drift assessment

We did not correct for potential drift in the ESM outputs in order to maintain the consistency of the internal mechanisms of the ESMs. However, the piControl simulations were assessed for drift and this is discussed in Sect. 4.2.1. Using the piControl data coincident with the Historical and RCP/SSP simulations (250-yr long piControl simulations), we were able to assess if the ESMs had reached a quasi-steady state prior to the Historical simulation. We assessed the drift of the vertical gradients of
Alk, DIC, nitrate and phosphate between the deep ocean and the surface, considering the difference between the last 20 years and their first 20 years of these piControl simulations. Similarly, we also estimated the drift in surface salinity and temperature as well as the spatially integrated exports of PIC and POC at 100 m. This drift assessment was, however, not possible for MRI-



ESM1 (CMIP5) due to the lack of available piControl outputs, and only carried it out for DIC and Alk for CanESM2 (CMIP5) as nitrate data were not available.

### 2.2.4 Open ocean mask

Our analysis focuses on the representation of Alk and the carbonate pump in the open ocean. Thus, most of the analysis and the associated figures consider the open ocean (defined in Appendix A 6.1 rather than the entire ocean. Indeed, our aim was to exclude coastal regions due to the coarse ESM resolution, but particularly to avoid inclusion of river discharge effects on Alk (see Sect. 2.4). The entire ocean was considered when values were integrated to compare with observationally-based estimates (e.g., PIC and POC exports at 100 m). Unless otherwise specified, differences between consideration of the open ocean and entire ocean were negligible.

### 2.3 Data products

We use the gridded data from the Global Ocean Data Analysis Project (GLODAP) to evaluate the model performance. This database is built with bias-corrected water column bottle data – merged with CTD data for salinity – from the ocean surface to bottom (Olsen et al., 2020). In particular, we made use of the second update of the second version of the gridded product (GLODAPv2.2020, Olsen et al. 2020), with an improved and extended coverage compared to the original second version (Lauvset et al., 2016) and the first version (Key et al., 2004), especially in the Arctic. GLODAPv2.2020 – referred to as GLODAPv2 hereafter – contains data from 946 cruises, covering the global ocean from 1972 to 2019 with two quality controls, and adjustments to minimize severe biases. Note that we use the GLODAPv2 product that was normalized to the year 2002 for DIC to avoid biases due to the accumulation of anthropogenic carbon over the observational period. For consistency, we use nutrient fields from GLODAPv2 rather than those given by WOA, maintaining the same method for mapping the nutrients as for DIC and Alk (Lauvset et al., 2016).

Regarding the export of CaCO$_3$ from the surface, we consider the latest estimate published by Sulpis et al. (2021) – referenced to 300 m though –, which is consistent with the other recent estimate from Battaglia et al. (2016) although the methodology employed differs. While Sulpis et al. (2021) is an observationally constrained probabilistic evaluation, Battaglia et al. (2016) is an assessment from seawater chemistry and water-age data. These estimates seem to mark a point of agreement (76.0 ± 12.0 Tmol yr$^{-1}$ for the former and 75.0 [60.0; 87.5] Tmol yr$^{-1}$ for the latter) in the evaluations carried out since the late 1980s, which range from about 45 to 150 Tmol yr$^{-1}$ (Sulpis et al., 2021). This echoes both the sparsity and collection biases of *in situ* data from sediment-trap measurements, and the difficulty to evaluate the contribution of CaCO$_3$ to the Alk budget with mapped observations and numerical tools. Finally, we considered the estimate from DeVries and Weber (2017) for the POC export at 100 m (558 Tmol yr$^{-1}$).

### 2.4 Salinity normalization

As Alk is highly correlated with salinity in the upper ocean due to freshwater fluxes (e.g., precipitation, evaporation and river discharge; Friis et al., 2003), salinity normalization is required to assess the influence of biogeochemical processes. We use the canonical normalization approach of dividing Alk and DIC values by the coincident salinity and multiplying this by a reference salinity value of 35 g kg$^{-1}$:

$$\begin{cases} sAlk = 35 \cdot \dfrac{Alk}{S} \\ sDIC = 35 \cdot \dfrac{Alk}{S} \end{cases} \tag{1}$$

This gives the Alk and DIC that the considered fluid parcel would have at a salinity of 35 g kg$^{-1}$ (e.g. Sarmiento and Gruber, 2006; Fry et al., 2015). This approach was deemed appropriate given that our analysis is focused on the global open ocean, and therefore near-zero salinity values simulated by certain ESMs in the coastal ocean or closed seas are not taken into account. Hereafter, salinity-normalized Alk and DIC are referred to as sAlk and sDIC, respectively. The influence of alternative salinity





normalization techniques (Robbins, 2001; Friis et al., 2003; Carter et al., 2014; Koeve et al., 2014; Sulpis et al., 2021) on our results was assessed and found to be limited (see Appendix B 6.2).

## 2.5    Estimating the biological pump and related quantities

The expression and quantification of the biological carbon pump is essential to understanding the influence of biological processes on the distribution of both Alk and DIC. The biological pump can be split into a soft tissue pump associated with the production and remineralization of organic matter, and the carbonate pump associated with the production and dissolution of CaCO$_3$. Here we define the pumps relative to the surface following Sarmiento and Gruber (2006). Thus, we express the soft tissue pump ($\delta C_{soft}$) and carbonate pump ($\delta C_{carb}$) as:

$$\delta C_{soft} = r_{C:P} \cdot \delta PO_4^{3-} = r_{C:N} \cdot \delta NO_3^{-} \tag{2}$$

$$\delta C_{carb} = \frac{1}{2}[\delta Alk + r_{Nut:P} \cdot \delta PO_4^{3-}] = \frac{1}{2}\left[\delta Alk + \frac{r_{Nut:P}}{r_{N:P}} \cdot \delta NO_3^{-}\right] \tag{3}$$

where NO$_3^-$ and PO$_4^{3-}$ respectively refer to the nitrate and phosphate concentrations, and for each tracer $\tau$, $\delta\tau = \tau - \tau^{5\,m}$. $r_{C:P}$, $r_{C:N}$ and $r_{N:P}$ are C:P, C:N and N:P ratios, and $r_{Nut:P}$ is a nutrient to phosphorus ratio with regards to the effect of the soft tissue pump on Alk. The C:P ratio is model-dependent in our analysis ($r_{C:P}$=106 for GLODAPv2 and CMIP5/6 ensemble mean),
whereas the N:P ratio is fixed ($r_{N:P}$=16) and $r_{P:P}$=1 by definition. We can thus infer $r_{Nut:P}$=$r_{N:P}$+$r_{P:P}$+2·$r_{S:P}$=21.8, where we assume the S:P ratio at $r_{S:P}$=2.4 for the observations (Wolf-Gladrow et al., 2007). Since the effect of sulfur is not taken into account in models, we use $r_{Nut:P}$=$r_{N:P}$+$r_{P:P}$=17 for the ESMs (e.g., Brewer et al., 1975; Sarmiento and Gruber, 2006). This definition of the biological pump does not take into account the influence of ocean circulation. As a consequence, we limit the consideration of these pumps to horizontally-averaged open ocean regions, and the calculation with phosphate is preferred in
the analysis, when possible. Indeed, very low nitrate concentrations can be observed at the ocean surface in locations where significant phosphate remains. Our restriction of this calculation to open ocean regions also reflects concerns that nutrient inputs from the ocean boundaries may also bias estimates of the pumps in coastal regions (Sarmiento and Gruber, 2006). We use the same decomposition approach for all ESMs and GLODAPv2 neglecting that: (i) the soft tissue pump has no impact on Alk in BFM4 and MEDUSA-2.0; (ii) CMCC ESMs, NOAA-GFDL ESMs (excluding GFDL-CESM4) and NCAR ESMs
involved in CMIP6 have variable or various $r_{N:P}$ and/or $r_{C:P}$ ; and (iii) CMCC for CMIP5 had no representation of the carbonate pump (see Supplementary Table S1).

From this definition, it is important to highlight the dependency of the carbonate pump on the soft tissue pump, combining Eq. (2) and (3):

$$\delta C_{carb} = \frac{1}{2}\left[\delta Alk + \frac{r_{Nut:P}}{r_{C:P}} \cdot \delta C_{soft}\right] \tag{4}$$

A positive soft tissue pump value refers to net remineralization at a given depth compared to the surface, and a negative one to net organic matter production. Similarly, a positive carbonate pump corresponds to net dissolution relative to the surface, and a negative one to net calcification. Net remineralization compared to the surface results in positive values for both the soft tissue and carbonate pumps, whereas net dissolution compared to the reference level results in positive values only for the carbonate pump. As highlighted by Sarmiento and Gruber (2006), $\delta C_{soft}$ and $\delta C_{carb}$ are "potential" pumps as they reveal the
biological processes that drive the distribution of sDIC and sAlk within the ocean but fail to account for the indirect effect of biology on air-sea CO$_2$ fluxes.

Alk and the carbonate cycle are closely linked due to the effect of calcification and dissolution, but the soft tissue pump also impacts Alk. To estimate the drivers of ESM sAlk vertical profile biases in the open ocean, we decompose sAlk. We start by differentiating sAlk:

$$dsAlk = 35 \cdot \frac{dAlk \cdot S - Alk \cdot dS}{S^2} \tag{5}$$

Rewriting Eq. (3) and (4), $\delta Alk$ can be expressed in terms of the carbonate and soft tissue pumps:

$$\delta Alk = 2 \cdot \delta C_{carb} - r_{Nut:P} \cdot \delta PO_4^{3-} = 2 \cdot \delta C_{carb} - \frac{r_{Nut:P}}{r_{C:P}} \cdot \delta C_{soft} \tag{6}$$

This means that at a given depth z, Alk can be expressed as follows:

$$Alk(z) = Alk^{5\,m} + 2 \cdot \delta C_{carb}(z) - \frac{r_{Nut:P}}{r_{C:P}} \cdot \delta C_{soft}(z) \tag{7}$$



where $Alk^{5\,m}$ refers to the surface Alk. Combining Eq. (7) and (5), this results in:

$$dsAlk = 35 \cdot \left[ \frac{1}{S} \cdot dAlk^{5\,m} - \frac{Alk}{S^2} \cdot dS + \frac{1}{S} \cdot 2 \cdot d\delta C_{carb} - \frac{1}{S} \cdot \frac{r_{Nut:P}}{r_{C:P}} \cdot d\delta C_{soft} \right] \qquad (8)$$

distinguishing four terms related to the role of surface Alk, salinity as well as both the carbonate and soft tissue pumps. Using the operator $\Delta$ defined on a tracer $\tau$ by $\Delta\tau = \tau^{Simu} - \tau^{Obs}$, we can express, as a first approximation, from Eq. (8), the difference of sAlk at a given depth z between the ESMs and the observations from GLODAPv2 using a reference value at the surface:

$$\Delta sAlk(z) = \frac{35}{\overline{S}} \cdot \Delta Alk^{5\,m} - 35 \cdot \overline{\left( \frac{Alk}{S^2} \right)} \cdot \Delta S(z) + \frac{70}{\overline{S}} \cdot \Delta(\delta C_{carb})(z) - \frac{35}{\overline{S}} \cdot \frac{r_{Nut:P}}{r_{C:P}} \cdot \Delta(\delta C_{soft})(z) \qquad (9)$$

where the overbar corresponds to the mean bias between the simulations and the observations. In this way, we compute a relative decomposition - since the components remain partly interdependent in their definition (e.g., the carbonate pump is computed from the soft tissue pump) - to compare the different terms with the observations. Although another approach was proposed by Oka (2020), our intention here is to isolate the role of the surface Alk bias, which is key in driving carbon fluxes.

The same approach can be followed for sDIC to estimate the drivers of ESM sDIC vertical profile biases. Equation (5) can be rewritten for DIC by substituting DIC for Alk, and Eq. (7) can be expressed for DIC as:

$$DIC(z) = DIC^{5\,m} + \delta C_{carb}(z) + \delta C_{soft}(z) + \delta C_{gas+ant}(z) \qquad (10)$$

where $DIC^{5\,m}$ refers to the surface DIC. $\delta C_{gas+ant}$ is considered as the combination of the gas exchange pump and the anthropogenic carbon uptake (Sarmiento and Gruber, 2006). This results in:

$$dsDIC = 35 \cdot \left[ \frac{1}{S} \cdot dDIC^{5\,m} - \frac{DIC}{S^2} \cdot dS + \frac{1}{S} \cdot d\delta C_{carb} + \frac{1}{S} \cdot d\delta C_{soft} + \frac{1}{S} \cdot d\delta C_{gas+ant} \right] \qquad (11)$$

distinguishing one additional term compared to the differentiation of sAlk associated with the combined influence of the gas exchange pump and anthropogenic carbon uptake. Once again, we can express, as a first approximation, from Eq. (11), the difference of sDIC at a given depth z between the ESMs and the observations using a reference value at the surface:

$$\Delta sDIC(z) = \frac{35}{\overline{S}} \cdot \Delta DIC^{5\,m} + 35 \cdot \overline{\left( \frac{DIC}{S^2} \right)} \cdot \Delta S(z) + \frac{35}{\overline{S}} \cdot \Delta(\delta C_{carb})(z) + \frac{35}{\overline{S}} \cdot \Delta(\delta C_{soft})(z) + \frac{35}{\overline{S}} \cdot \Delta(\delta C_{gas+ant})(z) \quad (12)$$

Although an approximation, this decomposition especially allows us to assess the roles of both the soft tissue and carbonate pumps, as well as surface biases on the sDIC vertical profile, analogous to the decomposition of the sAlk vertical profile biases in Eq. (9).

## 3  Results

### 3.1  Survey of relevant model parameterizations

Our review of the representation of Alk and the carbonate pump in the ESMs leads us to share a synthesis, which allows us to compare the different models and their evolution from CMIP5 to CMIP6 (Fig. 2). Here we document the key features we have identified regarding the carbonate pump and the global Alk budget. Additional model information is provided in Fig. C1 and in Appendix C 6.3, including specific model equations and parameters. A detailed overview of the modelling schemes used by the different groups is provided in the Supplementary Table S1. While CMIP models generally represent the carbonate

pump in a limited number of formulations, the specific details of these often vary between models.





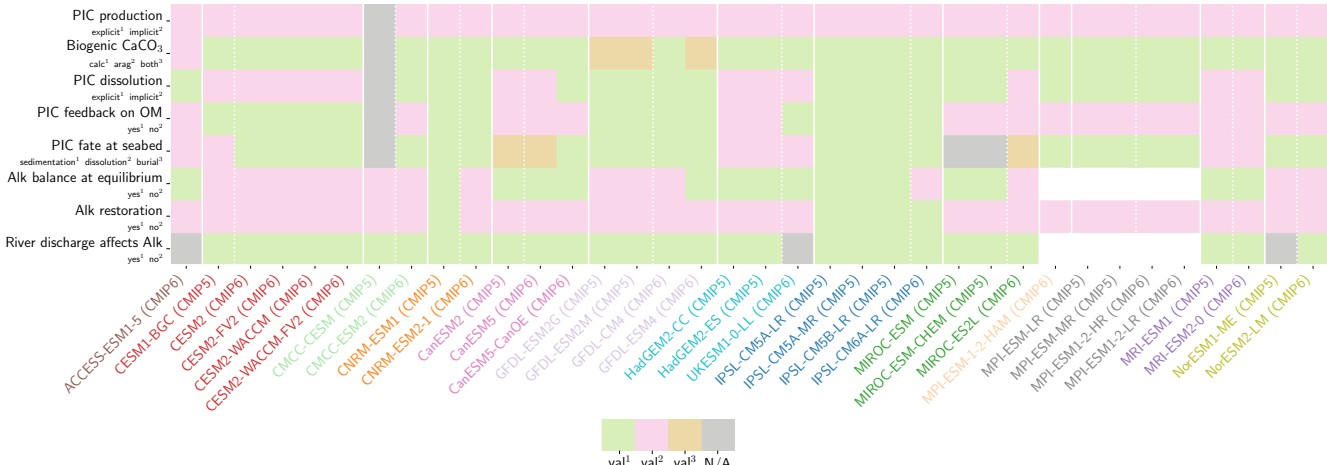

**Fig. 2: ESM description regarding the representation of Alk and the carbonate pump. The metrics displayed refer to the biogeochemical modelling schemes. For each metric, the name is given on the left of the table with the possible values in small ('val[1]', 'val[2]' and possibly 'val[3]'). A more complete representation of this figure is available in Fig. C1, and more information can be found in the Supplementary Table S1.**

### 3.1.1 Calcification

All biogeochemical models that consider the carbonate pump represent pelagic calcification implicitly in both CMIP5 and CMIP6. None of them explicitly incorporate a representation of a calcifying planktonic functional type (PFT). For most of the models, biogenic $CaCO_3$ is in the form of calcite although aragonite is also considered by NOAA-GFDL in TOPAZ2 and COBALTv2. Certain groups represent a generic biogenic $CaCO_3$ (CSIRO with WOMBAT for CMIP6, MIROC with OECO1/2 for CMIP5/6 and MOHC with diat-HadOCC for CMIP5), but attribute it either to calcite or aragonite based on their outputs and their consistency with these two forms of $CaCO_3$ (e.g., export distribution) to conform to CMIP output requirements. Finally, we note that none of the models represent benthic production of $CaCO_3$. This reinforces the decision to focus our analysis on the open ocean and to exclude the coastal ocean when possible.

The parameterizations used to represent implicit pelagic calcification are various and show a dependence on a variable number of drivers. Most of the models determine implicit calcification rates as a function of the fate of phytoplankton (through mortality and excretion by zooplankton after grazing), with certain models additionally considering the fate of zooplankton (through mortality and excretion by other zooplanktons after consumption). In contrast, in CMIP5, diat-HadOCC for MOHC, OECO1 for MIROC and TOPAZ2 for NOAA-GFDL are the only models for which calcification is directly related to phytoplankton growth. In addition, model calcification exhibits various dependencies on nutrient concentrations (phosphate, nitrate, iron and silica), temperature, light, depth, and the calcium carbonate saturation state ($\Omega$):

$$\Omega = \frac{[CO_3^{2-}] \cdot [Ca^{2+}]}{K_{sp}} \approx \frac{[CO_3^{2-}]}{[CO_{3sat}^{2-}]} \qquad (13)$$

where $[CO_3^{2-}]$ is the carbonate ion concentration, $[Ca^{2+}]$ is the calcium ion concentration – which is considered proportional to salinity – and $K_{sp}$ is the apparent solubility product of $CaCO_3$. $\Omega$ is approximated in some models to the ratio between the carbonate ion concentration and the one at saturation, $[CO_{3sat}^{2-}]$. NOAA-GFDL with TOPAZ2, BLINGv2 and COBALTv2, as well as MOHC with MEDUSA-2.0 all consider $CaCO_3$ production dependent on the saturation state with no calcification in undersaturated waters ($\Omega<1$). The implicit calcification parameterizations adopted by ESMs directly relate net $CaCO_3$ production to the export of PIC, as opposed to gross $CaCO_3$ production of which only a fraction is exported. By not explicitly resolving the grazing of calcifying plankton and partial egestion of $CaCO_3$ by zooplankton (e.g., due to gut dissolution), it is expected that simulated $CaCO_3$ production will generally be less than observational estimates of the total production of biogenic calcium carbonate.



### 3.1.2 Sinking and dissolution

The sinking of PIC is model-dependent with both explicit and implicit representations in the CMIP5 and CMIP6 ensembles.
When represented explicitly, a sinking speed is considered for PIC. This speed is constant in models with the exception of PISCESv1/2(-gas) where it is depth-dependent. PIC dissolution is computed using a dissolution rate and/or a dependence on the calcium carbonate saturation state. This dependence on the saturation state is variously represented in the models. Generally PIC dissolution in the water column occurs in undersaturated waters ($\Omega < 1$) with a linear dependence on the saturation state (BFM5.2, PISCESv2(-gas), HAMOCC5.2/6, HAMOCC5.1/iHAMOCC, TOPAZ2 and COBALTv2), although its
representation is more complex in PISCESv1 and BLINGv2.

When PIC sinking and dissolution is represented implicitly, a dissolution length-scale and an exponential decay of the downward flux divergence of $CaCO_3$ represents the combination of instantaneous sinking and dissolution in the water column. Such parameterizations result in the highest dissolution in the surface ocean and the lowest near the ocean bottom. diat-HadOCC, is unique in representing PIC dissolution homogeneously through the water column below a globally uniform
lysocline depth – upper limit of the transition zone, where sediments are subjected to very little dissolution.

### 3.1.3 Ballast and protection effects

In the water column, PIC can be considered both as a ballast for organic matter, increasing the sinking speed of POC, but also as a protector of organic matter reducing the rate at which it is remineralized. It is relevant to distinguish both processes, as their feedback on the soft tissue pump are often exclusively treated. However, what the modelling groups typically consider
as a ballast effect is generally better described as a protection effect, as it reduces organic matter remineralization during the sinking of POC. The formulation of this process is typically based on the model proposed by Armstrong et al. (2001) in which a component of the sinking POC flux is associated with sinking $CaCO_3$ and experiences reduced remineralization. It is generally parameterized using the data collated by MEDUSA-2.0, BEC, MARBL, TOPAZ2, BLINGv2 and COBALTv2). PISCESv1/2(-gas) and BEC are the only models which parameterize a ballast effect. In PISCESv1/2(-gas) half of the POC
produced by nanophytoplankton is associated to calcifiers and routed to fast sinking particles while in BEC and MARBL a fraction of the POC is associated to the higher dissolution length-scale for "hard" particles.

### 3.1.4 Sedimentation and Alk sources/sinks

The fate of PIC reaching the seafloor is one of the determinants of the ocean Alk inventory and closure of the $CaCO_3$ budget. There is a high diversity among models in their representation of sedimentation processes associated with calcium carbonate.
For some models, all of the PIC reaching the seafloor is considered permanently buried and lost from the ocean. Other models dissolve all of the PIC reaching the seafloor closing the calcium carbonate cycle and avoiding its processing in the seabed. Finally, a subset of models considers sediment processes, which can combine dissolution, burial and sediment mobilization.

Sedimentation is one way to balance broader inputs, and especially riverine discharge that many models either ignore or represent in only simplified ways (freshwater, Alk, DIC and nutrient discharge). At global scale, the sedimentation of PIC
at the seafloor corresponds to a net biological sink of Alk while sediment mobilization – essentially through the dissolution of $CaCO_3$ present in sediments – and river discharge are a net source. Although Alk sinks and sources are ideally balanced in steady state to avoid drift in the global Alk inventory, this is difficult to achieve in certain models and forced in others through the use of a fixed Alk inventory and a restoring term. As a result, sedimentation processes appear to be key to closing the $CaCO_3$ budget and are further discussed in Sect. 4.3.2.

## 3.2 Model performance

### 3.2.1 Alkalinity

The representation of surface Alk has evolved from CMIP5 to CMIP6 with a convergence of the global average value within the model ensembles, while regional disparities remain but to a lesser extent (Fig. 3). In CMIP5, the open ocean mean surface Alk is higher than the observations (+0.024 ± 0.028 mol m$^{-3}$; +1.0 %), in CMIP6 it is lower (-0.027 ± 0.023 mol m$^{-3}$; -1.1 %).




This reflects a global decrease in surface Alk between CMIP5 and CMIP6, with an inversion of the bias relative to GLODAPv2 observations (Fig. 3 and Fig. 4a). In addition, from CMIP5 to CMIP6, the variability among the ESMs with regards to the surface Alk was reduced, with a decrease in the standard deviation of the surface Alk (from 0.057 to 0.047 mol m$^{-3}$), especially in the Arctic Ocean (Fig. 3). However, in both CMIP5 and CMIP6, the global mean surface biases relative to the observations cannot be attributed to a specific and consistent regional bias among the ESMs (see Fig. D1).

Normalizing Alk by salinity (sAlk) to remove the impact of freshwater fluxes has little impact on CMIP ensemble biases (Fig. 4). Indeed, the open ocean mean surface sAlk bias compared to the observations is reversed and reduced in CMIP6 ($-0.014 \pm 0.016$ mol m$^{-3}$) compared to CMIP5 ($+0.038 \pm 0.022$ mol m$^{-3}$) and the inter-ESM standard deviation has decreased (from 0.044 in CMIP6 to 0.039 mol m$^{-3}$ in CMIP5; Fig. 4a). We can therefore infer that these changes are mainly driven by biogeochemical processes rather than changes in surface salinity driven by freshwater fluxes. In particular, the zonally
averaged sAlk for CMIP6 is closer to observations, with an enhancement in meridional variability (Fig. 4aFig. 3). This improvement between CMIP5 and CMIP6 seems to be mainly due to the elimination of some poor performing models in CMIP6 compared to CMIP5. The CNRM-CERFACS, MOHC, MIROC, and NCC ESMs in particular have a more consistent representation of the standard deviation of the sAlk surface distribution compared to observations, which goes alongside improved correlation (Fig. 4b). However, the standard deviation of sAlk in MRI ESMs has decreased from CMIP5 to CMIP6,
moving away from the observations, although the correlation is similar. On the other hand, there is only improvement in the correlation of sAlk in CMCC, while for IPSL the sAlk correlation is improved but this is accompanied with an excessive increase in the surface standard deviation.

    Associated with the global improvement in the representation of the surface sAlk is a significant increase of the sAlk vertical gradient (Fig. 5). The groups for which the ESMs show an improvement in the correlation and a considerable change
from CMIP5 to CMIP6 in the surface sAlk standard deviation – corresponding either to an improvement (CNRM-CERFACS, MOHC, MIROC and NCC) or a large bias (IPSL) – are the groups that reveal major improvement in the vertical profile of sAlk (Fig. 5). Indeed, from a relatively uniform sAlk profile, they have evolved towards a profile exhibiting increasing Alk at depth, more consistent with observations. For instance, the magnitude of the sAlk vertical gradient (the concentration anomaly at 5000 m with respect to the surface) has increased from 0.02 mol m$^{-3}$ in CMIP5 to 0.17 mol m$^{-3}$ in CMIP6 for the IPSL ESMs,
and from 0 mol m$^{-3}$ in CMIP5 to 0.17 mol m$^{-3}$ in CMIP6 for MOHC ESMs. The ESMs of these groups are predominantly responsible for the strengthened sAlk vertical gradient in CMIP6, which has increased from $0.05 \pm 0.05$ mol m$^{-3}$ to $0.12 \pm 0.05$ mol m$^{-3}$ (2.6-fold). The magnitude of the CMIP6 sAlk vertical gradient still remains below that of the observations (0.16 mol m$^{-3}$).





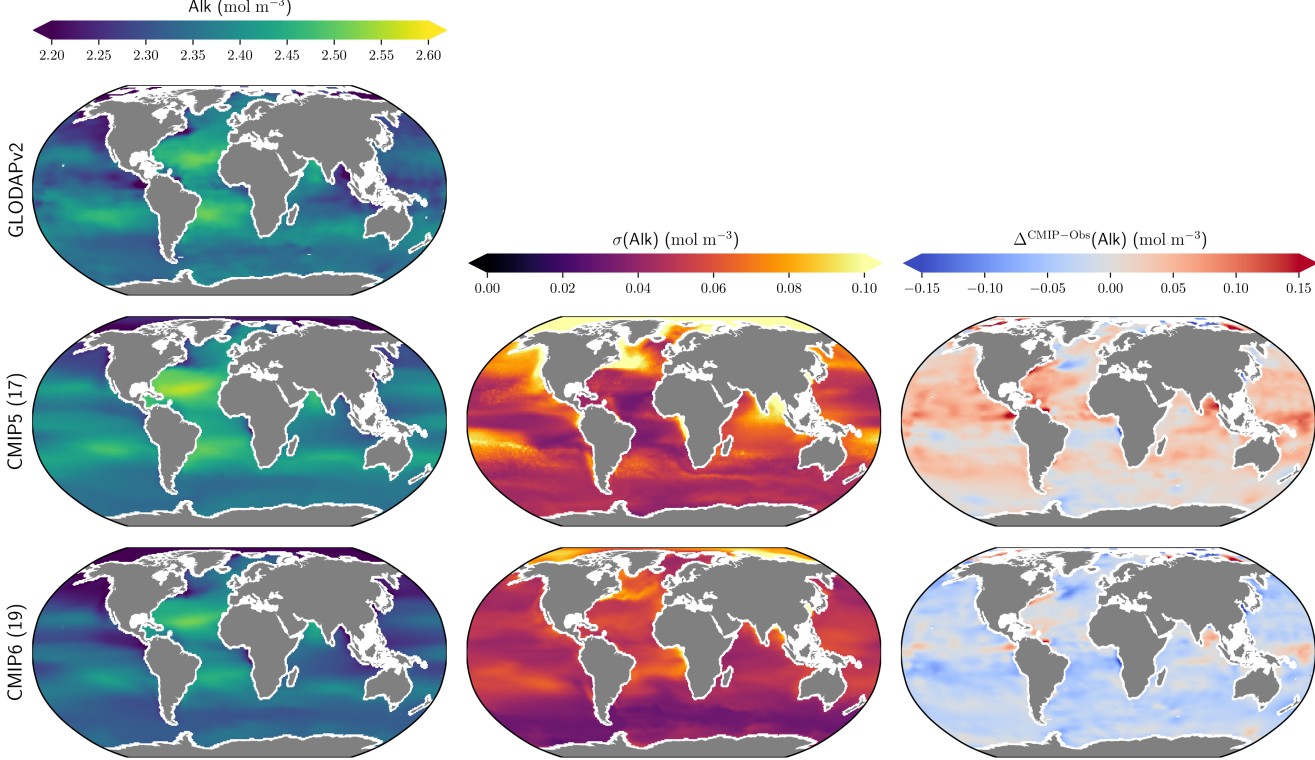


**Fig. 3: Alk surface distribution. ESM intercomparison of the open ocean surface Alk as simulated by ocean biogeochemical models involved in CMIP5 and CMIP6 compared to the observations from GLODAPv2. For each CMIP, the average, the standard deviation and the difference with the observations is shown.**



(a)

(b)

**Fig. 4: Spatial variability of the surface Alk. (a) Open ocean zonal averages of the surface Alk (left panel) and the surface sAlk (right panel) for the observations from GLODPv2 and both CMIP5 and CMIP6 ensemble means with their associated uncertainty. (b) Taylor diagram for the ESM intercomparison of the open ocean surface distribution of sAlk. The reference point corresponds to the observations from GLODAPv2 (black circle), and the two black markers refer to the CMIP5 and CMIP6 ensemble means. The CMIP5 (resp. CMIP6) ESMs are plotted with diamonds (resp. squares).**





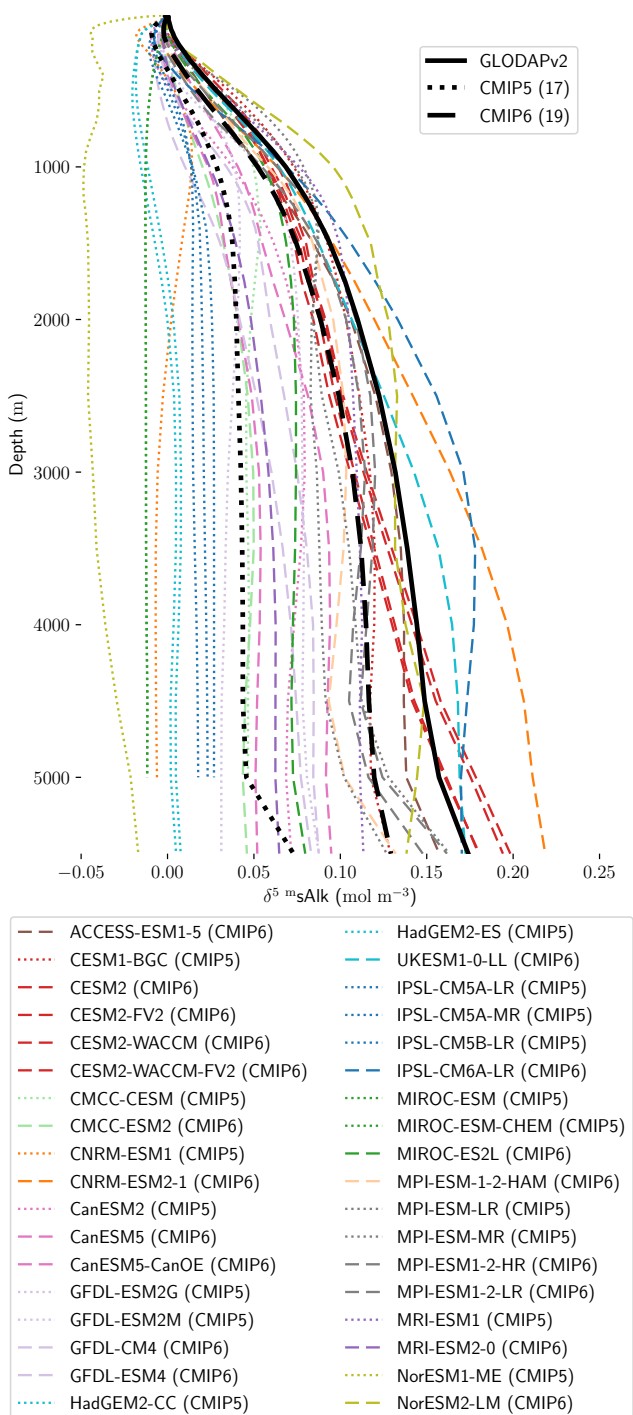

**Fig. 5: ESM intercomparison of the sAlk vertical profiles anomalies for the open ocean relative to the surface value. All the ESMs considered in this study are displayed in addition to both the CMIP5 and CMIP6 ensemble means as well as the observations from GLODAPv2.**




### 3.2.2    PIC export at 100 m

The global improvement in the representation of surface sAlk and its vertical gradient in CMIP6 is accompanied by a strengthening of the carbonate pump. This is illustrated by a global increase of 11 % in the PIC export at 100 m between CMIP5 (49 Tmol yr[-1]) and CMIP6 (55 Tmol yr[-1]; Fig. 6a). On the other hand, total POC export at 100 m has decreased by 7 % between CMIP5 (712 Tmol yr[-1]) and CMIP6 (659 Tmol yr[-1]). The combination of the two results in a 20 % increase in the rain ratio (RR) – defined as the ratio between PIC and POC export at 100 m (from 0.070 in CMIP5 to 0.083 in CMIP6; Fig.
6a). Overall, CMIP6 ESMs tend to better match with observational estimates of the exports and the RR compared to CMIP5 ESMs. We report a decoupling in the trends from CMIP5 to CMIP6 for the PIC and the POC exports, with an increase for the former and a decrease for the latter. While the CMIP6 average is 18 ± 27 % higher (+100 ± 151 Tmol yr[-1]) for POC export compared to the observationally informed estimates from DeVries and Weber (2017), it is 28 ± 26 % lower (-21 ± 20 Tmol yr[-1]) than the PIC export from Sulpis et al. (2021), highlighting inter-ESM variability (Fig. 6a).

The spatial distribution of the RR has also evolved from CMIP5 to CMIP6, echoing the contrasting trends of PIC and POC exports. There is a greater increase of the RR at high latitudes, resulting from both a greater increase in the PIC export and a smaller decrease in the POC export (Fig. 6b). However, as most of the PIC export is located in the tropics, and the changes are limited, the RR remains high for CMIP6 in the tropics. Although there is a global increase in the RR from CMIP5 to CMIP6, this shift is strongly associated with a few ESMs rather than illustrative of a general trend across ESMs. This is the
case for CNRM-CERFACS, IPSL and NCC, principally due to a PIC export increase for CNRM-CERFACS and IPSL, and to a POC export decrease for NCC (Fig. 6a). While our focus is on the open ocean, when the global ocean rather than the open ocean is considered, the integrated POC and the PIC export increase both by 31 % (+155 Tmol yr[-1] and +13 Tmol yr[-1] respectively) for the CMIP6 ensemble mean, highlighting the role of coastal ocean in these exports.

Finally, there is a noteworthy diversity among the ESMs regarding the spatial distribution of the PIC export at 100
m, with notable differences in the Pacific equatorial upwelling region, the great calcite belt in the Southern Ocean and the coastal ocean (see Fig. D2 andFig. D3 as well as Fig. D4 for the POC export). These points of disagreement between the ESMs are interestingly also found within the estimates proposed by Lee (2001), Jin et al. (2006), Sarmiento and Gruber (2006) and Battaglia et al. (2016) – which are summarized in Fig. 10 of the latter.





**Fig. 6: Exports of PIC and POC at 100 m. (a) Bar charts of the export at 100 m integrated over the entire ocean for the PIC and POC with a distinction for calcite and aragonite for all the ESMs, but for the CMIP5 and CMIP6 ensemble means, for which we consider the total PIC export. Assessments of the PIC and POC exports are respectively extracted from DeVries and Weber (2017) and Sulpis et al. (2021) – at 300 m for the PIC export. A global rain ratio (RR) is calculated from the ratio between the PIC export and the POC export at 100 m. (b) Circular bar plots respectively associated to the PIC export, the POC export and the RR at 100 m. For each one of these variables, (i) the amplitude of the bars refers to the relative difference between the CMIP6 and CMIP5 ensemble means; (ii) the color of the bars corresponds to the mean value for the CMIP6 ensemble mean (cf. gray colorbars at the bottom) ; and (iii) the error bars are associated to the standard deviation within the CMIP6 ensemble with values normalized between 0 and 0.5 (the minimum and maximum standard deviations are respectively marked by a black and a white dot on the colorbars at the bottom). The graphic is to be read in the trigonometric sense with the Atlantic, Indian and Pacific Oceans from south to north, and the global basins, excluding the Arctic, in the last quarter.**

### 3.2.3 The carbonate pump

The carbonate pump and the surface Alk are the two components that explain most of the sAlk biases between the ESMs and the observations at depth (Fig. 7). Indeed, using the decomposition of the sAlk bias at depth expressed in Eq. 9, we can





distinguish the roles of surface Alk, salinity, the carbonate pump and the soft tissue pump in driving this bias. Surface Alk and the carbonate pump are each responsible in average for 47 to 25 % and 48 to 66 % of the model sAlk bias at 5000 m (Fig. 7). Their respective influence has nevertheless changed from CMIP5 to CMIP6, with a greater relative contribution of surface Alk to the sAlk bias and a reduced contribution of the carbonate pump, which is further analysed in Section 3.3. In contrast, we find that the salinity and the soft tissue pump have minimal influence on the sAlk bias at 5000 m in both the CMIP5 and CMIP6

ensembles, contributing less than 5 %. Hence, the effect of the soft tissue pump and salinity on the vertical gradient of sAlk is minimal, with the biases essentially driven by the representation of the carbonate pump.

The CMIP6 increase in PIC export at 100 m globally acts to decrease sAlk at the ocean surface and increase it at depth, which could be explained by an enhanced production at the surface and/or dissolution at depth and/or sinking of the PIC. Using all the ESMs, we find a significant relationship between the sAlk vertical gradient in the open ocean and the global

PIC export at 100 m ($R^2$=0.54, p<0.01; Fig. 8a). This reflects inter-ESM consistency between higher export of PIC at 100 m and an associated increase in the vertical gradient of sAlk – expressed as the difference between the mean sAlk between 4000 and 5000 m and the mean sAlk in the upper layer, between 5 and 100 m. In particular, this relationship captures the shift towards higher values of PIC export at 100 m as well as a strengthened sAlk vertical gradient in CMIP6. Surprisingly, it encompasses the wide variety of CMIP modelling schemes used to represent Alk and the $CaCO_3$ cycle, especially regarding

sinking, dissolution and seabed processes. In addition, the relationship established between the global sAlk vertical gradient and PIC export across the CMIP5/6 ESMs can be combined with the sAlk vertical gradient from the GLODAPv2 observations to infer PIC export at 100 m. This approach, similar to so called "emergent constraint" methodologies (e.g., Eyring et al., 2019; Hall et al., 2019) provides a constrained PIC export estimate of 51-70 Tmol yr[-1] at 100 m (see black vertical line in Fig. 8a). This estimate is towards the lower end of PIC export values independently estimated by Sulpis et al. (2021; 76 ± 12 Tmol yr[-1]

at 300 m) and Battaglia et al. (2016; 75.0 [60.0; 87.5] Tmol yr-1) but is within the confidence interval in both cases. This reflects an apparent too small PIC export at 100 m for the resulting sAlk vertical gradient in the ESMs in comparison with the observations.

The sAlk vertical gradient across the combined CMIP5/6 ESM ensemble is also consistently related to the surface meridional distribution of sAlk through the meridional overturning circulation (MOC) with the upwelling of Alk-enriched

deep waters in the Southern Ocean. In particular, models with a higher sAlk vertical profile have higher meridional gradients of sAlk at the surface – expressed as the difference between surface sAlk in the Southern Ocean, [-90, -45]°, and the low latitudes, [-45, 45]° – ($R^2$=0.46, p<0.01; Fig. 8b). Here again, the shift towards higher values for the meridional sAlk gradient and the vertical sAlk gradient from CMIP5 to CMIP6 is noticeable. Despite known differences in the representation of the Southern upwelling as well as in the PIC export spatial distribution with the CMIP5/6 ensemble, the relationship found for the

ESMs agrees with the GLODAPv2 observations. Unfortunately, the linear relationships related to the sAlk vertical gradient are not sufficiently robust to be combined, which would have allowed us to directly use the meridional gradient of surface sAlk to constrain global PIC export at 100 m.





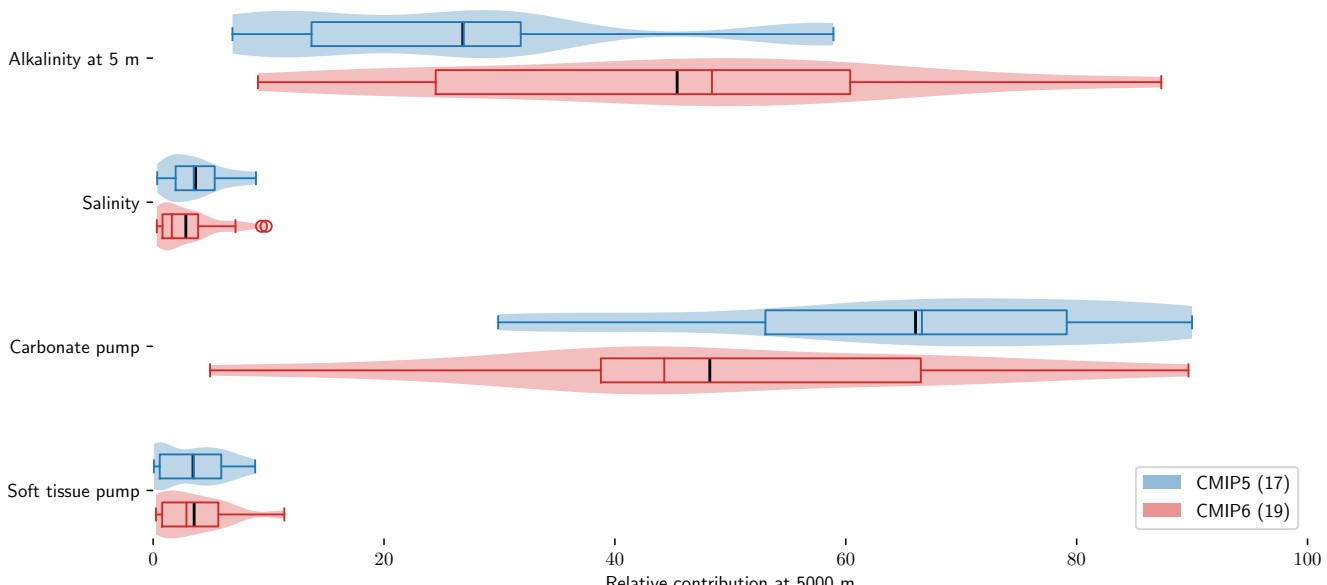

**Fig. 7: Relative contributions explaining the sAlk difference between the ESMs and the observations from GLODAPv2 at 5000 m.**
**Violin plots of the CMIP5 and CMIP6 ensembles with the ensemble mean (black tick) and the quantiles (boxes) displayed for each**
**component.**

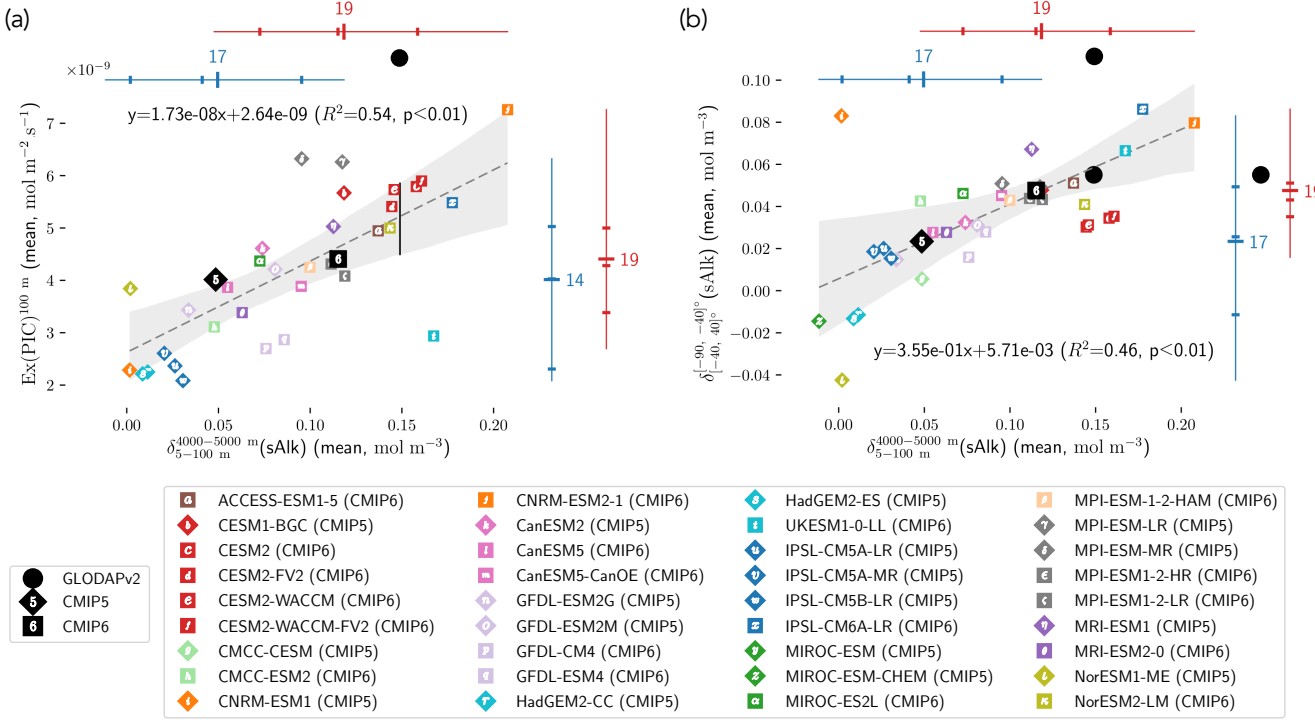

**Fig. 8: Inter-ESM relationships for the sAlk vertical profile. Relationship between the vertical gradient of sAlk and the PIC export**
**at 100 m (a) and the meridional surface sAlk difference between the Southern Ocean and the low latitudes (b). The marker notation**
**is the same as in Fig. 4 with the CMIP5 (resp. CMIP6) ESMs plotted with diamonds (resp. squares). To the right (resp. top), the**
**distribution for CMIP5 (in blue) and CMIP6 (in red) ESMs is displayed with the number of ESMs considered, the span of the values**





**(line), the average (major tick) and the quantiles (minor ticks). Observations from GLODAPv2 are marked with black circles. In**
**particular, it enables us to infer, from the estimated linear regression and the confidence interval at 95 %, a range of values for the**
**global PIC export at 100 m to be between 51 and 70 Tmol yr$^{-1}$ (a).**

### 3.3    CMIP6 improvements and persistent biases

The improvement in the representation of the carbonate pump not only strengthens the vertical gradient of sAlk, but also that of sDIC. Hence, not only has the sAlk vertical profile improved from CMIP5 to CMIP6 relative to the observations (Fig. 9a) but so too has the profile of sDIC and sAlk-sDIC (Fig. 9b,c). Although changes in the carbonate pump are largely
responsible for the improvement in the sAlk vertical profile, interestingly, they are also the main driver of the improvement in the sDIC profile, alongside changes in the soft tissue pump from CMIP5 to CMIP6. Whereas both sAlk and sDIC were on average in excess at the ocean surface in CMIP5, this bias has reduced in magnitude in CMIP6 and has also reversed with a slight negative bias in CMIP6 (see Sect. 3.2.1 for sAlk). In addition, although improved, we note that the vertical profiles of sAlk and sDIC are still significantly biased with regards to the observations. This vertical profile bias is consistent with a
persistent negative bias, with a slight deterioration from CMIP5 to CMIP6, in the open ocean mean Alk and DIC compared to GLODAPv2 (see Fig. C1). A partial explanation is discussed in Sect. 4.2.2. The use of the same $r_{Nut:P}$ for the observations and the ESMs (instead of respectively 21.8 and 17; see Section 2.5), would have driven a very minor offset in the bias associated with the carbonate pump estimate (e.g., a reduction of 0.009 mol m$^{-3}$ of the sAlk bias at 5000 m) without impacting the shape of the biases throughout the water column.
Despite the improvement in the representation of the sAlk and sDIC vertical profiles in CMIP6, the representation of the depth of the saturation horizons, both for calcite and aragonite, has slightly worsened. From CMIP5 to CMIP6, the downward shift of the saturation horizons has moved the ESMs away from the observations (Fig. 9). This results in an increased overestimation of the aragonite saturation horizon depth, too deep compared to the observations, and also a slight overestimation of the calcite saturation depth in CMIP6, whereas in the CMIP5 ensemble mean it was close to the observations.
Moreover, although the CMIP6 ensemble is more concentrated around the ensemble mean saturation depth of both calcite and aragonite, the range covered is still considerable. Focusing on the CMIP6 ensemble, the bias in the aragonite saturation horizon seems driven by excessively deep remineralization of organic matter, which leads to a peak in the soft tissue pump bias relative to the observations at around 900 m. This is consistent with an overestimation of the aragonite saturation horizon depth. In particular, it seems to be essentially driven by a biased signal in the Atlantic, mainly with regards to the Intermediate Waters
(see Fig. D5b). In contrast, the global bias for the calcite saturation depth in both CMIP5 and CMIP6 results from a partial compensation between a saturation depth in the equatorial Pacific Ocean that is too shallow and a saturation depth in the North Pacific that is too deep. These regional biases showed greater global compensation in CMIP5, although the individual regional biases have been reduced in CMIP6 (see Fig. D5b). It also seems that there is a lack of remineralization of organic matter and dissolution of CaCO$_3$ in the ESMs with a negative bias for both the carbonate and the soft tissue pump that increases at depth
for the CMIP5 and CMIP6 ensemble means. Although there is a slight deterioration in the representation of the saturation horizons from CMIP5 to CMIP6, they remain globally in agreement with the observations. This is due to compensation between the negative biases of the respective sAlk and sDIC vertical profiles, notably at depth. The resulting vertical profile biases of sAlk-sDIC, an approximation of the carbonate ion concentration, are effectively typically much lower than that of sAlk and sDIC with a noteworthy decrease in absolute biases associated with the carbonate and soft tissue pumps from CMIP5
to CMIP6 (Fig. 9c).



**Fig. 9: Assessing the biases of the vertical distribution of key variables relative to the observations from GLODAPv2. Relative contributions for the vertical difference of sAlk (a), sDIC (b) and sAlk-sDIC (c) vertical profiles for both CMIP5 (left panels) and**





CMIP6 (right panels) ensemble means in comparison with GLODAPv2. The black dotted (resp. dashed) lines refer to the CMIP5 (resp. CMIP6) ensemble means difference with the observations. The white horizontal lines refer to the saturation horizon depths for both calcite and aragonite for GLODAPv2 (solid), CMIP5 (dotted) and CMIP6 (dashed). The distributions of these saturation horizon depths are shown on the right in blue (resp. red) for the CMIP5 (resp. CMIP6) ensemble with the span of the values (line), the average (major tick) and the quantiles (minor ticks). The very thin lines in (c) correspond to the vertical profiles of the carbonate
ion concentration difference with the observations.Fig. 9

## 4    Discussion

### 4.1    CaCO₃ cycle model development from CMIP5 to CMIP6

In general, only limited modifications have been made with respect to the representation of the carbonate pump in the CMIP6 models compared to respective CMIP5 versions (Fig. 2, see also Fig. C1, Appendix C 6.3 and the Supplementary Table S1).
Such changes are insufficient to explain the increase in the intensity of the carbonate pump, and in the vertical Alk profile as seen from CMIP5 to CMIP6. Although a CaCO₃ cycle has been added to the BFM biogeochemical model, in other models, parameterizations generally changed little between the two CMIP exercises. Improvements in the range of processes that can be represented with respect to the CaCO₃ cycle are limited and model dependent. Certain ESM groups have changed their embedded ocean biogeochemical model between CMIP5 and CMIP6 with consequent changes in the CaCO₃ cycle scheme.
For example, the transition from TOPAZ2 to COBALTv2, which includes enhanced resolution of the plankton food web, in NOAA-GFDL has changed the parameterization of aragonite and calcite production. One trend is towards a more complete representation of the fate of PIC at the seafloor in CMIP6 with the expansion of the use of sediment modules, to at least partly balance the global ocean Alk content. This indicates that most of the model performance changes from CMIP5 to CMIP6 are likely associated with parameter tuning, or *ad hoc* settings, and potentially with a general increase in the horizontal and vertical
model resolution and improved representation of ocean circulation (Séférian et al., 2020).

### 4.2    Inconsistencies in protocols and future recommendations

This analysis and review of the modelling schemes has provided insight into the protocols followed by the modelling groups and the implications this has on ESM outputs, leading us to make several recommendations for the ocean biogeochemical modelling community.

#### 4.2.1    Drift and spin-up

The drift that we assess in the ESMs is low enough to have minimal influence on our non-drift corrected results centered on 2002 (see Sect. 2.2.1; Fig. 10 and see also Fig. D6Fig. D7), but the influence of these drifts remains possible in the projections and was not evaluated in this study. For instance, the model drift per century of the sAlk and sDIC vertical gradients is less than 5 % of the observed vertical gradients. Similarly, model drifts in surface ocean salinity, temperature and
exports at 100 m are also limited and should have minimal influence on the results in part due to salinity normalization. The largest drifts were observed for CMCC-CESM (CMIP5) and CNRM-ESM1 (CMIP5); specifically, the global DIC inventory of CMCC-CESM (CMIP5) had not reached equilibrium prior to the Historical simulation. In CMIP6, these two groups have especially increased the spin-up duration although the relative part of their online spin-up has decreased.

There is great diversity with regards to the spin-up strategy employed by the different groups (Séférian et al., 2016).
Séférian et al. (2020) discussed this and pointed out that the spin-up duration has increased for all groups except IPSL and NOAA-GFDL. For these two groups, the considerable increase in resolution from CMIP5 to CMIP6 was balanced against a reduced spin-up duration, as well as the completion of a fully online spin-up in the case of IPSL. Finally, we highlight two contrasting spin-up strategies with consequences on the mean ocean state of Alk and the CaCO₃ cycle. In MPI-M ESMs for CMIP5, the model was initialized with the same Alk and DIC values in all ocean grid cells and, to reduce the spin-up duration,
Alk was indirectly tuned to achieve a consistent representation of the ocean CO₂ sink. This was achieved through increasing weathering fluxes and the CaCO₃ content in sediments, leading to an increase in Alk and DIC to maintain the desired $p$CO₂ field (Ilyina et al., 2013b). This explains the strong offset in both Alk and DIC content for MPI-M in CMIP5 (see Fig. C1). An





alternative strategy was developed by NCAR for CMIP6 regarding the balancing of the global ocean budget of Alk. During the spin-up, the saturation state threshold for the burial of $CaCO_3$ was tuned to balance the loss of Alk from the burial of $CaCO_3$ and the riverine input of Alk before starting the experimental simulations (Long et al., 2021). This resulted in the choice of an unusual threshold for the burial of $CaCO_3$ in their runs (see MARBL in Appendix C 6.3). Similarly, NOAA-GFDL for CMIP6 in COBALTv2 set sediment calcite concentrations such that Alk lost through calcite burial balanced river Alk inputs at a certain year during the spin-up (Dunne et al., 2012). While it is probably advisable that model groups continue to work to balance the Alk budget at quasi-steady state, observations suggest there may have been a slight net sink of Alk during the Holocene and therefore potentially an ocean carbon source to the atmosphere (Ciais et al., 2013; Cartapanis et al., 2018). The strategy of maintaining a degree of freedom at the seafloor during the spin-up with the tuning of parameters associated with the $CaCO_3$ sedimentation processes at the bottom of the ocean seems to be quite relevant to balance the overall Alk budget at equilibrium. However, given the difficulty in running ESM simulations to equilibrium during the model development process, it is likely that drift correction of ocean $CO_2$ system variables will continue to be a requisite of robust ESM intercomparisons.

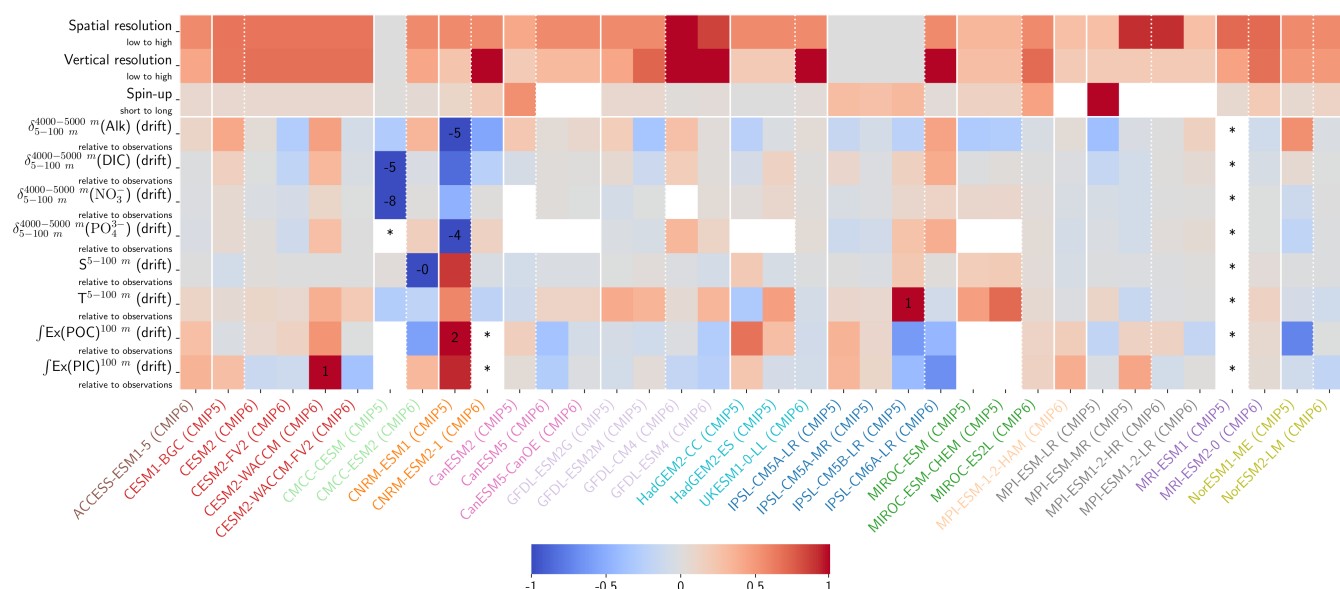

**Fig. 10: Evaluation of the drift in both CMIP5 and CMIP6 ensembles.** The metrics displayed refer to both the resolution and the spin-up (top) and a drift evaluation of key variables considered in this analysis (bottom). For each metric, the name is given to the left of the table with a characterization in small. Concerning the metrics associated with the ESM resolution and the spin-up, the shading was normalized between 0 and 1 so that the maximum and minimum values correspond to respectively 0 and 1. The higher resolution (resp. longer spin-up), the darker the cell is. A more complete representation of this section of the figure is available in Fig. C1 and more information can be found in the Supplementary Table S1. In contrast, for the metrics associated with the drifts, the shading was normalized between -1 and 1 so that the maximum of the absolute value corresponds to an extremity of the colorbar centered on 0. For each row, the black figure given for the ESM with the highest drift corresponds to the percentage that the drift represents relative to the value of the observations from GLODAPv2, or estimates from Devries and Weber (2017) for the POC export and from Sulpis et al. (2021) for the PIC export – given at 300 m though. For each metric associated with the drift, the better the model does, the lighter the cell is. ESMs with an issue are marked with an asterisk. Note that both the POC and PIC exports at 100 m for CNRM-ESM2-1 (CMIP6) were not included – since it completely erases the other values with the normalization – due to intermittent extreme values localized in the Japan Sea, which can influence the means and might also partly feed its very high PIC export at 100 m (Fig. 6).

## 4.2.2 Alk and DIC initialization

As previously discussed, drift can depend not only on the spin-up strategy but also on the initialization strategy. Indeed, it is interesting to examine the initialization of Alk and DIC in ESMs, and to assess how this may influence model performance. Alk and DIC fields are recommended to be initialized using the second version of the GLODAP product (GLODAPv2, Lauvset



et al., 2016) following the OMIP-BGC protocol for CMIP6 (Orr et al., 2017). For CMIP5, many groups initialized with the
first gridded version (GLODAPv1, Key et al., 2004) even though this was not specific to a protocol. GLODAPv2 surface fields
are more heterogeneous than GLODAPv1, but it is mainly in the coastal ocean and in particular in the Arctic where the two
datasets diverge due to the addition of new observations.

At a global scale, the difference between the GLODAP products does not appear to have a significant impact on the
representation of both present-day Alk and DIC (Fig. 11a,b and see also Appendix E 6.5). Thus, neither the change in the
GLODAP mapping method nor the increase in the number of observations are responsible for the improvement in the Alk
representation from CMIP5 to CMIP6. In fact, a number of groups did not follow the OMIP-BGC protocol and instead
continued using GLODAPv1 to initialize their CMIP6 models (see the Supplementary Table S1).

Surprisingly, it is assumptions in the conversion of the GLODAP Alk field from gravimetric to volumetric units that
contribute to differences between ocean Alk distributions across ESMs. The Alk field is provided in µmol kg$^{-1}$ in the GLODAP
mapped product, and must, with the exception of the NOAA-GFDL models, be converted to volumetric units (e.g., mol m$^{-3}$)
to be used by the marine biogeochemical models. Multiple approaches are employed by the groups when performing this
conversion. For most of the ESMs, the conversion was made using a constant seawater density, which itself varies between
ESMs (from 1,024 kg m$^{-3}$ for ACCESS-ESM1-5 (CMIP6) to 1,028 kg m$^{-3}$ for CNRM-CERFACS, IPSL and MIROC; see Fig.
C1). Although no conversion strategy was recommended in the biogeochemical protocol for CMIP6 (Orr et al., 2017) this
should be performed using the *in situ* seawater density rather than a constant density or the potential density, both of which
produce conversion biases. The impact of not using i*n situ* density on sAlk and sDIC vertical profiles is a bias at depth, as the
difference between *in situ* and potential density increases (Fig. 11c). Although global mean Alk and DIC are only reduced by
1 % when using the potential density for conversion (respectively -0.023 mol m$^{-3}$ for Alk and -0.022 mol m$^{-3}$ for DIC), the
vertical gradient of sDIC between the surface and 5500 m decreases by 15 % and that of sAlk decreases by 32 %. Erroneous
conversion from gravimetric to volumetric units can therefore affect the evaluation of a model that attempts to reproduce the
observed Alk profile by adjusting the parameters of the carbonate pump. It can also impact strategies developed to estimate
the PIC export that are based on the Alk vertical gradient (e.g., Battaglia et al., 2016).

Further investigation reveals that the method of volumetric conversion for the initial Alk field drives a surface Alk
bias that influences both surface DIC and the global DIC content of the ocean. The potential influence of different Alk
volumetric conversion methods on the ocean $CO_2$ system has been previously reported (Orr and Epitalon, 2015). Although
this bias in the initialization is confined to depth, it results in a perturbed surface ocean $CO_2$ system once the ESM reaches
quasi-steady state. Indeed, in an ocean with a conservative Alk inventory, the biological pump, and in particular the carbonate
pump, only influence the distribution of Alk. Whatever the initialization of ocean DIC, the Alk content of the ocean is fixed at
its initial value. As a result, without taking into account salinity and temperature, the equilibrium between surface Alk, DIC
and atmospheric $p$CO$_2$ tends to set the surface DIC at the end of spin-up through air-sea carbon fluxes. The global content of
DIC is therefore directly impacted by surface Alk after spin-up with an atmosphere considered an infinite carbon reservoir.
Finally, although the NOAA-GFDL models avoid initialization issues by keeping tracers in gravimetric units, the use of a
constant density of 1,035 kg m$^{-3}$ during post-processing, to produce 'talk' and 'dissic' fields, distorts the ocean $CO_2$ system
from the real ESM outputs, in particular with excessive surface values (see Fig. D1).

In addition to correct initialization of the global Alk inventory, initialization with a spatial Alk distribution consistent
with observations is also required to allow accurate computation of the ocean $CO_2$ system and likely air-sea carbon fluxes (see
*mocsy 2.0*; Orr and Epitalon, 2015). Indeed, the ocean biogeochemical models are generally only able to directly affect the
alkalinity components associated with carbonates, phosphates, and sometimes silicates, while the other components are
computed from pH (water alkalinity), salinity (fluoride alkalinity) or both (borate; Uppström, 1974; Lee et al., 2010). Thus, an
incorrect initialization of the Alk distribution (e.g., MPI ESMs in CMIP5 with a constant mean value) could indirectly rearrange
the partitioning of Alk between its different components, especially those unaffected directly by biogeochemical processes in
the models. Yet, the influence of even small biases in borate alkalinity have been shown to have non-negligible effects on the
ocean $CO_2$ system (Orr and Epitalon, 2015). In summary, correctly initializing model Alk reduces biases in the representation
of surface ocean carbonate chemistry. This requires the use of *in situ* density for the conversion of three-dimensional Alk fields
from the GLODAP product.



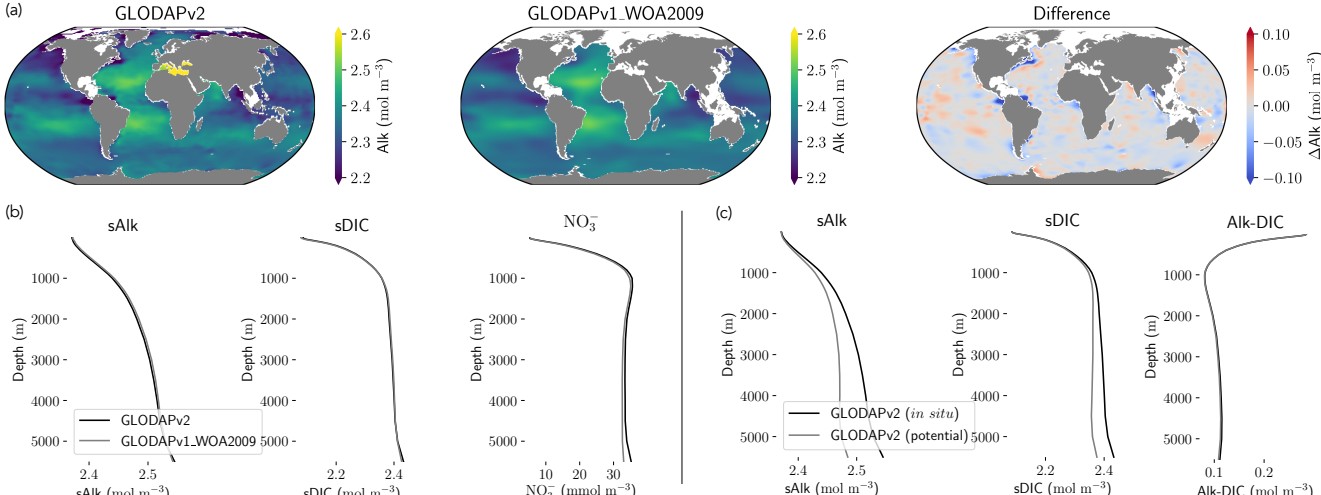

**Fig. 11: Influence of the choice of the observations and the conversion strategy for Alk and DIC. (a) Maps of the surface ocean Alk**
**for (i) GLODAPv2 and (ii) GLODAPv1_WOA2009 and their difference, (i)-(ii). (b) Vertical profiles of sAlk, sDIC and nitrate**
**concentration for the open ocean to compare GLODAPv2 and GLODAPv1_WOA2009. (c) Vertical profiles of sAlk, sDIC and Alk-**
**DIC for the open ocean to compare GLODAPv2 data converted using *in situ* or potential density.**

### 4.2.3 Improving model assessment and traceability

The different strategies for initializing Alk and DIC highlight the importance of clear data sharing and precise protocols
to enable robust ESM assessments and intercomparisons. In the following, we recommend increasing the priority of certain
variables in future CMIP exercises (Orr et al., 2017). First, we suggest that in future intercomparisons, model groups share the
three-dimensional export of POC and PIC ('expc', 'expcalc' and 'exparag'), and not only the exports at 100 m, as some groups
have done for CMIP6. This would enable a more consistent estimate of POC and PIC export, as well as the resulting rain ratio.
Indeed, the potential inaccuracy of soft tissue pump estimates based on fixed-depth POC export has been previously discussed
in the literature (e.g., Buesseler et al., 2020; Koeve et al., 2020). The simulated increase in sDIC in the upper 100 m of the
water column, despite relatively consistent sAlk, indicates that net remineralization of POC is occurring in much shallower
waters than net dissolution of PIC in the ESMs. As such, POC export values should be used with caution when assessing the
rain ratios (see Fig. 6). Sharing of the export fields would ideally be accompanied by three-dimensional fields of
remineralization and dissolution ('remoc', 'dcalc' and 'darag') to facilitate analysis of processes such as the biological pump
throughout the water column. Finally, sharing of vertically integrated calcite and aragonite production ('intpcalcite' and
'intparag') and POC and PIC burial ('froc' and 'fric') would also improve assessments of the influence of the biological pump
on vertical DIC and Alk profiles (see Fig. 9a,b). Fig. 8Alongside the absence of certain model outputs, one issue that has
presented itself throughout our analysis is a lack of model traceability between CMIP5 and CMIP6. In the absence of
publications documenting model changes, it is typically not possible to trace ocean biogeochemical model developments
without contacting individual developers and in certain instances asking for the model code. To address this, we propose that
developers utilize a common online platform to share their code and provide an associated model guide. Such a platform would
critically improve model traceability, enhancing ocean biogeochemical model transparency and accessibility (e.g., sharing
river discharge values for both Alk and DIC). Within this context, the Earth System Documentation (ES-DOC, https://es-
doc.org) project is potentially highly relevant. However, in its current form, the tool is broadly insufficient for specific studies
due to the paucity of ESMs participating and the level of model documentation provided.

### 4.3 Implications of the improved Alk representation in CMIP6

Although no major trend emerges in terms of evolution from CMIP5 to CMIP6 with regards to the modelling schemes (see
Sect. 4.1), there is nevertheless an improvement in the representation of Alk associated with a strengthened carbonate pump



(see Fig. 1). Here we discuss the potential implications that this improvement may have on the ocean response to anthropogenic
carbon emissions considering potential $CO_2$-feedbacks and the impact on ocean acidification projections.

### 4.3.1    Ocean carbon uptake and ocean acidification

In a $CO_2$-concentration–driven simulation, the surface Alk and DIC are directly connected to each other through
equilibration via air-sea $CO_2$ fluxes. As a result, a modification of global scale surface Alk has a direct effect on surface DIC
(Fig. 12a). Indeed, neglecting the effect of temperature and salinity on the partial pressure of $CO_2$ at the ocean surface, we can
differentiate $pCO_2$ as follows:

$$dpCO_2 = \frac{\partial pCO_2}{\partial Alk}\bigg|_{DIC,S,T} \cdot dAlk + \frac{\partial pCO_2}{\partial DIC}\bigg|_{Alk,S,T} \cdot dDIC \qquad (14)$$

where $pCO_2$, Alk and DIC all refer to surface values and the partial differentials are both at fixed temperature and salinity.
Rewriting the differentials 'd' as the difference 'Δ' between two ESMs gives:

$$\Delta DIC = -\overline{\frac{\partial pCO_2}{\partial Alk} \cdot \frac{\partial DIC}{\partial pCO_2}} \cdot \Delta Alk + \overline{\frac{\partial DIC}{\partial pCO_2}} \cdot \Delta pCO_2 \qquad (15)$$

where the overbars correspond to the mean surface ocean values for the partial differentials. Yet, at the global scale, we can
assume that for a surface ocean in balance with atmospheric $pCO_2$, for any ESM, $\Delta pCO_2=0$. Consequently, surface differences
in Alk and DIC between two ESMs are linearly related. Approximating the carbonate ion concentration to Alk-DIC, and using
the expression for $pCO_2$ given in Sarmiento and Gruber (2006), this results in:

$$\Delta DIC \simeq \frac{Alk}{3 \cdot Alk - 2 \cdot DIC} \cdot \Delta Alk \qquad (16)$$

at the ocean surface. Substituting the mean surface ocean Alk and DIC of the combined CMIP5 and CMIP6 ensembles, this
gives $\Delta DIC \simeq 0.81 \cdot \Delta Alk$, and indeed a very similar relationship between surface ocean Alk and DIC anomalies for individual
ESMs relative to CMIP5 ensemble mean values is found ($\Delta DIC \simeq 0.842(\pm 0.009) \cdot \Delta Alk$; $R^2=0.98$, p<0.01; Fig. 12a).
This relationship between anomalies in surface Alk and DIC has implications for the wider surface ocean $CO_2$ system.
As the slope associated to the linear regression is less than 1, an ESM with higher surface Alk will tend to have a higher Alk-
DIC and therefore higher surface concentration of $CO_3^{2-}$ and pH (Fig. 12a). The decrease in surface Alk from CMIP5 to CMIP6
therefore results in a slight decrease in pH, generally lower calcite and aragonite saturation state in CMIP6, with the global
surface ocean carbonate ion concentration decreasing by 2.9 ± 2.7 % and up to 5 % in certain regions (Fig. 12b). As the
timescale of air-sea $CO_2$ exchange can be approximated as proportional to the carbonate ion concentration (Sarmiento and
Gruber, 2006), other factors being equal, CMIP6 mean surface ocean $pCO2$ is likely to equilibrate faster with the atmosphere
than that of CMIP5. An exception to this is the Southern Ocean, where upwelled deep waters are far from equilibrium with the
atmosphere. In CMIP6, enhanced carbonate dissolution at depth results in upwelled Southern Ocean waters with a higher
carbonate ion concentration implying Southern Ocean $pCO2$ has a longer equilibration timescale. While the change in the
carbonate pump from CMIP5 to CMIP6 seems to have a slight effect on the representation of the present-day ocean $CO_2$ system
at the surface, it is likely to have negligible feedback on the projected ocean carbon sink with an overall decrease in the Revelle
factor ($\gamma_{DIC}$) of 0.2 ± 1.3 % (Fig. 12b). Besides, the maximum potential influence of the carbonate pump and in turn surface
Alk on the uptake of anthropogenic carbon over the historical era has previously been estimated as 5 % (Murnane et al. 1999).
However, in the equatorial Pacific, where upwelling variability induced by the El Nino-Southern Oscillation (ENSO) strongly
modulates surface concentrations of DIC and Alk, accurately reproducing the observed Alk vertical gradient in ESMs is
important to correctly simulate the observed interannual variability of $CO_2$ fluxes in this region (i.e., anomalously outgassing
during El Niño and ingassing during La Niña events; Feely et al., 2006). In a recent study, Vaittinada Ayar et al. (2022) show
that the mean state of the Alk vertical profile in the tropical Pacific influences both projections of ENSO-driven $CO_2$ fluxes
and long-term carbon uptake in the region.




Finally, the trend in surface Alk and DIC from CMIP5 to CMIP6 also influences spatial heterogeneity, especially between the mid-latitudes ([-40, 40]°) and the Southern Ocean ([-90, -40]°) with enhanced meridional surface gradients of Alk and DIC in CMIP6 compared to CMIP5 (respectively +0.024 and +0.013 mol m$^{-3}$; Fig. 13). Neglecting model differences in

ocean dynamics we can estimate that differences in the amplitude of the soft tissue and carbonate pumps between CMIP6 and CMIP5 impact the meridional surface gradients of DIC and Alk. To estimate this effect, we define an attenuation coefficient ($\alpha$) as the ratio between the meridional surface Alk gradient at the surface ($\delta^{Southern\text{-}midlat}sAlk^{5\,m}$) and the vertical open ocean Alk gradient ($\delta^{5000\text{-}5\,m}sAlk$):

$$\alpha = \frac{(\delta^{Southern-midlat}sAlk^{5\,m})_{CMIP6} - (\delta^{Southern-midlat}sAlk^{5\,m})_{CMIP5}}{(\delta^{5000-5\,m}sAlk)_{CMIP6} - (\delta^{5000-5\,m}sAlk)_{CMIP5}} \tag{17}$$

We associate this coefficient with the upwelling that determines the vertical Alk gradient in the Southern Ocean. Hence, by multiplying the deviations of both the soft tissue and carbonate pumps at depth for CMIP6 relative to CMIP5 by this attenuation coefficient, we are able to trace the origin of the changes in the meridional surface gradients of Alk and DIC from CMIP5 to CMIP6; the remaining component being attributed to the gas exchange pump and to anthropogenic carbon uptake (Fig. 13). This highlights that the increase in the carbonate pump from CMIP5 to CMIP6 is the main driver of the enhanced meridional

surface gradients of DIC and Alk.

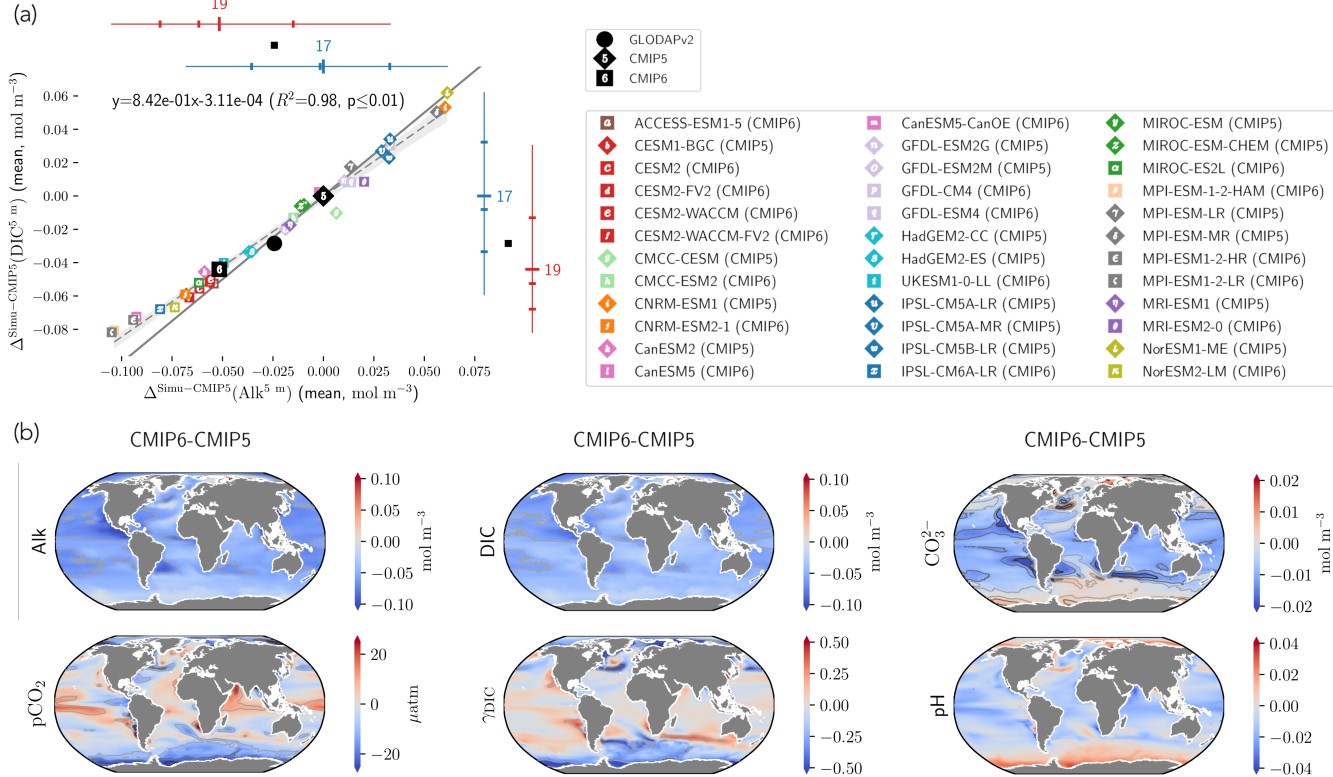

**Fig. 12: Carbonate chemistry at the ocean surface. (a) Connection between the surface Alk and DIC open ocean means. The caption is the same as in Fig. 4 and 8 with the CMIP5 (resp. CMIP6) ESMs plotted with diamonds (resp. squares). To the right (resp. top),**
**the distribution for CMIP5 (in blue) and CMIP6 (in red) ESMs is displayed with the number of ESMs considered, the span of the values (line), the average (major tick) and the quantiles (minor ticks). Observations from GLODAPv2 are marked with black circles. The dashed line corresponds to the linear regression and the solid line refers to an equivalent Alk-DIC as the CMIP5 ensemble mean. (b) Maps of the surface difference between the CMIP6 and CMIP5 ensemble means for key variables of the carbonate system. The contours from light to dark represent the relative variation in absolute value at 2.5, 5.0 and 7.5 %.**



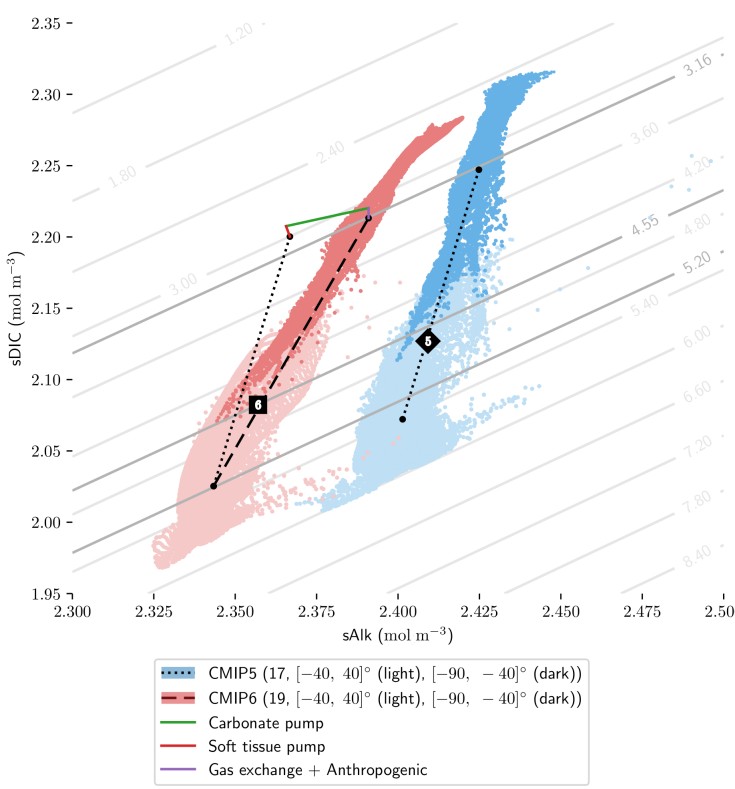


**Fig. 13: Carbonate chemistry at the ocean surface (continuation of Fig. 12). Scatter plot of the surface sDIC and sAlk at mid-latitudes (light) and in the Southern Ocean (dark) for the CMIP5 and CMIP6 ensemble means, respectively in blue and red. The open ocean average is pointed with a diamond (CMIP5) or a square (CMIP6) and the meridional gradient between the average for the mid-latitudes and the one for the Southern Ocean is displayed with a dotted line for CMIP5 and a dashed one for CMIP6. The additional**

**lines plotted for CMIP6 enable to explain how the meridional gradient has increased from CMIP5 to CMIP6 distinguishing the biological pumps and a residual component. The background grey lines give an idea of the calcite saturation state calculated with the mean open ocean surface temperature, salinity, silicate and phosphate.**

### 4.3.2    Transient changes in the ocean Alk budget and its distribution

A quasi-equilibrium of Alk on centennial timescales is commonly accepted, but observational data have only recently allowed

the overall global Alk budget to be closed (Middelburg et al., 2020). The riverine input of Alk is mainly balanced by the burial of PIC, but also to a lesser extent, by the remobilization of sediments (essentially through submarine weathering and anaerobic remineralization of organic matter) and the burial of organic matter. Interplay between these fluxes makes Alk central to the understanding of the processes driving atmospheric $p$CO$_2$ over glacial and interglacial cycles (Archer and Maier-Reimer, 1994; Kerr et al., 2017; Boudreau et al., 2018). In modelling studies, drift in the Alk inventory would induce drift in the surface air-

sea carbon flux to maintain a surface ocean in equilibrium with atmospheric $p$CO$_2$. However, anthropogenic perturbation of the carbon cycle, as well as the effect of climate change, could cause transient changes in ocean Alk budget and its vertical distribution. The timescales over which such perturbations in Alk budget and distribution will occur, and the potential consequences for the projections of anthropogenic carbon uptake, are not clear.

Ocean Alk might increase through enhanced terrestrial rock weathering and an associated increase in riverine input

in response to climatic drivers (e.g., enhanced precipitation, permafrost thaw; Raymond and Cole, 2003; Drake et al., 2018). In addition, shoaling of the saturation horizon due to ocean acidification is thought to explain recent observations of enhanced CaCO$_3$ dissolution at the seafloor (Sulpis et al., 2018) initiating chemical carbonate compensation. This highlights the potential importance of representing sediment processes in models given that CaCO$_3$ dissolution enriches waters in Alk and can therefore





enhance ocean carbon uptake when these waters are recirculated to the surface ocean (Archer et al., 1998; Gehlen et al., 2008).
Potential dissolution of coral reef CaCO₃ in response to climate change (Cooley et al., 2022) may also increase Alk on centennial timescales – echoing the coral reef hypothesis and its potential effect on atmospheric $p$CO₂ for glacial-interglacial transitions (Berger, 1982; Opdyke and Walker, 1992).

   Finally, the possibility of implementing large scale ocean Alk enhancement (OAE, Renforth and Henderson, 2017; Bach et al., 2019) could increase the global Alk inventory on relatively short timescales. Representation of CaCO₃ burial and
submarine weathering is necessary in simulations of OAE, even on decadal to centennial timescales since it could entail abrupt changes. To date OAE model studies have typically directly enhanced surface ocean Alk (Köhler et al., 2013; Ilyina et al., 2013a; Hauck et al., 2016; ; González and Ilyina, 2016; Lenton et al., 2018; González et al., 2018), however in reality Alk is likely to be provided via the addition of a mineral such as olivine, of which only a fraction will dissolve in the surface ocean. The explicit simulation of alkaline mineral addition could enhance Alk at depth, deepening the carbonate saturation horizon
and suppressing the dissolution of sediment carbonate, that might otherwise occur, as well as increasing the burial of sinking CaCO₃.

   A number of the strategies employed to close the Alk budget in ESMs, are potentially questionable in simulations, where the global Alk inventory may not be in quasi-equilibrium (Keller et al., 2018). Indeed, in CMOC and CanOE, CaCO₃ burial at the seafloor is redissolved at the ocean surface locally to close the Alk budget. This parameterization is intended to
represent fluvial Alk sources (Christian et al., 2022), but does not impact the spatial distribution of surface Alk in the same manner as riverine fluxes. It also makes the riverine discharge of Alk dependent on the carbonate pump in projections. As previously discussed, in CMIP6, NCAR and NOAA-GFDL with COBALTv2 control the Alk balance at depth, by tuning respectively the calcite saturation state threshold for burial and the sediment calcite concentration in order to balance the loss of Alk through burial and the gain through river inputs; same for IPSL and CNRM in CMIP5 with PISCESv1, which forced
this balance, not explicitly taking into account the processes at the sediment interface. Other ESMs dissolve the entirety of the PIC that reaches the seafloor in the last ocean level, effectively avoiding the consideration of an Alk burial sink (WOMBAT, BEC, diat-HadOCC and NPZD-MRI). Finally, for the models that consider a sediment module, this might complicate the control of the global Alk budget under preindustrial conditions. On the other hand, the approach used by some groups (CNRM-CERFACS and IPSL in CMIP5 and CMIP6) to restore the global Alk inventory to ensure its conservation can mask Alk budget
imbalances and potentially bias Alk vertical profiles. In particular, this could lead to drifts if Alk is no longer restored after spin-up (CNRM-CERFACS in CMIP6).

### 4.3.3  Potential changes in the carbonate pump

An ESM representation of the carbonate pump that includes ocean acidification and climate change sensitivities, has the potential to produce CaCO₃ cycle climate feedback in centennial projections. Such a model would require a relatively high
level of biological realism, taking into account the complexity of the response of the CaCO₃ cycle to environmental stressors (Gattuso and Hansson, 2011; Schlunegger et al., 2019). Most studies suggest a negligible CaCO₃ cycle climate feedback this century although on longer timescales this feedback can be more important (e.g., Gehlen et al., 2007; Ridgwell et al., 2007; Schmittner et al., 2008; Hofmann and Schellnhuber, 2009; Gangstø et al., 2011; Pinsonneault et al., 2012; Krumhardt et al., 2019). However, uncertainties and diverse responses of calcifying organisms to environmental changes (e.g., Kroeker et al.,
2013) make it difficult to constrain model parameterizations with confidence. Furthermore, most studies only consider a subset of potential impacts on the CaCO₃ cycle. Here we discuss current ESMs, highlighting developments that may affect the CaCO₃ cycle climate feedback.

   Whereas projected ocean acidification and climate change impacts (e.g., warming and lower upper ocean nutrient availability) have diverse effects on calcification, meta-analyses generally agree on an expected decrease in calcification due
to ocean acidification (Kroeker et al., 2013; Meyer and Riebesell, 2015; Seifert et al., 2020). This would lead to an increase in surface Alk and a decrease in Alk at depth, with possible effects on anthropogenic carbon uptake. The increased energetic cost of calcification could also impact the high diversity of pelagic calcifiers and their niches (e.g., for the coccolithophores; Monteiro et al., 2016). While the representation of saturation state dependent calcification was previously prioritized (Ridgwell et al., 2009; Gehlen et al., 2007; Gangstø et al., 2008, 2011; Hofmann and Schellnhuber, 2009; Pinsonneault et al., 2012), this
is no longer the case. NOAA-GFDL and MOHC in CMIP6 are the only groups in this intercomparison to consider such a





dependence. We recommend that the other groups follow their lead to improve the realism of the carbonate pump response to anthropogenic emissions.

The implicit representation of pelagic calcification in current ESMs depends on the fate of organic matter and therefore indirectly on net primary production (NPP). However projected NPP changes this century are highly uncertain across 860 the CMIP5/6 ensembles (Kwiatkowski et al., 2020). For all ESMs any environmental impact on planktonic calcifiers does not affect associated organic matter production. Moreover changes in the growth rate of calcifiers does not impact $CaCO_3$ production, with the exception of BEC and MARBL, in which an implicit calcite pool is considered in the phytoplankton group. The independence and potential influence of calcifiers on organic matter production (Gattuso and Hansson, 2011) would require an explicit representation through one or more PFTs. Explicit pelagic calcification was recently implemented in 865 MARBL and PlankTOM respectively for coccolithophores (Krumhardt et al., 2019) or for the three main planktonic calcifier groups: coccolithophores, foraminifers and pteropods (Buitenhuis et al., 2019). However, the notable challenge to represent both organic carbon and carbonate biomasses at the same time echoes the scattered data products available (see Marine Ecosystem Model Inter-comparison Project (MAREMIP) papers; O'Brien, 2012; Bednaršek et al., 2012; Schiebel and Movellan, 2012). Finally, a comprehensive representation of the carbonate pump, and thus Alk, in coastal areas, even in ESMs, 870 will probably require the representation of benthic calcifiers as they typically dominate coastal calcification (O'Mara and Dunne, 2019; Middelburg et al., 2020). This would also be valuable within the context of climate change and ocean acidification ecosystem impact projections. CSIRO recently included benthic carbonate production for regional applications addressing the Great Barrier Reef (Steven et al., 2019), and NOAA-GFDL have developed neritic environments, such as coral reefs and carbonate-rich/poor shelves (O'Mara and Dunne, 2019). However no benthic calcification is currently represented 875 in ESMs. Although $CaCO_3$ dissolution is essentially abiotic, certain models explicitly represent the sinking of $CaCO_3$ with dissolution dependent on the saturation state and therefore sensitive to ocean acidification. In particular, they represent dissolution with a linear dependence on the saturation state in undersaturated waters (see Sect. 3.1.2) although a consensus seems to emerge from laboratory studies for an exponent >1 in the water column (most likely around 3 to 4 for calcite), while keeping a linear dependence at the seafloor, diffusion being limiting for the process (e.g., Subhas et al., 2015; Sulpis et al., 880 2017; Boudreau et al., 2020). This mismatch between laboratory experiment results and model parameterization would increase dissolution in less undersaturated waters, and thus in shallower waters, in the models compared to laboratory experiments. This could partially compensate for the lack of pelagic aragonite production in most models, a carbonate mineral with a lower thermodynamic stability than calcite. Besides, this absence of aragonite representation could influence projections in ESMs of how the distribution of Alk responds to acidification. The shoaling of the saturation horizon should increase Alk at depth with 885 a reduction in $CaCO_3$ burial (see Sect. 4.3.2) but also change the Alk vertical distribution with an increase in dissolution, and thus Alk, in sub-surface waters.

There is no consensus on the interaction between POC and PIC in the ocean interior considering a protection and/or ballast effect (see Sect. 3.1.3 and Fig. 2Fig. 2). Observational and laboratory data diverge (Klaas and Archer, 2002; Passow and De La Rocha, 2006; De La Rocha and Passow, 2007; Lee et al., 2009; Engel et al., 2009; Moriceau et al., 2009), but a 890 recent expedition dataset brings it back to the forefront, especially explaining dissolution in shallow supersaturated waters with a PIC microenvironment influenced by POC remineralization (Subhas et al., 2022). Potential coupling between POC and PIC export may have side effects in a climate change scenario with a decrease in $CaCO_3$ production and export, modifying organic matter remineralization (Hofmann and Schellnhuber, 2009).

## 5    Conclusion

Our assessment of key properties of marine biogeochemical models involved in the 5[th] and 6[th] phases of CMIP highlights the diverse representation of processes associated with the carbonate pump. CMIP6 models simulate implicit pelagic calcification, no benthic calcification and generally calcite but not aragonite. In contrast, sinking and dissolution of $CaCO_3$ particles are represented both implicitly and explicitly, and variably sensitive to the local seawater saturation state. The fate of PIC reaching the seafloor also differs between models due to differences in external Alk sources such as riverine fluxes and the need to 900 conserve the global Alk inventory. We mostly report sparse, and generally singular, developments from CMIP5 to CMIP6

Our inter-ESM analysis reveals an improvement in the representation of the Alk distribution as compared to observations. In particular, the surface distribution and mean vertical profile of sAlk show improvements in CMIP6. This is





consistent with a strengthened carbonate pump resulting from a global increase in PIC export at 100 m from CMIP5 to CMIP6.
Despite such improvements, both the sAlk vertical gradient and PIC export remain globally underestimated in the CMIP6
ensemble compared to observationally constrained estimates. We were able to constrain a PIC export estimate of 51-70 Tmol
yr$^{-1}$ at 100 m for the ESMs to match with the observed vertical profile of Alk. The increase in the carbonate pump from CMIP5
to CMIP6 is not only the main driver of the improvement in the representation of the sAlk vertical profile, but also of the
vertical profile of sDIC, although the representation of the saturation horizons has slightly worsened. In addition, the shift
towards lower surface values of sAlk results in lower surface values of sDIC, producing a slight difference in surface ocean
carbonate chemistry between the CMIP ensembles. Specifically, the CMIP6 ESMs tend to have slightly lower surface ocean
pH and carbonate concentrations and exhibit enhanced meridional surface gradients in sAlk and sDIC. The changes in the Alk
distribution in CMIP6, however, have negligible impact on the simulated Revelle factor and therefore are likely to have little
effect on the magnitude of the projected ocean carbon sink for a given emissions scenario.

Although the limitation associated with the Alk and DIC observations fades overall with increasing measurements, it
is a rather biological limitation that seems to hold us back here in the effort to be more realistic in representing the carbonate
pump. Our incomplete mechanistic understanding of the $CaCO_3$ cycle gives rise to uncertainties for the carbonate pump in the
face of ocean acidification and climate change that are certainly notable, but difficult to estimate with current models.
Concerted work with biologists and observers seems necessary to improve or develop parameterizations (e.g., saturation state
dependency, POC and PIC coupling, benthic production), and then allow for a robust inter-ESM analysis of the response and
feedback of the carbonate pump to projected anthropogenic emissions.

Finally, future marine biogeochemical model intercomparison projects would benefit from the provision of additional
model outputs, notably three-dimensional fields of organic and inorganic export fluxes, remineralization, dissolution,
integrated organic and inorganic carbon production, and PIC and POC burial. Greater harmonization of initialization protocols
with respect to Alk would avoid model biases arising from gravimetric to volumetric unit conversions as well as long-term
drift. Finally, model traceability could be substantially improved if a shared platform were utilized to document changes to
ocean biogeochemical models and the assumptions or observations underlying these developments.

# 6    Appendices

## 6.1    Appendix A: Open ocean mask

Throughout our analysis the open ocean was defined as >250 km from the coast, neglecting small islands (Fig. A1).
This acts to (i) mask closed seas – which can have environments very different from the open ocean and are typically poorly
simulated by ESMs – and (ii) maximize coincident spatial coverage in the model ensemble and observations.



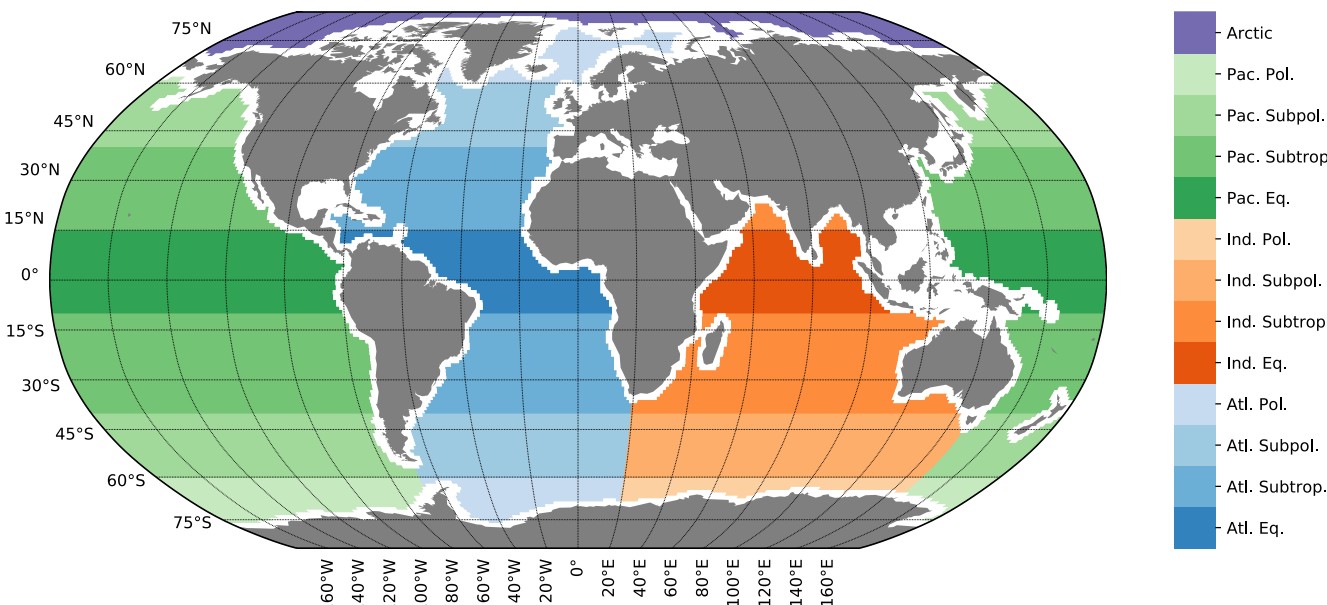

**Fig. A1: Map of the different ocean basins considered in the analysis, the open ocean being the shaded area at 250 km from the coast**
**(small islands were not taken into account). Note that an asymmetry was considered in the tropics for the regional basins between the southern and northern hemispheres due to the difference in the location of the subtropical gyres.**

## 6.2   Appendix B: Salinity normalization

The historical way to normalize Alk and DIC in the whole ocean is to divide the values by the salinity and multiply them using a salinity value of reference, generally 35 g kg$^{-1}$ (e.g., Postma, 1964; Millero et al., 1998). This strategy, that we employed,
gives the Alk and DIC that would have the considered fluid parcel at a salinity of 35 g kg$^{-1}$ (e.g. Sarmiento and Gruber, 2006; Fry et al., 2015). The errors associated with this strategy are essentially confined in the ocean mixed layer where evaporation and precipitation occur (Friis et al., 2003). However, other salinity normalization techniques have been developed ever since (Millero et al., 1998; Robbins, 2001; Friis et al., 2003; Lee et al., 2006; Carter et al., 2014) on the basis of observation-based relationships between surface salinity and Alk.
945        In order to assess the potential biases of the historical method that Sarmiento and Gruber (2006), Krumhardt et al. (2020) and us considered (sAlk), we compare it to the strategy developed by Robbins (2001), Friis et al. (2003) and Carter et al. (2014), also used by Sulpis et al. (2021; sAlk*):

$$sAlk = 35 \cdot \frac{Alk}{S} \tag{B1}$$

$$sAlk^* = Alk - \frac{\overline{Alk^{5\,m}}}{\overline{S^{5\,m}}} \cdot S \tag{B2}$$

where the overbar corresponds to the mean surface value of the observations from GLODAPv2 or the CMIP5/6 ensemble mean. Hence, the method of salinity normalization was found to have negligible impact on vertical profiles of sAlk (Fig. B1b). A slight difference between the two methods is apparent in the surface distribution of normalized Alk, particularly in the Arctic Ocean (Fig. B1c). However, the conventional salinity normalization approach was preferred throughout our analysis as it maintains the order of magnitude of Alk, avoids empirical relationships and is therefore easier to interpret.





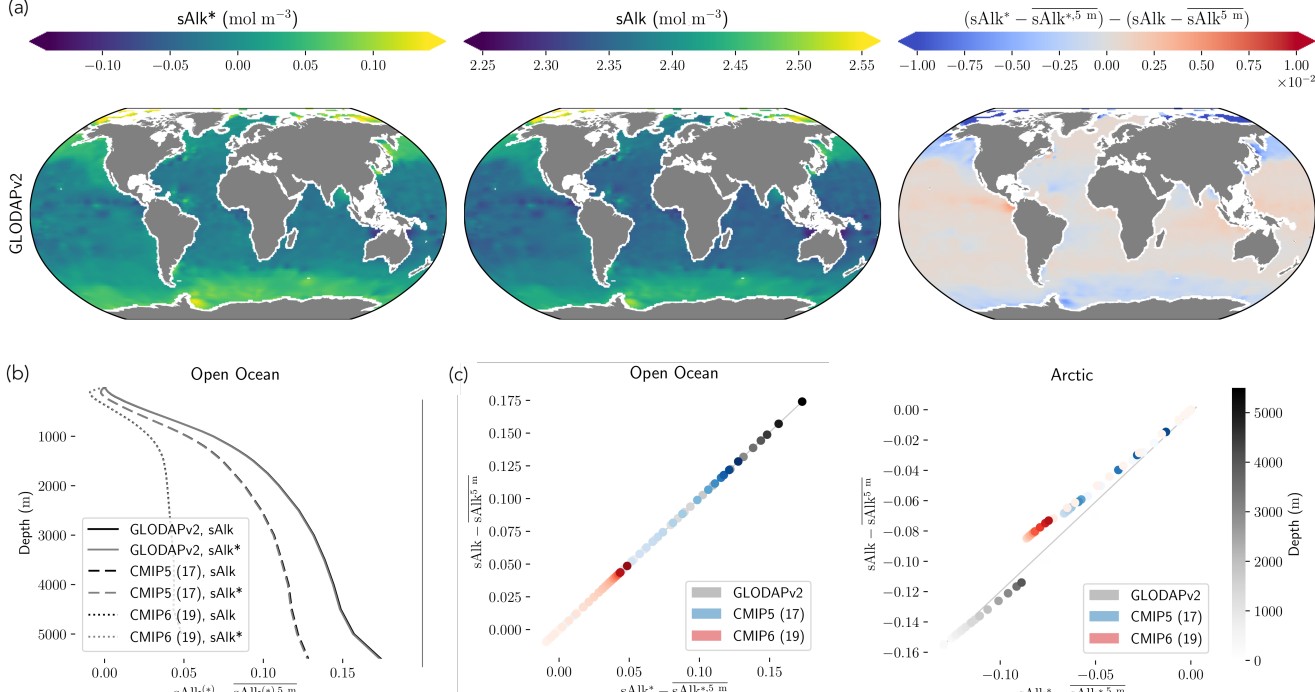

**Fig. B1: Evaluation of two strategies for salinity normalization; sAlk, defined in Sect. 2.4, and sAlk\* expressed in Appendix B 6.2. (a) Maps of the open ocean surface sAlk\*, sAlk and the difference of their values substracted from the surface average. (b) Profiles of sAlk\* and sAlk for the observations as well as the CMIP5 and CMIP6 ensemble means. (c) Scatter plots of the sAlk\* and sAlk values substracted from the surface average in the open ocean and the Arctic for the observations as well as the CMIP5 and CMIP6 vertical profiles of the ensemble means.**

### 6.3 Appendix C: Model equations and parameters

In the following, we present the equations and parameters governing the calcium carbonate cycle in the ocean for each one of the biogeochemical models involved in our CMIP5 and CMIP6 analysis. The marine biogeochemical models are classified following Table 1, and the parameters are displayed in alphabetical order in Tables C2-C18. The formalism used is shared in Table C1. If some processes are not included in the equations, we will generally write 'N/A' in the following, but related information can be found in the Supplementary Table S1. Finally, for the PIC balance equation, a Lagrangian derivative is considered when the PIC is advected (we improperly include its diffusion in this derivative if considered) rather than a partial derivative.









**Fig. C1: Model description and performances regarding the representation of Alk and the carbonate pump. The metrics displayed refer to the biogeochemical modelling schemes (top), both the resolution and the spin-up (middle), as well as some model performances (bottom). For the top and middle parts, details were added compared to Fig. 2 and 10 (see black letters) with for each**
**metric the explanation of the letters shared to the right. Note that for the PIC production metric, the capital letter(s) that is(are) underlined correspond(s) to the element(s) explicitly modelled on which the PIC production relies on. For these metrics, an asterisk is added when specificities can be found in the Supplementary Table S1. In addition, some metrics were added to the bottom in order to assess model performances. For each of them, the shading was normalized between -1 and 1 so that the maximum of the absolute value corresponds to an extremity of the colorbar centered on 0. For each row, the black figure given for the ESM with the greatest**
**bias corresponds to the percentage that this bias represents compared to the observations from GLODAPv2. The lighter the cell is, the better the ESM does with regards to the considered metric.**

**Table C1: Formalism used with regards to the variables and parameters.**

| Variable/Parameter | Unit | Description |
|---|---|---|
| $P_{[\ldots]}$ | $\propto$ molC | Phytoplankton biomass |
| $Z_{[\ldots]}$ | $\propto$ molC | Zooplankton biomass |
| $PIC_{[\ldots]}$ | $\propto$ molC | Particulate inorganic carbon |
| $POC_{[\ldots]}$ | $\propto$ molC | Particulate organic carbon |
| $X^N$ | $\propto$ molN | X shared in N (nitrogen) units rather than C (carbon) units |
| $X^P$ | $\propto$ molP | X shared in P (phosphorus) units rather than C (carbon) units |
| $X^m$ | $\propto$ unit(X) kg$^{-1}$ | Mass concentration of X |
| $X^s$ | $\propto$ unit(X) m$^{-2}$ | Surface concentration of X |
| $X^v$ | $\propto$ unit(X) m$^{-3}$ | Volume concentration of X |
| Bu(X) | unit(X) m$^{-2}$ d$^{-1}$ | Burial of X |
| Di(X) | unit(X) d$^{-1}$ | Dissolution of X |
| Ex(X) | unit(X) m$^{-2}$ d$^{-1}$ | Export of X |
| Pr(X) | unit(X) d$^{-1}$ | Production of sinking calcite |
| Re(X) | unit(X) d$^{-1}$ | Remineralization |
| St(X) | unit(X) m$^{-2}$ d$^{-1}$ | Temporary storage of X |
| PAR | $\propto$ W m$^{-2}$ | Photosynthetically active radiation |
| PILi | $\propto$ molC | Particulate inorganic lithogenic material |
| PISi | $\propto$ molSi | Particulate inorganic silicon |
| $K_{sp,[\ldots]}$ | $\propto$ (mol kg$^{-1}$)$^2$ | Solubility product |
| T | °C | Potential temperature |
| $w_{[\ldots]}$ | m d$^{-1}$ | Sinking speed |
| z | m | Depth, positive downward, and origin at the ocean surface |
| z' | m | Depth (used in integrals), positive downward, and origin at the ocean surface |
| $\Omega_{[\ldots]}$ | | Saturation state |
| f_X | unit(X) | X is considered through a function |

### 6.3.1   CMOC

Reference used: Zahariev et al. (2008).
Marine biogeochemical model of CanESM2 (CMIP5) and CanESM5 (CMIP6).

PIC balance: N/A.

PIC export at 100 m:



$$Ex(PIC_{calc})(100) = r_{PIC_{calc}} \cdot \frac{exp[0.6 \cdot (T - 10)]}{1 + exp[0.6 \cdot (T - 10)]} \cdot r_{C:N} \cdot w_{POC} \cdot POC^{v,N}(n_{z_{eld}}^{above})$$

PIC production: N/A.

PIC sinking speed: N/A.

PIC dissolution:
$$\text{Di}(PIC_{calc}^{v}) = \frac{max(0, \; z - z_{eld})}{z - z_{eld}} \cdot \frac{1}{D_{PIC_{calc}}} \text{Ex}(PIC_{calc})(z_{eld}) \cdot exp\left(\frac{-z + z_{eld}}{D_{PIC_{calc}}}\right)$$

Ballast effect: N/A.

Protection effect: N/A.

PIC burial: N/A.

**Table C2: CMOC parameters.**

| Variable/Parameter | Value | Unit | Description |
|---|---|---|---|
| $D_{PIC_{calc}}$ | 2700 | m | Calcite dissolution length-scale |
| $n_{z_{eld}}^{above}$ | | | First layer above the euphotic layer |
| $r_{C:N}$ | 6.6 | molC (molN)$^{-1}$ | C:N ratio of organic matter |
| $r_{PIC_{calc}}$ | 0.085 | | Production ratio parameter for calcite |
| $w_{POC}$ | 10 | m d$^{-1}$ | POC sinking speed |
| $z_{eld}$ | 100 | m | Bottom euphotic layer depth |

**6.3.2    CanOE**

Reference used: Christian et al. (2022).
Marine biogeochemical model of CanESM5-CanOE (CMIP6).

PIC balance:
$$\frac{DPIC_{calc}^{v}}{Dt} = Pr(PIC_{calc}^{v}) - Di(PIC_{calc}^{v}) - w_{PIC_{calc}} \frac{\partial PIC_{calc}^{v}}{\partial z}$$

PIC export at 100 m:
$$Ex(PIC_{calc})(100) = w_{PIC_{calc}} \cdot PIC_{calc}^{v}(100)$$

PIC production:
$$Pr(PIC_{calc}^{v}) = r_{PIC_{calc}} \cdot [m^{PZ_{small}} \cdot (P_{small}^{v} + Z_{small}^{v}) + \mu^{PZ_{small}} \cdot ((P_{small}^{v})^2 + (Z_{small}^{v})^2)]$$

PIC sinking speed: $w_{PIC_{calc}}$

PIC dissolution:
$$Di(PIC_{calc}^{v}) = \lambda_{PIC_{calc}} \cdot PIC_{calc}^{v}$$



Ballast effect: N/A.

Protection effect: N/A.

PIC burial:

$$Bu(PIC_{calc}) = \frac{max(0, \Omega_{calc} - 1)}{\Omega_{calc} - 1} \cdot w_{PIC_{calc}} \cdot (PIC_{calc}^v)^{seafloor}$$

**Table C3: CanOE parameters.**

| Variable/Parameter | Value | Unit | Description |
|---|---|---|---|
| $m^{PZ_{small}}$ | 0.05 | d$^{-1}$ | Small phytoplankton/zooplankton mortality rate |
| $\mu^{PZ_{small}}$ | 0.06 | (mmolC m$^{-3}$)$^{-1}$ d$^{-1}$ | Quadratic mortality coefficient of small phytoplankton/zooplankton |
| $r_{PIC_{calc}}$ | 0.05 | | Production ratio parameter for calcite |
| $w_{PIC_{calc}}$ | 20 | m d$^{-1}$ | PIC$_{calc}$ sinking speed |
| $\lambda_{PIC_{calc}}$ | 0.0074 | d$^{-1}$ | Calcite dissolution rate |

### 6.3.3    BFM4

Marine biogeochemical model of CMCC-CESM (CMIP5).
No calcium carbonate cycle.

### 6.3.4    BFM5.2

References used: Vichi et al. (2011, 2013, 2020) and Lovato et al. (2022).
Marine biogeochemical model of CMCC-ESM2 (CMIP6).

PIC balance:
$$\frac{D(PIC_{calc}^v)}{Dt} = Pr(PIC_{calc}^v) - Di(PIC_{calc}^v) - w_{PIC_{calc}} \frac{\partial PIC_{calc}^v}{\partial z}$$

PIC export at 100 m:
$$Ex(PIC_{calc})(100) = w_{PIC_{calc}}(100) \cdot PIC_{calc}^v(100)$$

PIC production:
$$Pr(PIC_{calc}^v) = r_{PIC_{calc}}$$
$$\cdot \left[ \eta^{Z_{micro}} \cdot f\_graz^{Z_{micro}} \cdot Z_{micro}^v + \min\left(\frac{r_{C:N}^{P_{nano},ref}}{f\_r_{C:N}^{P_{nano}}}, \frac{r_{C:P}^{P_{nano},ref}}{f\_r_{C:P}^{P_{nano}}}\right) \cdot m^{P_{nano},max} \cdot \frac{K_m'^{P_{nano}}}{f\_nut^{P_{nano}} \cdot K_m'^{P_{nano}}} \right.$$
$$\left. \cdot P_{nano}^v \right]$$

PIC sinking speed: $w_{PIC_{calc}}$

PIC dissolution:



$$Di(PIC_{calc}^v) = \lambda_{PIC_{calc}} \cdot max(0, 1 - \Omega_{calc})^{\alpha_{PIC_{calc}}} \cdot PIC_{calc}^v$$

Ballast effect: N/A.

Protection effect: N/A.

PIC burial:
$$Bu(PIC_{calc}) = w_{PIC_{calc}}^{seafloor} \cdot (PIC_{calc}^v)^{seafloor}$$

**Table C4: BFM5.2 parameters.**

| Variable/Parameter | Value | Unit | Description |
|---|---|---|---|
| f_graz$^{Z_{micro}}$ | f(P$_{nano}$, P$_{diat}$, T) | d$^{-1}$ | Function associated with the microzooplankton grazing rate |
| f_nut$^{P_{nano}}$ | f$(PO_4^{2-}, NO_3^-, NH_4^+)$ | | Function regarding the nutrient stress term for nanophytoplankton |
| f_r$_{C:N}^{P_{nano}}$ | Quota | molC (molN)$^{-1}$ | Quota of C with respect to N |
| f_r$_{C:N}^{P_{nano}}$ | Quota | molC (molP)$^{-1}$ | Quota of C with respect to P |
| K$_m^{'P_{nano}}$ | 0.1 | | Saturation constant associated to nutrient stress |
| m$^{P_{nano},max}$ | 0.05 | d$^{-1}$ | Maximum mortality rate associated to nutrient-stress |
| r$_{C:N}^{P_{nano},ref}$ | $\frac{106}{16}$=6.625 | molC (molN)$^{-1}$ | Reference C:N ratio |
| r$_{C:P}^{P_{nano},ref}$ | 106 | molC (molP)$^{-1}$ | Reference C:P ratio |
| r$_{PIC_{calc}}$ | 0.1 | | Production ratio parameter for calcite |
| w$_{PIC_{calc}}$ | 3 | m d$^{-1}$ | PIC$_{calc}$ sinking speed |
| w$_{PIC_{calc}}^{seafloor}$ | 30 | m d$^{-1}$ | PIC burial velocity |
| α$_{PIC_{calc}}$ | 1 | | Exponent for the dissolution rate of calcite |
| η$^{Z_{micro}}$ | 0.5 | | Fraction associated with grazing inefficiency by microzooplankton |
| λ$_{PIC_{calc}}$ | 10.9 | d$^{-1}$ | Calcite dissolution rate |

### 6.3.5    PISCESv1

Reference used: Aumont (2005).
Marine biogeochemical model of CNRM-ESM1 (CMIP5), IPSL-CM5A-LR (CMIP5), IPSL-CM5A-MR (CMIP5), IPSL-CM5B-LR (CMIP5).

PIC balance:
$$\frac{DPIC_{calc}^v}{Dt} = Pr(PIC_{calc}^v) - Di(PIC_{calc}^v) - w_{PIC_{calc}} \cdot \frac{\partial PIC_{calc}^v}{\partial z}$$

PIC export at 100 m:
$$Ex(PIC_{calc})(100) = w_{PIC_{calc}}(100) \cdot PIC_{calc}(100)$$

PIC production:





$$Pr(PIC_{calc}^v) = \text{PREREQ}$$

$$\cdot \left[ \eta^Z \cdot (f\_graz^{Z_{micro}} \cdot Z_{micro}^v + f\_graz^{Z_{meso}} \cdot Z_{meso}^v) + m^{P_{nano}} \cdot \frac{P_{nano}^v}{K_m^{P_{nano}} + P_{nano}^v} \cdot P_{nano} + \mu^{P_{nano}} \right.$$

$$\left. \cdot (P_{nano}^v)^2 \right]$$

$$\text{PREREQ} = r_{PIC_{calc}} \cdot f\_nut^{P_{nano}} \cdot max\left(0.0001, \frac{T}{2+T}\right) \cdot max\left(1, \frac{P_{nano}^v}{2}\right)$$

PIC sinking speed:

$$w_{PIC_{calc}} = w_{POC_{large}} = w_{POC_{large}}^{min} + (w_{POC_{large}}^{ref} - w_{POC_{large}}^{min}) \cdot \frac{max(0, z - z_{mld})}{z_{ref}}$$

PIC dissolution:

$$Di(PIC_{calc}^v) = \lambda_{PIC_{calc}}^{max} \cdot \frac{max(0, [CO_3^{2-}]_{sat}^{calc} - [CO_3^{2-}])}{K_{PIC_{calc}} + [CO_3^{2-}]_{sat}^{calc} - [CO_3^{2-}]} \cdot PIC_{calc}^v$$

Ballast:

$$\frac{\partial POC_{small}^v}{\partial t} = (1 - 0.5 \cdot R_{PIC_{calc}}) \cdot \left( m^{P_{nano}} \cdot \frac{P_{nano}^v}{K_m^{P_{nano}} + P_{nano}^v} \cdot P_{nano}^v + \mu^{P_{nano}} \cdot (P_{nano}^v)^2 \right) + f\_bal^{POC_{small}}$$

$$\frac{\partial POC_{large}^v}{\partial t} = 0.5 \cdot R_{PIC_{calc}} \cdot \left( m^{P_{nano}} \cdot \frac{P_{nano}^v}{K_m^{P_{nano}} + P_{nano}^v} \cdot P_{nano}^v + \mu^{P_{nano}} \cdot (P_{nano}^v)^2 \right) + f\_bal^{POC_{large}}$$

Protection effect: N/A.

PIC burial: N/A.

**Table C5: PISCESv1 parameters.**

| Variable/Parameter | Value | Unit | Description |
|---|---|---|---|
| $f\_graz^{Z_{meso}}$ | $f(P_{nano}, P_{diat}, POC_{small}, POC_{large}, T)$ | d$^{-1}$ | Function associated with the mesozooplankton grazing rate |
| $f\_graz^{Z_{micro}}$ | $f(P_{nano}, P_{diat}, POC_{small}, T)$ | d$^{-1}$ | Function associated with the microzooplankton grazing rate |
| $f\_nut^{P_{nano}}$ | $f(PO_4^{2-}, Fe^{2+}, NO_3^-, NH_4^+)$ | | Function regarding the nutrient limitation term for nanophytoplankton |
| $f\_bal^{POC_{small/large}}$ | f(miscellaneous) | | Function completing the $POC_{small/large}$ balance |
| $K_m^{P_{nano}}$ | 0.1 | µmolC L$^{-1}$ | Half-saturation constant for nanophytoplankton mortality |
| $K_{PIC_{calc}}$ | Not found | µmolC L$^{-1}$ | Half-saturation constant for calcite dissolution |
| $m^{P_{nano}}$ | 0.01 | d$^{-1}$ | Nanophytoplankton mortality rate |
| $r_{PIC_{calc}}$ | 0.4 | | Production ratio parameter for calcite |
| $w_{POC_{large}}^{min}$ | 50 | m d$^{-1}$ | Minimum POC sinking speed |
| $w_{POC_{large}}^{ref}$ | 200 | m d$^{-1}$ | Value of reference for POC sinking speed |



| $z_{mld}$ | Depth at which the potential density is higher by 0.01 compared to the surface value | m | Bottom mixed layer depth |
|---|---|---|---|
| $z_{ref}$ | 2000 | m | Reference depth for the POC sinking speed |
| $\eta^Z$ | 0.5·0.3=0.15 | | Fraction associated with grazing inefficiency by zooplankton |
| $\lambda_{PIC_{calc}}^{max}$ | 0.03 | d$^{-1}$ | Maximum calcite dissolution rate |
| $\mu^{P_{nano}}$ | 0.01 | (µmolC L$^{-1}$)$^{-1}$ d$^{-1}$ | Quadratic mortality of nanophytoplankton |

### 6.3.6 PISCESv2 and PISCESv2-gas

Reference used: Aumont et al. (2015).
Marine biogeochemical model of CNRM-ESM2-1 (CMIP6) and IPSL-CM6A-LR (CMIP6).

PIC balance:
$$\frac{DPIC_{calc}^v}{Dt} = Pr(PIC_{calc}^v) - Di(PIC_{calc}^v) - w_{PIC_{calc}} \cdot \frac{\partial PIC_{calc}^v}{\partial z}$$

PIC export at 100 m:
$$Ex(PIC_{calc})(100) = w_{PIC_{calc}}(100) \cdot PIC_{calc}^v(100)$$

Production :
$$Pr(PIC_{calc}^v) = \text{PREREQ}$$
$$\cdot \left[ \eta^{Z_{micro}} \cdot f\_graz^{Z_{micro}} \cdot Z_{micro}^v + \eta^{Z_{meso}} \cdot f\_graz^{Z_{meso}} \cdot Z_{meso}^v + m^{P_{nano}} \cdot \frac{P_{nano}}{K_m^{P_{nano}} + P_{nano}} \cdot P_{nano}^v + sh \right.$$

$$\left. \cdot \mu'^{P_{nano}} \cdot (P_{nano}^v)^2 \right]$$
$$\text{PREREQ} = r_{PIC_{calc}} \cdot f\_nut^{P_{nano}} \cdot \frac{T}{0.1 + T} \cdot max\left(1, \frac{P_{nano}^v}{2}\right) \cdot \frac{max(0, PAR - 1)}{4 + PAR} \cdot \frac{30}{30 + PAR} \cdot \left[1 + exp\left(\frac{-(T - 10)^2}{25}\right)\right]$$
$$\cdot min\left(1, \frac{50}{z_{mld}}\right)$$

PIC sinking speed:
$$w_{PIC_{calc}} = w_{POC_{large}} = w_{POC_{large}}^{min} + (w_{POC_{large}}^{ref} - w_{POC_{large}}^{min}) \cdot \frac{max(0, z - max(z_{eld}, z_{mld}))}{z_{ref}}$$

PIC dissolution:
$$Di(PIC_{calc}^v) = \lambda_{PIC_{calc}}^{max} \cdot \left(max\left(0, 1 - \frac{[CO_3^{2-}]}{[CO_3^{2-}]_{sat}^{calc}}\right)\right)^{\alpha_{PIC_{calc}}} \cdot PIC_{calc}^v$$

Ballast effect:





$$\frac{\partial POC^v_{small}}{\partial t} = (1 - 0.5 \cdot R_{PIC_{calc}}) \cdot \left( m^{P_{nano}} \cdot \frac{P^v_{nano}}{K^{P_{nano}}_m + P^v_{nano}} \cdot P_{nano} + sh \cdot \mu\,'^{P_{nano}} \cdot (P^v_{nano})^2 \right) + f\_bal^{POC_{small}}$$

$$\frac{\partial POC^v_{large}}{\partial t} = 0.5 \cdot R_{PIC_{calc}} \cdot \left( m^{P_{nano}} \cdot \frac{P^v_{nano}}{K^{P_{nano}}_m + P^v_{nano}} \cdot P_{nano} + sh \cdot \mu\,'^{P_{nano}} \cdot (P^v_{nano})^2 \right) + f\_bal^{POC_{large}}$$

Protection effect: N/A.

PIC burial:

$$Bu(PIC_{calc}) = 0.6 \cdot min\left(1, 1.3 \cdot \frac{\Omega_{calc} - 0.8}{\Omega_{calc} - 0.6}\right) \cdot Ex(PIC_{calc})^{seafloor}$$

A bug has been noted and addressed with regards to the burial of PIC in recent release of the model.

**Table C6: PISCESv2 and PISCESv2-gas parameters.**

| Variable/Parameter | Value | Unit | Description |
|---|---|---|---|
| $f\_bal^{POC_{small/large}}$ | f(miscellaneous) | | Function completing the $POC_{small/large}$ balance |
| $f\_graz^{Z_{meso}}$ | $f(P_{nano}, P_{diat}, POC_{small}, POC_{large}, T)$ | $d^{-1}$ | Function associated with the mesozooplankton grazing rate |
| $f\_graz^{Z_{micro}}$ | $f(P_{nano}, P_{diat}, POC_{small}, T)$ | $d^{-1}$ | Function associated with the microzooplankton grazing rate |
| $f\_nut^{P_{nano}}$ | $f(PO_4^{2-}, Fe^{2+}, NO_3^-, NH_4^+)$ | | Function regarding the nutrient limitation term for nanophytoplankton |
| $K^{P_{nano}}_m$ | 0.2 | µmolC L$^{-1}$ | Half-saturation constant for nanophytoplankton mortality |
| $m^{P_{nano}}$ | 0.01 | $d^{-1}$ | Nanophytoplankton mortality rate |
| $r_{PIC_{calc}}$ | 0.3 | | Production ratio parameter for calcite |
| sh | 1 in the mixed layer 0.01 below | $s^{-1}$ | Shear rate |
| $w^{min}_{POC_{large}}$ | 30 | m d$^{-1}$ | Minimum POC sinking speed |
| $w^{ref}_{POC_{large}}$ | 200 | m d$^{-1}$ | Value of reference for POC sinking speed |
| $z_{eld}$ | Depth at which PAR is equal to 0.01·PAR$^{surf}$ | m | Bottom euphotic layer depth |
| $z_{mld}$ | Depth at which the potential density is higher by 0.01 compared to the surface value | m | Bottom mixed layer depth |
| $z_{ref}$ | 5000 | m | Reference depth for the POC sinking speed |
| $\alpha_{PIC_{calc}}$ | 1 | | Exponent for the dissolution rate of calcite |
| $\eta^{Z_{micro}}$ | 0.5 | | Fraction associated with grazing inefficiency by microzooplankton |



| | | | |
|---|---|---|---|
| $\eta^{Z_{meso}}$ | 0.75 | | Fraction associated with grazing inefficiency by mesozooplankton |
| $\lambda^{max}_{PIC_{calc}}$ | 0.197 | $d^{-1}$ | Maximum calcite dissolution rate |
| $\mu^{'P_{nano}}$ | 0.01 | $(\mu molC\ L^{-1})^{-1}\ s\ d^{-1}$ | Quadratic mortality coefficient of nanophytoplankton |


### 6.3.7 WOMBAT

References used: Oke et al. (2013), Law et al. (2017) and Ziehn et al. (2020).
Marine biogeochemical model of ACCESS-ESM1-5 (CMIP6).

PIC balance:
$$\frac{DPIC^v}{Dt} = Pr(PIC^v) - Di(PIC^v) - w_{PIC} \cdot \frac{\partial PIC^v}{\partial z}$$

PIC export at 100 m:
$$Ex(PIC_{calc})(100) = w_{PIC} \cdot PIC^v(100)$$

PIC production:
$$Pr(PIC^v) = r_{PIC} \cdot r_{C:N} \cdot [\eta^Z \cdot PREREQ \cdot Z^{v,N} + \mu^Z \cdot (Z^{v,N})^2 + \mu^P \cdot (P^{v,N})^2]$$
$$PREREQ = g^{Z,max} \cdot \frac{\epsilon^Z \cdot (P^{v,N})^2}{g^{Z,max} + \epsilon^Z \cdot (P^{v,N})^2}$$

PIC sinking speed: $w_{PIC}$

PIC dissolution:
$$Pr(PIC^v) = \lambda_{PIC} \cdot PIC^v$$

Ballast effect: N/A.

Protection effect: N/A.

PIC burial: N/A.

**Table C7: WOMBAT parameters.**

| Variable/Parameter | Value | Unit | Description |
|---|---|---|---|
| $g^{Z,max}$ | 1.575 | $d^{-1}$ | Maximum grazing rate |
| $r_{C:N}$ | $\frac{106}{16}=6.625$ | $molC\ (molN)^{-1}$ | C:N ratio |
| $r_{PIC}$ | 0.062 | | Production ratio parameter for PIC |
| $w_{PIC}$ | 6 | $m\ d^{-1}$ | PIC sinking speed |
| $\epsilon^Z$ | 1.6 | $d^{-1}\ (mmolN\ m^{-3})^{-2}$ | Prey capture coefficient |
| $\eta^Z$ | 0.075 | | Fraction associated with grazing inefficiency by zooplankton |
| $\lambda_{PIC}$ | 0.001714 | $d^{-1}$ | Calcium carbonate dissolution rate |
| $\mu^Z$ | 0.34 | $d^{-1}\ (mmolN\ m^{-3})^{-2}$ | Quadratic mortality of zooplankton |





| | | | |
|---|---|---|---|
| $\mu^P$ | 0.25 | d$^{-1}$ (mmolN m$^{-3}$)$^{-2}$ | Quadratic mortality of phytoplankton |

### 6.3.8    OECO1

References used: Yoshikawa et al. (2008) and Watanabe et al. (2011).
Marine biogeochemical model of MIROC-ESM (CMIP5) and MIROC-ESM-CHEM (CMIP5).

PIC balance:

$$\frac{D(PIC^v)}{Dt} = \begin{cases} Pr(PIC^v) - Di(PIC^v) - w_{PIC} \cdot \frac{\partial PIC^v}{\partial z} & \text{for } z<200 \text{ m} \\ Pr(PIC^v) - Di(PIC^v) + w_{PIC} \cdot PIC^v(200) \cdot 0.858 \cdot \left(\frac{200^{0.858}}{z^{1.858}}\right) & \text{for } z\geq 200 \text{ m} \end{cases}$$

PIC Production:

$$Pr(PIC^v) = r_{PIC} \cdot r_{C:N} \cdot \left[\frac{g^{Z,max} \cdot \epsilon^Z \cdot (P^{v,N})^2}{g^{Z,max} + \epsilon^Z \cdot (P^{v,N})^2} \cdot Z^{v,N} + (f\_growth^P - m^P) \cdot P^{v,N}\right]$$

PIC sinking speed: $w_{PIC}$ in the top 200 m and N/A below 200 m.

PIC dissolution:
$$Di(PIC^v) = \lambda_{PIC} \cdot (1.066)^T \cdot PIC^v$$

Ballast effect: N/A.

Protection effect: N/A.

PIC burial: N/A.

**Table C8: OECO1 parameters.**

| Variable/Parameter | Value | Unit | Description |
|---|---|---|---|
| f_growth$^P$ | f(T, PAR, NO$_3^-$) | | Function associated with the growth rate of phytoplankton |
| g$^{Z,max}$ | 2.0 | d$^{-1}$ | Maximum grazing rate |
| m$^P$ | 0.05 | d$^{-1}$ | Phytoplankton mortality rate |
| r$_{C:N}$ | $\frac{106}{16}$ | mmolC (mmolN)$^{-1}$ | C:N ratio |
| r$_{PIC}$ | 0.005 | | Production ratio parameter for PIC |
| w$_{PIC}$ | 5 | m d$^{-1}$ | PIC sinking speed |
| $\epsilon^Z$ | 1.0 | (mmolN m$^{-3}$)$^{-2}$ d$^{-1}$ | Prey capture coefficient |
| $\lambda_{PIC}$ | 0.05 | d$^{-1}$ | Calcium carbonate dissolution rate |

### 6.3.9    OECO2

References used: Schmittner et al. (2008) and Hajima et al. (2020).
Marine biogeochemical model of MIROC-ES2L (CMIP6).



PIC balance:

$$\frac{D(PIC^v)}{Dt} = Pr(PIC^v) - Di(PIC^v)$$

PIC export at 100 m:

$$Ex(PIC)(100) = \int_{z'} Pr(PIC^v)dz \cdot exp(\frac{-100}{D_{PIC}}) = \int_{z'} Pr(PIC^v)dz' - \int_0^{100} Di(PIC^v)dz'$$

PIC production:

$$Pr(PIC^v) = r_{PIC} \cdot r_{C:N} \cdot \left[ \eta^Z \cdot \frac{g^{Z,max} \cdot \epsilon^Z \cdot (P_{non-diaz}^{v,N})^2}{g^{Z,max} + \epsilon^Z \cdot (P_{non-diaz}^{v,N})^2} \cdot Z^{v,N} + \mu^{P_{non-diaz}} \cdot (P_{non-diaz}^{v,N})^2 + \mu^Z \cdot (Z^{v,N})^2 \right]$$

PIC sinking speed: N/A.

PIC dissolution:

$$Di(PIC^v)(z) = \frac{\partial(Ex(PIC))}{\partial z}(z) = \frac{1}{D_{PIC}} \int_{z'} Pr(PIC^v)dz' \cdot exp(\frac{-z}{D_{PIC}})$$

Ballast effect: N/A.

Protection effect: N/A.

PIC burial: N/A.

**Table C9: OECO2 parameters.**

| Variable/Parameter | Value | Unit | Description |
|---|---|---|---|
| $D_{PIC}$ | 6500 | m | PIC dissolution length-scale |
| $g^{Z,max}$ | 2.0 | d$^{-1}$ | Maximum grazing rate |
| $r_{C:N}$ | $\frac{106}{16}$ | mmolC (mmolN)$^{-1}$ | C:N ratio |
| $r_{PIC}$ | 0.005 | | Production ratio parameter for PIC |
| $\epsilon^Z$ | 1.0 | (mmolN m$^{-3}$)$^{-2}$ d$^{-1}$ | Prey capture coefficient |
| $\eta^Z$ | 0.25 | | Fraction associated with grazing inefficiency by zooplankton |
| $\mu^{P_{non-diaz}}$ | 0.05 | (mmolN m$^{-3}$)$^{-1}$ d$^{-1}$ | Quadratic mortality of non-diazotrophic phytoplankton |
| $\mu^Z$ | 0.2 | (mmolN m$^{-3}$)$^{-1}$ d$^{-1}$ | Quadratic mortality of zooplankton |

### 6.3.10 diat-HadOCC

Reference used: Totterdell (2019).
Marine biogeochemical model of HadGEM2-CC (CMIP5) and HadGEM2-ES (CMIP5).

PIC balance:

$$\frac{\partial PIC^v(n)}{\partial t} = Pr(PIC^v)(n) - Di(PIC^v)(n)$$

PIC export at 100 m:





$Ex(PIC)(100) = \sum_{m \text{ for } z \leq 100 \text{ m}} Pr(PIC^v)(m) \cdot h(m)$

PIC production:
$Pr(PIC^v)(n) = r_{PIC} \cdot r_{C:N} \cdot Pr(P^{v,N})(n)$

PIC sinking speed: N/A.

PIC dissolution:

$Di(PIC^v(n)) = \begin{cases} \dfrac{\sum_m Pr(PIC^v)(m) \cdot h(m)}{z_{sfd} - z_{lyd}} & \text{layer n below the lysocline} \\ 0 & \text{layer n above the lysocline} \end{cases}$

When $z_{sfd} \leq z_{lyd}$ all the $PIC$ is dissolved in the bottom level.

Ballast effect: N/A.

Protection effect: N/A.

PIC burial: N/A.

**Table C10: diat-HadOCC parameters.**

| Variable/Parameter | Value | Unit | Description |
|---|---|---|---|
| h(n) | | m | Height of the layer |
| n | | | Layer considered |
| $r_{C:N}$ | $\dfrac{106}{16}$=6.625 | mmolC (mmolN)$^{-1}$ | C:N ratio of miscellaneous phytoplankton |
| $r_{PIC}$ | 0.0195 | | Production ratio parameter |
| $z_{lyd}$ | 2113 | m | Prescribed lysocline depth |
| $z_{sfd}$ | Bathymetry | m | Seafloor depth |

### 6.3.11    MEDUSA-2.1

Reference used: Yool et al. (2013).
Marine biogeochemical model of UKESM1-0-LL (CMIP6).


PIC balance:
$Ex(PIC_{calc})(n+1) = Ex(PIC_{calc})(n) - Di(Ex(PIC_{calc}))(n) \cdot h(n) + Pr(PIC^v_{calc})(n) \cdot h(n)$

PIC export at 100 m: No specific calculation.

PIC production:



$Pr(PIC_{calc}^v) = r_{PIC_{calc}}$

$$\cdot \left[ \left( r_{C:N}^{P_{diat}} \cdot \eta^{P_{diat}} \cdot m^{P_{diat},max} \cdot \frac{P_{diat}}{K_m^{P_{diat}} + P_{diat}} \cdot P_{diat}^v \right) \right.$$
$$\left. + \left( r_{C:N}^{Z_{meso}} \cdot \eta^{Z_{meso}} \cdot m^{Z_{meso},max} \cdot \frac{Z_{meso}}{K_m^{Z_{meso}} + Z_{meso}} \cdot Z_{meso}^v \right) \right] \cdot [max(0, \Omega_{calcite} - 1)]^{\beta_{PIC_{calc}}}$$

PIC sinking speed: N/A.

PIC dissolution:

$Di(Ex(PIC_{calc}(n))) = \begin{cases} Ex(PIC_{calc}(n)) \cdot \dfrac{1 - \exp\left(-\dfrac{h(n)}{D_{PIC_{calc}}}\right)}{h(n)} & \text{layer n below the lysocline} \\ 0 \text{ layer n above the lysocline} \end{cases}$

Ballast effect: N/A.

Protection effect:

$Ex(POC_{fast})(n+1) = Ex(POC_{fast}^{protected})(n) + (Ex(POC_{fast})(n) - Ex(POC_{fast}^{protected})(n)) \cdot exp\left(-\dfrac{h(n)}{D_{excess}}\right)$

$Ex(POC_{fast}^{protected})(n) = Ex(POC_{fast}^{protected,PIC_{calc}})(n) + Ex(POC_{fast}^{protected,PISi})(n)$

$Ex(POC_{fast}^{protected,PIC_{calc}})(n) = f_{PIC_{calc}} \cdot \dfrac{M_{CaCO_3}}{M_{org}} \cdot Ex(PIC_{calc})(n)$

PIC temporary storage at the seafloor (no burial):
$St(PIC_{calc}) = Ex(PIC_{calc})^{seafloor} - \lambda_{PIC_{calc}}^{seafloor} \cdot (PIC_{calc}^s)^{seafloor}$

**Table C11: MEDUSA-2.1 parameters.**

| Variable/Parameter | Value | Unit | Description |
|---|---|---|---|
| $D_{excess}$ | 188 | m | Excess POC$_{fast}$ remineralization length-scale |
| $D_{PIC_{calc}}$ | 3500 | m | Calcite dissolution length-scale |
| $f_{PIC_{calc}}$ | 0.070 | | Calcite protection ratio |
| $h(n)$ | | m | Height of the layer n |
| $K_m^{P_{diat}}$ | 0.5 | mmolN m$^{-3}$ | Half-saturation constant for diatom phytoplankton mortality |
| $K_m^{Z_{meso}}$ | 0.75 | mmolN m$^{-3}$ | Half-saturation constant for diatom mesozooplankton mortality |
| $m^{P_{diat},max}$ | 0.1 | d$^{-1}$ | Maximum diatom phytoplankton mortality rate |
| $m^{Z_{meso},max}$ | 0.2 | d$^{-1}$ | Maximum mesozooplankton mortality rate |
| $M_{org}$ | 12.011 | g molC$^{-1}$ | Organic equivalent C molar mass |
| $M_{CaCO_3}$ | 100.086 | g molC$^{-1}$ | CaCO$_3$ equivalent C molar mass |
| n | | | Layer considered |
| $r_{C:N}^{P_{diat}}$ | $\dfrac{106}{16}$=6.625 | mmolC (mmolN)$^{-1}$ | C:N ratio of diatom phytoplankton |
| $r_{C:N}^{Z_{meso}}$ | 5.625 | mmolC (mmolN)$^{-1}$ | C:N ratio of mesozooplankton |
| $r_{PIC_{calc}}$ | 0.026 | | Production ratio parameter for calcite |
| $\beta_{PIC_{calc}}$ | 0.81 | | Exponent for the calcification rate of calcite |





| | | | |
|---|---|---|---|
| $\eta^{P_{diat}}$ | 0.33 | | Part of the fast detritus of diatom phyoplankton losses associated with $PIC_{calc}$ |
| $\eta^{Z_{meso}}$ | 1.00 | | Part of the fast detritus of mesozooplankton losses associated with $PIC_{calc}$ |
| $\lambda^{seafloor}_{PIC_{calc}}$ | 0.01 | d$^{-1}$ | Benthic calcite dissolution rate |

### 6.3.12 HAMOCC5.2 and HAMOCC6

Reference used: Ilyina et al. (2013).
Marine biogeochemical model of MPI-ESM-LR (CMIP5), MPI-ESM-MR (CMIP5), MPI-ESM1-2-LR (CMIP6) and MPI-ESM1-2-HR (CMIP6).

PIC balance:
$$\frac{\partial PIC^v_{calc}}{\partial t} = Pr(PIC^v_{calc}) - Di(PIC^v_{calc}) - w_{PIC_{calc}} \cdot \frac{\partial PIC^v_{calc}}{\partial z}$$

PIC export at 100 m:
$$Ex(PIC_{calc}) = w_{PIC_{calc}} \cdot PIC^v_{calc}(100)$$

PIC production:
$$Pr(PIC^v_{calc}) = r_{PIC_{calc}} \cdot r_{C:P} \cdot \frac{K_{Si(OH)_4}}{K_{Si(OH)_4} + [Si(OH)_4]}$$
$$\cdot \left[ m^P \cdot (P^{v,P} - P^{v,P,min}) + \eta^Z \cdot g^Z \cdot \frac{P^{v,P} - P^{v,P,min}}{K_Z + P^{v,P,min}} \cdot Z^v + \eta'^Z \cdot \mu^Z \cdot (Z^{v,P} - Z^{v,P,min})^2 \right]$$

PIC sinking speed: $w_{PIC_{calc}}$

PIC dissolution:
$$Di(PIC^v_{calc}) = \lambda_{PIC_{calc}} \cdot max\left(0, \frac{K_{sp,calc}}{[Ca^{2+}]} - [CO_3^{2-}]\right) \cdot PIC^v_{calc}$$

Ballast effect: N/A.

Protection effect: N/A.

PIC burial:
$$Bu(PIC_{calc}) = f\_bur^{PIC_{calc}}$$

**Table C12: HAMOCC5.2 and HAMOCC6 parameters.**

| Variable/Parameter | Value | Unit | Description |
|---|---|---|---|
| f_bur$^{PIC_{calc}}$ | f$\left(PIC^{bottom}_{calc}, miscellaneous\right)$ | | Function defining the $PIC_{calc}$ burial |
| g$^{Z,max}$ | 1 | d$^{-1}$ | Maximum grazing rate |
| K$_{Si(OH)_4}$ | 1.0·10$^{-6}$ | kmolSi m$^{-3}$ | Half-saturation constant for Si(OH)$_4$ uptake |
| K$_Z$ | 4·10$^{-8}$ | kmolP m$^{-3}$ | Half-saturation constant for grazing |
| m$^P$ | 0.008 | d$^{-1}$ | Phytoplankton mortality rate |



| Variable/Parameter | Value | Unit | Description |
|---|---|---|---|
| $P^{v,P,min}$ | $1\cdot10^{-11}$ | kmolP m$^{-3}$ | Minimum concentration of phytoplankton |
| $r_{C:P}$ | 122 | molC (molP)$^{-1}$ | C:P ratio |
| $r_{PIC_{calc}}$ | $\dfrac{20}{122}$ | | Production ratio parameter for calcite |
| $w_{PIC_{calc}}$ | 30 | m d$^{-1}$ | Calcite sinking speed |
| $Z^{v,P,min}$ | $1\cdot10^{-11}$ | kmolP m$^{-3}$ | Minimum concentration of zooplankton |
| $\lambda_{PIC_{calc}}$ | 0.0075 | d$^{-1}$ | Calcite dissolution rate |
| $\eta^Z$ | 0.2 | | Fraction associated with grazing inefficiency by herbivore zooplankton |
| $\eta^{'Z}$ | 0.05 | | Fraction associated with grazing inefficiency by carnivore zooplankton |
| $\mu^Z$ | $3\cdot10^6$ | (kmolP m$^{-3}$)$^{-1}$ d$^{-1}$ | Quadratic mortality of zooplankton |
| $[Ca^{2+}]$ | 10.3 | mmol kg$^{-1}$ | Fixed calcium ion concentration |


### 6.3.13 NPZD-MRI

References used: Schmittner et al. (2008) and Tsujino et al. (2010, 2017).
Marine biogeochemical model of MRI-ESM1 (CMIP5) and MRI-ESM2-0 (CMIP6).

PIC balance:

$$\frac{\partial PIC^v}{\partial t} = Pr(PIC^v) - Di(PIC^v)$$

PIC export at 100 m:

$$Ex(PIC_{calc})(100) = \int_{z'} Pr(PIC^v_{calc})dz' \cdot exp(\frac{-100}{D_{PIC_{calc}}}) = \int_{z'} Pr(PIC^v_{calc})dz' - \int_0^{100} Di(PIC^v_{calc})dz'$$

PIC production:

$$Pr(PIC^v_{calc}) = r_{PIC_{calc}} \cdot r_{C:N} \cdot \left[ \eta^Z \cdot \frac{g^{Z,max} \cdot \epsilon^Z \cdot (P^{v,N})^2}{g^{Z,max} + \epsilon^Z \cdot (P^{v,N})^2} \cdot Z^{v,N} + \mu^P \cdot (P^{v,N})^2 + \mu^Z \cdot (Z^{v,N})^2 \right]$$

PIC sinking speed: N/A.

PIC dissolution:

$$Di(PIC^v_{calc})(z) = -\frac{\partial(Ex(PIC^v_{calc}))}{\partial z}(z) = \frac{1}{D_{PIC_{calc}}} \cdot \int_{z'} Pr(PIC^v_{calc})dz' \cdot exp(\frac{-z}{D_{PIC_{calc}}})$$

Ballast effect: N/A.

Protection effect: N/A.

PIC burial: N/A.

**Table C13: NPZD-MRI parameters.**

| Variable/Parameter | Value | Unit | Description |
|---|---|---|---|



| | | | |
|---|---|---|---|
| $D_{PIC_{calc}}$ | 3500 | m | Calcite dissolution length-scale |
| $g^{Z,max}$ | 1.575 | d$^{-1}$ | Maximum grazing rate |
| $r_{C:N}$ | $\dfrac{112}{16}=7$ | molC (molN)$^{-1}$ | C:N ratio |
| $r_{PIC_{calc}}$ | 0.03 | | Production ratio parameter for calcite |
| $\epsilon^Z$ | 1.6 | (mmolN m$^{-3}$)$^{-2}$ d$^{-1}$ | Prey capture coefficient |
| $\eta^Z$ | 0.075 | | Fraction associated with grazing inefficiency by zooplankton |
| $\mu^P$ | 50 | (molN m$^{-3}$)$^{-1}$ d$^{-1}$ | Quadratic mortality of phytoplankton |
| $\mu^Z$ | 340 | (molN m$^{-3}$)$^{-1}$ d$^{-1}$ | Quadratic mortality of zooplankton |

### 6.3.14  BEC

References used: Armstrong et al. (2001), Moore et al. (2001, 2004) and a kindly shared document written by Ivan Lima in 2016.
Marine biogeochemical model of CESM1-BGC (CMIP5).

PIC balance:

$$\frac{\partial PIC_{calc}^v}{\partial t} = Pr(PIC_{calc}^v) - Di(PIC_{calc}^v)$$

PIC export at 100 m:

$$Ex(PIC_{calc})(100) = \int_{z'} Pr(PIC_{calc}^v)dz' - \int_0^{100} Di(PIC_{calc}^v)dz'$$

PIC production:

$$Pr(PIC_{calc}^v) = \frac{P_{calc}^v}{P_{small}^v} \cdot (\eta^Z \cdot g^{P_{small},max} \cdot 2 \cdot exp\left(\frac{T-30}{10}\right) \cdot \frac{(P_{small}^v)^2}{(P_{small}^v)^2 + K_g'^{P_{small}}} \cdot Z^v + m^{P_{small}} \cdot P_{small}^v$$
$$+ min[b^{P_{small},max} \cdot P_{small}^v, \mu^{P_{small}} \cdot (P_{small}^v)^2])$$

Calcite pool in the phytoplankton group:

$$Pr(PIC_{calc}^v) = \begin{cases} min\left(PREREQ \cdot \dfrac{P_{small}^v}{3.0}, 0.40 \cdot Pr(P_{small}^v)\right) & \text{if } P_{small}^v > 3.0 \\ PREREQ & \text{else} \end{cases}$$

$$PREREQ = r_{PIC_{calc}}' \cdot Pr(P_{small}^v) \cdot f\_nut^{P_{small}} \cdot \left[1 + \left(\frac{max(T+2, 0)}{4} - 1\right) \cdot \frac{max(2-T, 1)}{2-T}\right]$$

PIC sinking speed: N/A.

PIC dissolution:

$$Di(PIC_{calc})(z) = -\frac{\partial(Ex(PIC_{calc}^v))}{\partial z}(z)$$
$$= \int_{z'} Pr(PIC_{calc}^v)dz' \cdot \left[(1-\psi) \cdot \frac{1}{D_{PIC_{calc}}} \cdot exp(\frac{-z}{D_{PIC_{calc}}}) + \psi \cdot \frac{1}{D_{POC_{hard}}} \cdot exp(\frac{-z}{D_{POC_{hard}}})\right]$$

Ballast effect: N/A.

Protection effect:
$$Ex(POC)(z) = Ex(POC^{free})(z) + Ex(POC^{protected})(z)$$





$$Ex(POC^{protected})(z) = r''_{PIC_{calc}} \cdot \frac{M_{PIC_{calc}}}{M_{POC}} \cdot Ex(POC^{protected,PIC_{calc}})(z) + f\_exp^{PISi,dust}$$

$$Ex(POC^{protected,PIC_{calc}})(z) = Ex(POC_{soft}^{protected,PIC_{calc}})(z) + Ex(POC_{hard}^{protected,PIC_{calc}})(z)$$

$$Ex(POC_{soft}^{protected,PIC_{calc}})(z) = (1-\psi) \cdot \int_{z'} Pr(PIC_{calc}^v)dz' \cdot exp(\frac{-z}{D_{PIC_{calc}}})$$

$$Ex(POC_{hard}^{protected,PIC_{calc}})(z) = \psi \cdot \int_{z'} Pr(PIC_{calc}^v)dz' \cdot exp(\frac{-z}{D_{POC_{hard}}})$$

PIC burial: N/A.

**Table C14: BEC paramaters.**

| Variable/Parameter | Value | Unit | Description |
|---|---|---|---|
| $b^{P_{small},\,max}$ | 0.75 | $d^{-1}$ | Maximum aggregation rate for small phytoplankton |
| $D_{PIC_{calc}}$ | 800 | m | Calcite dissolution length-scale for "soft" particles |
| $D_{POC_{hard}}$ | 40,000 | m | Dissolution length-scale for "hard" particles |
| $f\_exp^{PISi,dust}$ | f(PISi, dust) | | Remaining export function |
| $f\_nut^{P_{small}}$ | $f(PO_4^{2-}, Fe^{2+}, NO_3^-, NH_4^+)$ | | Nutrient limitation term for small phytoplankton |
| $g^{P_{small},max}$ | 2.5 | $d^{-1}$ | Maximum zooplankton growth rate when grazing phytoplankton |
| $K_g^{'P_{small}}$ | 1.0 | $(mmolC\ m^{-3})^2$ | Grazing coefficient for small phytoplankton |
| $m^{P_{small}}$ | 0.15 | $d^{-1}$ | Small phytoplankton mortality rate |
| $M_{POC}$ | 12.01 | $g\ molC^{-1}$ | Organic equivalent C molar mass |
| $M_{PIC_{calc}}$ | 100.09 | $g\ molC^{-1}$ | Calcite equivalent C molar mass |
| $r'_{PIC_{calc}}$ | 0.042 | | Baseline fraction of small phytoplankton production as calcite production |
| $r''_{PIC_{calc}}$ | 0.05 | | Associated POC/ $PIC_{calc}$ mass ratio for particulate matter |
| $\eta^Z$ | 0.67 | | Fraction associated with grazing inefficiency by zooplankton |
| $\mu^{P_{small}}$ | 0.0035 | $(mmolC\ m^{-3})^{-1}\ d^{-1}$ | Quadratic mortality of small phytoplankton |
| $\psi$ | 0.55 | | Fraction of $PIC_{calc}$ that is routed to "hard" particles |


### 6.3.15 MARBL

Reference used : Long et al. (2021).
Marine biogeochemical model of CESM2 (CMIP6), CESM2-WACCM-FV2 (CMIP6), CESM2-FV2 (CMIP6) and CESM2-WACCM (CMIP6).


PIC balance:
$$\frac{\partial PIC_{calc}^v}{\partial t} = Pr(PIC_{calc}^v) - Di(PIC_{calc}^v)$$

PIC export at 100 m:



$$Ex(PIC_{calc})(100) = \int_{z'} Pr(PIC_{calc}^v)dz' - \int_0^{100} Di(PIC_{calc}^v)dz'$$

PIC production:

$$Pr(PIC_{calc}^v) = \frac{P_{calc}^v}{P_{small}^v}$$
$$\cdot \left( \eta^Z \cdot g^{P_{small},max} \cdot 2 \cdot exp\left(\frac{T-30}{10}\right) \cdot \frac{(P_{small}^v)}{(P_{small}^v) + K_g^{P_{small}}} \cdot Z^v + m^{P_{small}} \cdot P_{small}^v \right.$$
$$\left. + min[b^{P_{small},max} \cdot P_{small}^v, \mu^{P_{small}} \cdot (P_{small}^v)^2] \right)$$

Calcite pool in the phytoplankton group:

$$Pr(PIC_{calc}^v) = \begin{cases} min\left( PREREQ \cdot \frac{P_{small}^v}{2.5}, 0.40 \cdot Pr(P_{small}^v) \right) & \text{if } P_{small}^v > 2.5 \\ PREREQ & \text{else} \end{cases}$$

$$PREREQ = r'_{PIC_{calc}} \cdot Pr(P_{small}^v) \cdot (f\_nut^{P_{small}})^2 \cdot \left[ 1 + \left( \frac{max(T+2, 0)}{6} - 1 \right) \cdot \frac{max(4-T, 1)}{4-T} \right]$$

PIC sinking speed: N/A.

PIC dissolution:

$$Di(PIC_{calc}^v) = \frac{1}{f\_dist^{PIC_{calc}}} \int_{z'} Pr(PIC_{calc}^v)dz' \cdot exp\left(\frac{-z}{f\_dist^{PIC_{calc}}}\right)$$

$$f\_dist^{PIC_{calc}} = \begin{cases} D_{PIC_{calc}}^{100} & \text{if } z \leq 100 \\ D_{PIC_{calc}}^{1000} & \text{if } z \geq 1000 \\ D_{PIC_{calc}}^{prev} + \frac{D_{PIC_{calc}}^{next} - D_{PIC_{calc}}^{prev}}{z^{next} - z^{prev}} \cdot (z - z^{prev}) & \text{otherwise} \end{cases}$$

Ballast effect: N/A.

Protection effect:

$$Ex(POC)(z) = Ex(POC^{free})(z) + Ex(POC^{protected})(z)$$
$$Ex(POC^{protected})(z) = r''_{PIC_{calc}} \cdot \frac{M_{PIC_{calc}}}{M_{POC}} \cdot Ex(POC^{protected,PIC_{calc}})(z) + f\_exp^{PISi,dust}$$
$$Ex(POC^{protected,PIC_{calc}})(z) = Ex(POC_{soft}^{protected,PIC_{calc}})(z) + Ex(POC_{hard}^{protected,PIC_{calc}})(z)$$
$$Ex(POC_{soft}^{protected,PIC_{calc}})(z) = (1-\psi) \cdot \int_{z'} Pr(PIC_{calc}^v)dz' \cdot exp\left(\frac{-z}{D_{PIC_{calc}}}\right)$$
$$Ex(POC_{hard}^{protected,PIC_{calc}})(z) = \psi \cdot \int_{z'} Pr(PIC_{calc}^v)dz' \cdot exp\left(\frac{-z}{D_{POC_{hard}}}\right)$$

PIC burial:

$$Bu(PIC_{calc}) = \frac{max(0, \Omega_{calc} - \Omega_{calc}^{crit})}{\Omega_{calc} - \Omega_{calc}^{crit}} \cdot Ex(PIC_{calc})^{seafloor}$$




**Table C15: MARBL parameters.**

| Variable/Parameter | Value | Unit | Description |
|---|---|---|---|
| $b^{P_{small},\,max}$ | 0.75 | $d^{-1}$ | Maximum aggregation rate for small phytoplankton |
| $D^{100}_{PIC_{calc}}$ | 500 | m | Calcite dissolution length-scale at 100 m |
| $D^{250}_{PIC_{calc}}$ | 1800 | m | Calcite dissolution length-scale at 250 m |
| $D^{500}_{PIC_{calc}}$ | 2350 | m | Calcite dissolution length-scale at 500 m |
| $D^{1000}_{PIC_{calc}}$ | 2400 | m | Calcite dissolution length-scale at 1000 m |
| $D_{POC_{hard}}$ | 40,000 | m | Dissolution length-scale for "hard" particles |
| $f\_exp^{PISi,dust}$ | f(PISi, dust) | | Remaining export function |
| $f\_nut^{P_{small}}$ | $f(PO_4^{2-}, Fe^{2+}, NO_3^-, NH_4^+)$ | | Nutrient limitation term for small phytoplankton |
| $g^{P_{small},max}$ | 3.3 | $d^{-1}$ | Maximum zooplankton growth rate when grazing phytoplankton |
| $K_g^{P_{small}}$ | 1.2 | $mmolC\ m^{-3}$ | Grazing coefficient for small phytoplankton |
| $m^{P_{small}}$ | 0.1 | $d^{-1}$ | Small phytoplankton mortality rate |
| $M_{POC}$ | 12.01 | $g\ molC^{-1}$ | Organic equivalent C molar mass |
| $M_{PIC_{calc}}$ | 100.09 | $g\ molC^{-1}$ | Calcite equivalent C molar mass |
| $r'_{PIC_{calc}}$ | 0.07 | | Baseline fraction of small phytoplankton production as calcite production |
| $r''_{PIC_{calc}}$ | 0.01 | | Associated POC/ $PIC_{calc}$ mass ratio for particulate matter |
| $\eta^Z$ | 0.67 | | Fraction associated with grazing inefficiency by zooplankton |
| $\mu^{P_{small}}$ | 0.01 | $(mmolC\ m^{-3})^{-1}\ d^{-1}$ | Quadratic mortality coefficient of nanophytoplankton |
| $\Omega_{calc}^{crit}$ | 0.89 | | Saturation state threshold for calcite burial |
| $\psi$ | 0.02 | | Fraction of $PIC_{calc}$ that is routed to "hard" particles |

### 6.3.16 HAMOCC5.1 and iHAMOCC

References used: Assmann et al. (2010), Tjiputra et al. (2010), Tjiputra et al. (2013), Schwinger et al. (2016) and Tjiputra et al. (2020).
Marine biogeochemical models of NorESM1-ME (CMIP5) and NorESM2-LM (CMIP6).

PIC balance:
$$\frac{DPIC^v_{calc}}{Dt} = Pr(PIC^v_{calc}) - Di(PIC^v_{calc}) - w_{PIC_{calc}} \cdot \frac{\partial PIC^v_{calc}}{\partial z}$$

PIC export at 100 m:
$$Ex(PIC_{calc})(100) = w_{PIC_{calc}} \cdot PIC^v_{calc}(100)$$

PIC production:





$$Pr(PIC_{calc}^v) = r_{PIC_{calc}} \cdot r_{C:P} \cdot \frac{K_{Si(OH)_4}}{K_{Si(OH)_4} + [Si(OH)_4]}$$

$$\cdot \left[ \eta'^Z \cdot \mu^Z \cdot max(0, (Z^{v,P} - 2 \cdot Z^{v,P,min})^2) + m^P \cdot max(0, P^{v,P} - 2 \cdot P^{v,P,min}) + \eta^Z \cdot g^{Z,max} \right.$$

$$\left. \cdot \frac{P^{v,P} - P^{v,P,min}}{K_g^Z + P^{v,P}} \cdot Z^{v,P} \right]$$

PIC sinking speed: $w_{PIC_{calc}}$

PIC dissolution:

$$Di(PIC_{calc}^v) = min\left[ \frac{PREREQ}{\Delta t}, \lambda_{PIC_{calc}} \cdot PIC_{calc}^v \right]$$

$$PREREQ = max\left( 0, \frac{[CO_3^{2-}]}{\Omega_{calc}} - [CO_3^{2-}] \right)$$

A dimension issue was noted.

Ballast effect: N/A.

Protection effect: N/A.

PIC burial:

$$Bu(PIC_{calc}) = f\_bur^{PIC_{calc}}$$

**Table C16: HAMOCC5.1 and iHAMOCC parameters.**

| Variable/Parameter | Value | Unit | Description |
|---|---|---|---|
| $f\_bur^{PIC_{calc}}$ | $f(PIC_{calc}^{bottom}, \Omega_{calc}, \text{sediment dynamics})$ | molC m$^{-2}$ d$^{-1}$ | Function defining the $PIC_{calc}$ burial |
| $g^{Z,max}$ | 1.0 (CMIP5), 1.2 (CMIP6) | | Maximum grazing rate |
| $K_g^Z$ | $4 \cdot 10^{-8}$ (CMIP5), $8 \cdot 10^{-8}$ (CMIP6) | kmolP m$^{-3}$ | Half-saturation constant for grazing |
| $K_{Si(OH)_4}$ | $1.5 \cdot 10^{-6} (CMIP5), 5.0 \cdot 10^{-6} (CMIP6)$ | kmolSi m$^{-3}$ | Half-saturation constant for Si(OH)$_4$ uptake |
| $m^P$ | 0.008 | d$^{-1}$ | Mortality rate of phytoplankton |
| $P^{v,P,min}$ | $1 \cdot 10^{-11}$ | kmolP m$^{-3}$ | Minimum concentration of phytoplankton |
| $r_{C:P}$ | 122 | molC (molP)$^{-1}$ | C:P ratio |
| $r_{PIC_{calc}}$ | $\frac{35}{122}$ (CMIP5), $\frac{33}{122}$ (CMIP6) | | Production ratio parameter for calcite |
| $w_{PIC_{calc}}$ | 30 | m d$^{-1}$ | $PIC_{calc}$ sinking speed |
| $Z^{v,P,min}$ | $1 \cdot 10^{-10}$ | kmolP m$^{-3}$ | Minimum concentration of zooplankton |
| $\Delta t$ | $\frac{1}{24}$ | d | Time step |
| $\eta^Z$ | 0.2 (CMIP5), 0.15 (CMIP6) | | Fraction associated with grazing inefficiency by herbivore zooplankton |
| $\eta'^Z$ | 0.05 | | Fraction associated with grazing inefficiency by carnivore zooplankton |





| | | | |
|---|---|---|---|
| $\lambda_{PIC_{calc}}$ | 0.05x24=1.2 | d$^{-1}$ | Calcite dissolution rate |
| $\mu^Z$ | $5 \cdot 10^6$(CMIP5), $3 \cdot 10^6$(CMIP6) | (kmolP m$^{-3}$)$^{-1}$ d$^{-1}$ | Quadratic mortality of zooplankton |

### 6.3.17   TOPAZ2

Reference used: Dunne et al. (2013).
Marine biogeochemical model of GFDL-ESM2G (CMIP5) and GFDL-ESM2M (CMIP5).


PIC balance:
$$\frac{DPIC^m_{calc/arag}}{Dt} = Pr(PIC^m_{calc/arag}) - Di(PIC^m_{calc/arag}) - w \cdot \frac{\partial PIC^m_{calc/arag}}{\partial z}$$

PIC export at 100 m:
$$Ex(PIC_{calc/arag})(100) = \rho \cdot w \cdot PIC^m_{calc/arag}(100)$$

PIC production:

$$Pr(PIC^{m,N}_{calc}) = r_{PIC_{calc}} \cdot r_{C:N} \cdot exp(-0.0539 \cdot T) \cdot min\left(\frac{1}{\Delta t}, g_0 \cdot exp(0.063 \cdot T) \cdot \frac{(P^{m,N}_{small})^2}{P^g_{small} \cdot (P^{m,N}_{small} + P'^g_{small})}\right) \cdot P^{m,N}_{small}$$
$$\cdot min[\Omega^{max}, max(0, \Omega_{calc} - 1)]$$
$$Pr(PIC^{m,N}_{arag}) = r_{PIC_{arag}} \cdot r_{C:N} \cdot min\left(\frac{1}{\Delta t}, f\_graz^{P_{large}}\right) \cdot P^{m,N}_{large} \cdot min[\Omega^{max}, max(0, \Omega_{arag} - 1)]$$

PIC sinking speed: w

PIC dissolution:
$$Di(PIC^{m,N}_{calc/arag}) = \lambda_{PIC_{calc/arag}} \cdot max(0, 1 - \Omega_{calc/arag}) \cdot PIC^{m,N}_{calc/arag}$$

Ballast effect: N/A.

Protection effect:
$$Re(POC^{m,N}) = f\_rem^{POC} \cdot max\left[0, POC^{m,N} - \left(p \cdot \left(PIC^m_{calc} + PIC^m_{arag}\right) + f\_prot^{POC}\right)\right]$$

PIC burial:
$$Bu(PIC_{calc}) = f\_bur^{PIC_{calc}}$$
$$Bu(PIC_{arag}) = 0$$

**Table C17: TOPAZ2 parameters.**

| Variable/Parameter | Value | Unit | Description |
|---|---|---|---|
| $f\_graz^{P_{large}}$ | $f(P^{m,N}_{large}, P^{m,N}_{small}, T)$ | d$^{-1}$ | Function associated with the grazing of large phytoplankton |
| $f\_bur^{PIC_{calc}}$ | $f(\Omega_{calc}, PIC^{bottom}_{calc}, PILi^{bottom})$ | molC $m^{-2}$ d$^{-1}$ | Function defining the $PIC_{calc}$ burial |
| $f\_prot^{POC}$ | $f(PISi, PILi)$ | molC $kg^{-1}$ | Function completing the protection of POC |
| $f\_rem^{POC}$ | $f(O_2, NO_3^-)$ | d$^{-1}$ | Function completing the remineralization parametrization of POC |
| $g_0$ | 0.19 | d$^{-1}$ | Grazing rate at 0°C |





| | | | |
|---|---|---|---|
| p | $\dfrac{79}{12} \simeq 5.8$ | molC (molC)$^{-1}$ | Protection from remineralization |
| $P_{small}^{g}$ | $\dfrac{1.9 \cdot 16}{106} \cdot 10^{-6} \simeq 0.29 \cdot 10^{-6}$ | molN kg$^{-1}$ | Pivot phytoplankton concentration for grazing allometry |
| $P_{small}^{'g}$ | $1.0 \cdot 10^{-10}$ | molN kg$^{-1}$ | Minimum phytoplankton concentration threshold for grazing |
| $r_{C:N}$ | $\dfrac{106}{16} = 6.625$ | molC (molN)$^{-1}$ | C:N ratio |
| $r_{PIC_{arag}}$ | 0.01 | | Production ratio parameter for aragonite |
| $r_{PIC_{calc}}$ | 0.005 | | Production ratio parameter for calcite |
| w | 100 | m d$^{-1}$ | Sinking speed of particulates |
| Δt | $\dfrac{2}{24} = \dfrac{1}{12}$ | $d$ | Time step |
| $\lambda_{PIC_{arag}}$ | $0.13 = \dfrac{100}{760}$ | d$^{-1}$ | Aragonite dissolution rate |
| $\lambda_{PIC_{calc}}$ | $0.074 = \dfrac{100}{1343}$ | d$^{-1}$ | Calcite dissolution rate |
| ρ | 1,035 | kg m$^{-3}$ | Density constant |
| $\Omega_{max}$ | 10 | | Maximum saturation state |

### 6.3.18  BLINGv2

Reference used: Dunne et al. (2020).
Marine biogeochemical model of GFDL-CM4 (CMIP6).

PIC balance:
$$Ex(PIC_{calc})(n) = min(1, n-1) \cdot Ex(PIC_{calc})(n-1) + [Pr(PIC_{calc}^{m})(n) - Di(PIC_{calc}^{m})(n)] \cdot \rho \cdot h(n)$$

PIC export at 100 m: No specific calculation.

PIC production:
$$Pr(PIC_{calc}^{m}(n)) = r_{PIC_{calc}} \cdot r_{C:P} \cdot (1 - f\_frac^{Plarge}) \cdot Pr(P^{m,P}(n)) \cdot exp(-0.0539 \cdot T) \cdot min[\Omega_{calc}^{max}, max(0, \Omega_{calc} - 1)]$$

PIC sinking speed: N/A.

PIC dissolution:
$$Di(PIC_{calc}^{m}(n)) = \left[ \sum_{k=1}^{n-1} \left( Pr(PIC_{calc}^{m}(k)) - Di(PIC_{calc}^{m}(k)) \right) + Pr(PIC_{calc}^{m}(n)) \right] \cdot \left[ 1 - \frac{1}{1 + \dfrac{h(n)}{D_{PIC_{calc}}} \cdot min(1, 1 - \Omega_{calc})} \right]$$

Ballast effect: N/A.

Protection effect:
$$Ex(POC)(n+1) = Ex(POC)(n) + f\_bal^{POC}(n)$$

PIC burial:



$$Bu(PIC_{calc}) = f\_bur^{PIC_{calc}}$$
$$Bu(PIC_{arag}) = 0$$

**Table C18: BLINGv2 parameters.**

| Variable/Parameter | Value | Unit | Description |
|---|---|---|---|
| $D_{PIC_{calc}}$ | 1343 | m | Calcite dissolution length-scale |
| $f\_frac^{P_{large}}$ | $f(PAR, T, Fe, PO_4^{2-})$ | | Fraction of phytoplankton that is large |
| $f\_bal^{POC}$ | $f(z, O_2, PIC_{calc}^m, PILi, POC^{m,P})$ | molC $m^{-2}$ d$^{-1}$ | Function completing the POC balance |
| $f\_bur^{PIC_{calc}}$ | $f(\Omega_{calc}, PIC_{calc}^{bottom}, PILi, POC^{m,P})$ | molC $m^{-2}$ d$^{-1}$ | Function defining the $PIC_{calc}$ burial |
| $h(n)$ | | m | Height of the layer n |
| n | | | Layer considered (starting at 1) |
| $Pr(P^m(n))$ | $f(PAR, T, Fe, PO_4^{2-})$ | | Production of phytoplankton |
| $r_{C:P}$ | 106 | molC (molP)$^{-1}$ | C:P ratio |
| $r_{PIC_{calc}}$ | $\dfrac{0.53}{106}=0.005$ | | Production ratio parameter for calcite |
| $\rho$ | 1,035 | kg m$^{-3}$ | Density constant |
| $\Omega_{calc}^{max}$ | 10 | | Maximum saturation state |


### 6.3.19    COBALTv2

References used: Stock et al. (2014, 2020).
Marine biogeochemical model of GFDL-ESM4 (CMIP6).

PIC balance:
$$\frac{DPIC_{calc/arag}^m}{Dt} = Pr(PIC_{calc/arag}^m) - Di(PIC_{calc/arag}^m) - w \cdot \frac{\partial PIC_{calc/arag}^m}{\partial z}$$

PIC export at 100 m:
$$Ex(PIC_{calc/arag})(100) = \rho \cdot w \cdot PIC_{calc/arag}^m(100)$$

PIC production:
$$Pr(PIC_{calc}^m) = r_{PIC_{calc}} \cdot r_{C:N} \cdot min[\Omega^{max}, max(0, \Omega_{calc} - 1)]$$
$$\cdot \left[\eta^{Z_{medium}} \cdot f\_cons'^{Z_{small}} \cdot Z_{small}^{m,N} + \eta^{Z_{small}} \cdot f\_graz'^{P_{small}} \cdot P_{small}^{m,N} + \eta^{Z_{large}} \cdot f\_graz'^{P_{large}} \cdot P_{large}^{m,N}\right.$$
$$\left. + f\_agg^{P_{small}} \cdot P_{small}^{m,N} + f\_agg^{P_{large}} \cdot P_{large}^{m,N}\right]$$
$$Pr(PIC_{arag}^m) = r_{PIC_{arag}} \cdot r_{C:N} \cdot min[\Omega^{max}, max(0, \Omega_{arag} - 1)]$$
$$\cdot \left[\eta^{Z_{large}} \cdot f\_cons'^{Z_{medium}} \cdot Z_{medium}^m + \eta^{hp} \cdot f\_hpcons'^{Z_{medium}} \cdot Z_{medium}^{m,N} + \eta^{hp} \cdot f\_hpcons'^{Z_{large}}\right.$$
$$\left. \cdot Z_{large}^{m,N}\right]$$

PIC sinking speed: w

PIC dissolution:
$$Di(PIC_{calc}^m) = \lambda_{PIC_{calc/arag}} \cdot max(0, 1 - \Omega_{calc/arag}) \cdot PIC_{calc/arag}^m$$

Ballast effect: N/A.





Protection effect:
$$Re(POC^{m,N}) = f\_rem^{POC} \cdot max\left[0,\ POC^m - \left(p \cdot \left(PIC_{calc}^m + PIC_{arag}^m\right) + f\_prot^{POC}\right)\right]$$

PIC burial:
$$Bu(PIC_{calc}) = f\_bur^{PIC_{calc}}$$
$$Bu(PIC_{arag}) = 0$$

**Table C19: COBALTv2 parameters.**

| Variable/Parameter | Value | Unit | Description |
|---|---|---|---|
| $f\_agg^{P_{small/large}}$ | $f(P_{small/large}, Pr(P_{small/large}))$ | $d^{-1}$ | Aggregation function for small and large phytoplankton |
| $f\_bur^{PIC_{calc}}$ | $f\left(\Omega_{calc}, PIC_{calc}^{bottom}, PILi^{bottom}\right)$ | molC $m^{-2}$ $d^{-1}$ | Function defining the $PIC_{calc}$ burial |
| $f\_cons'^{Z_{medium}}$ | $f\left(Z_{large}, Z_{medium}, P_{large}, T\right)$ | $d^{-1}$ | Consumption rate of medium zooplankton by large zooplankton |
| $f\_cons'^{Z_{small}}$ | $f\left(Z_{medium}, Z_{small}, P_{small}, T\right)$ | $d^{-1}$ | Consumption rate of small zooplankton by medium zooplankton |
| $f\_hpcons^{Z_{medium/large}}$ | $f\left(Z_{medium}, Z_{large}, T\right)$ | $d^{-1}$ | Consumption rate of medium/large zooplankton by higher predators (e.g., fish) |
| $f\_graz'^{P_{large}}$ | $f(Z_{medium}, Z_{large}, P_{large}, T)$ | $d^{-1}$ | Consumption rate of large phytoplankton by medium and large zooplankton |
| $f\_graz'^{P_{small}}$ | $f(Z_{small}, P_{small}, Bacteria, T)$ | $d^{-1}$ | Consumption rate of small phytoplankton by small zooplankton |
| $f\_prot^{POC}$ | $f(PISi, PILi)$ | molC $kg^{-1}$ | Function completing the protection of POC |
| $f\_rem^{POC}$ | $f(O_2, T, z)$ | $d^{-1}$ | Function completing the remineralization parametrization of POC |
| $r_{C:N}$ | $\frac{106}{16}$=6.625 | molC $(molN)^{-1}$ | C:N ratio |
| $r_{PIC_{arag}}$ | 0.030 | | Production ratio parameter for aragonite |
| $r_{PIC_{calc}}$ | 0.013 | | Production ratio parameter for calcite |
| w | 100 | m $d^{-1}$ | Sinking speed of particulates |
| $\eta^{hp}$ | 0.35 | | Fraction associated with grazing inefficiency by higher predators |
| $\eta^{Z_{large}}$ | 0.30 | | Fraction associated with grazing inefficiency by large zooplankton |
| $\eta^{Z_{medium}}$ | 0.20 | | Fraction associated with grazing inefficiency by medium zooplankton |
| $\eta^{Z_{small}}$ | 0.10 | | Fraction associated with grazing inefficiency by small zooplankton |
| $\lambda_{PIC_{arag}}$ | 0.13=$\frac{100}{760}$ | $d^{-1}$ | Aragonite dissolution rate |
| $\lambda_{PIC_{calc}}$ | 0.074=$\frac{100}{1343}$ | $d^{-1}$ | Calcite dissolution rate |
| $\rho$ | 1,035 | kg $m^{-3}$ | Density constant |
| $\Omega^{max}$ | 10 | | Maximum saturation state |





## 6.4    Appendix D: Results and Discussion

We share here additional figures to offer a more detailed view on what is addressed in the main text, or to supplement it.


**Fig. D1: Alk surface distribution. ESM intercomparison of the open ocean surface Alk as simulated by ocean biogeochemical models involved in CMIP5 (first 3 columns) and CMIP6 (last 3 columns) compared to the observations from GLODAPv2. For each CMIP, the first row gives the average, the standard deviation and the difference with the observations – which are shown on the first panel along with the error given in the GLODAPv2 gridded product and the difference between CMIP6 and CMIP5 ensemble means on the third panel.**






**Fig. D2: Distribution of the calcite export at 100 m.** ESM intercomparison of the calcite export at 100 m as simulated by ocean biogeochemical models involved in CMIP5 (first 3 columns) and CMIP6 (last 3 columns). For each CMIP, the first row gives the average and the standard deviation and the difference between CMIP6 and CMIP5 ensemble means is also displayed in the third panel of the first row.



CMIP6-CMIP5

$\text{Ex}(\text{PIC})^{100\,\text{m}}_{\text{arag}}$ (mol m$^{-2}$ s$^{-1}$)

$\sigma\big(\text{Ex}(\text{PIC})^{100\,\text{m}}_{\text{arag}}\big)$ (mol m$^{-2}$ s$^{-1}$)

$\Delta\big(\text{Ex}(\text{PIC})^{100\,\text{m}}_{\text{arag}}\big)$ (mol m$^{-2}$ s$^{-1}$)

CMIP5 (2, mean)     CMIP5 (2, std)     CMIP6 (2, mean)     CMIP6 (2, std)

CESM1-BGC (CMIP5)     CMCC-CESM (CMIP5)     CNRM-ESM1 (CMIP5)     ACCESS-ESM1-5 (CMIP6)     CESM2 (CMIP6)     CESM2-FV2 (CMIP6)

CanESM2 (CMIP5)     GFDL-ESM2G (CMIP5)     GFDL-ESM2M (CMIP5)     CESM2-WACCM (CMIP6)     CESM2-WACCM-FV2 (CMIP6)     CMCC-ESM2 (CMIP6)

HadGEM2-CC (CMIP5)     HadGEM2-ES (CMIP5)     IPSL-CM5A-LR (CMIP5)     CNRM-ESM2-1 (CMIP6)     CanESM5 (CMIP6)     CanESM5-CanOE (CMIP6)

IPSL-CM5A-MR (CMIP5)     IPSL-CM5B-LR (CMIP5)     MIROC-ESM (CMIP5)     GFDL-CM4 (CMIP6)     GFDL-ESM4 (CMIP6)     UKESM1-0-LL (CMIP6)

MIROC-ESM-CHEM (CMIP5)     MPI-ESM-LR (CMIP5)     MPI-ESM-MR (CMIP5)     IPSL-CM6A-LR (CMIP6)     MIROC-ES2L (CMIP6)     MPI-ESM-1-2-HAM (CMIP6)

MRI-ESM1 (CMIP5)     NorESM1-ME (CMIP5)     MPI-ESM1-2-HR (CMIP6)     MPI-ESM1-2-LR (CMIP6)     MRI-ESM2-0 (CMIP6)

NorESM2-LM (CMIP6)

**Fig. D3: As Fig. D2 but for the aragonite export at 100 m.**





420    **Fig. D4: As Fig. D2 but for the POC export.**





**Fig. D5: Complements about the carbonate system. (a) Open ocean zonal averages of the surface sDIC (left panel) and the surface carbonate ions (right panel) as a complement of Fig. 4a. (b) Atlantic-Pacific zonal average of the difference between the CMIP6 and CMIP5 ensemble means of the carbonate ion concentration. The black lines refer to the calcite saturation horizon depth and the white lines to the aragonite ones for the observations from GLODAPv2 as well as both CMIP5 and CMIP6 ensemble means.**

1425













**Fig. D7: Complement on the evaluation of the drift in both CMIP5 and CMIP6 ensembles (continuation of Fig. D6). Drift evaluation for the open ocean of the upper ocean values for salinity and temperature, as well as the integrated POC and PIC export at 100 m for the global ocean, distinguishing calcite and aragonite. On each panel, (i) the first bar refers to the observations or estimates with a standard deviation for the PIC export ([1]assessment at 300m); (ii) the color refers to the drift, given per century; (iii) the bar height refers to the mean over the first 20 years of the Historical period in the piControl (~1850-1870), and the extremity of the black bar refers to the mean over the last 20 years of the RCP/ SSP period in the piControl (~2080-2100). A model is written with an asterisk at the end, if its values were not shown due to too important drifts compared to the others, but for MRI-ESM1 (CMIP5) for which the piControl data were not found.**

## 6.5    Appendix E: GLODAPv2 observations

We discuss here the observations from GLODAPv2, since they seem to have a slight offset either in surface DIC or surface Alk compared to both CMIP5 and CMIP6 ensembles (see Fig. 12a). We have investigated whether the way we averaged the data around 2002 – as the observations from GLODAPv2 are normalized in 2002 – could drive a DIC offset, but it does not actually explain it. Indeed, we report that to center the data in 2002 regarding the ocean uptake of anthropogenic carbon, we should have in fact averaged from 1992 to between 2010 and 2011. Averaging between 1992-2012 induces a slight excess of DIC due to the non-linear increase in carbon uptake over this period. Although this excess in DIC is globally confined in the ocean surface layer, it does not impact our analysis with an increase of less than 0.001 mol m$^{-3}$ for all the ESMs at the surface ocean, which is negligible. A bias in the assessment of the Alk components that are diagnosticated to compute the $CO_2$ system, especially the borate one, might drive a $pCO_2$ offset resulting in a change in the y-intercept in Fig. 12a (see Eq. (15)) of a few mmol m$^{-3}$ (Orr and Epitalon, 2015).

Similarly, we assessed the possible bias of using GLODAPv2 nutrients rather than the 2009 update of the WOA product (Boyer et al., 2018) which was available at the time of CMIP5 simulations. We report only a small difference between the observational products, essentially confined to deep waters, with a decrease of 3.8 % in the magnitude of the nitrate vertical profile at 5000 m (see right panel in Fig. 11b). Estimating the observed soft tissue pump from the WOA product would therefore slightly reduce the bias between observations and the CMIP ESMs with a globally negligible effect on our analysis of the vertical biases of sAlk and sDIC compared to the observations (see Fig. 9). The 10 % larger volume considered in WOA compared to GLODAPv2 (Lauvset et al., 2016) might partly explain the difference between the two data products at depth for nitrate and phosphate.

All that remains is to note the following biases in the observations and keep in mind their possible consequences: (i) much more observations are shared in the surface layer of the ocean than at depth, and in particular the density of observations is divided by about 8 between the surface and the deep ocean; (ii) there are much more observations in both hemispheres during summer than winter; and (iii) the spatial resolution of the gridded product is inequal with an excellent coverage of the North Atlantic Ocean and relatively bad coverage of the Southern Ocean (Olsen et al., 2020). In particular, this results in some important local common differences between the ESMs and GLODAPv2 (e.g., at high latitudes where the ocean is seasonally covered by sea ice, and in the Weddell Sea especially).

***Data availability***: We share additional figures (https://doi.org/10.5281/zenodo.7144330) to offer an ESM intercomparison (profiles, sections and maps) of the main three-dimensional variables processed in this study (Alk, sAlk, DIC, sDIC, NO3, PO4, CO3, T, S), both for CMIP5 and CMIP6, and in comparison with GLODAPv2 observations. All the ESMs data, both for CMIP5 and CMIP6, were available on at least one of the Earth System Grid Federation (ESGF) nodes, except for CNRM-ESM1 (CMIP5; https://climatedata.cnrm-game-meteo.fr/cnrm/CMIP5/output/CNRM-CERFACS/CNRM-ESM1/) as well as 'dissic' and 'talk' for the piControl of CanESM2 (CMIP5; shared by James R. Christian) which were not available.

***Author contributions (CRediT)***: This work is in the framework of the OMIP-BGC group, which contributed collectively to this study, through the organization and execution of the CMIP exercises, as well as in the sharing of both simulation outputs and model parameterizations. Alban Planchat: Conceptualization, Investigation, Methodology, Formal analysis, Visualization, Writing – original draft preparation – and Project administration. Laurent Bopp and Lester Kwiatkowski: Supervision, Funding acquisition, Methodology, Resources, Conceptualization and Writing – original draft preparation. Olivier Torres: Software. All coauthors: Resources and Writing – review & editing.



*Competing interests*: The authors declare that they have no conflict of interest.

*Disclaimer*: This article reflects only the authors' views; the funding agencies and their executive agencies are not responsible for any use that may be made of the information that the article contains.

*Fundings*: Alban Planchat, Laurent Bopp, Lester Kwiatkowski and Olivier Torres thank the ENS-Chanel research chair. Laurent Bopp acknowledges the European Union's Horizon 2020 research and innovation program under grant agreement No. 820989 (project COMFORT, Our common future ocean in the Earth system-quantifying coupled cycles of carbon, oxygen, and nutrients for determining and achieving safe operating spaces with respect to tipping points). Momme Butenschön and Tomas Lovato acknowledge funding from the Italian Ministry of Education, Universities and Research via the JPI Ocean and Climate program CE2COAST. Roland Séférian and Laurent Bopp acknowledge the European Union's Horizon 2020 research and innovation program under grant agreement No. 101003536 (ESM2025 – Earth System Models for the Future). The ACCESS CMIP6 submission work was jointly funded through CSIRO and the Australian Government's National Environmental Science Program (NESP). Michio Watanabe and Akitomo Yamamoto were supported by the Integrated Research Program for Advancing Climate Models (TOUGOU) Grant Number JPMXD0717935715 from the Ministry of Education, Culture, Sports, Science and Technology (MEXT), Japan. Hiroyuki Tsujino's contribution is partly supported by the Environment Research and Technology Development Fund (JPMEERF21S20810) of the Environmental Restoration and Conservation Agency of Japan. Kristen M. Krumhardt thanks the National Science Foundation grant (OCE-1735846). Jörg Schwinger and Jerry Tjiputra acknowledge the Research Council of Norway funded projects INES (270061) and COLUMBIA (275268). They also acknowledge funding from the European Union's Horizon 2020 research and innovation programme under grant agreement number 869357 (project OceanNETs), and from the Research Council of Norway through the projects INES (270061) and COLUMBIA (275268).

*Acknowledgements*: We are very pleased to have been able to work in synergy with all the marine biogeochemistry modelling groups. We acknowledge the World Climate Research Programme's Working Group on Coupled Modelling, which is responsible for the CMIP exercises. For CMIP, the US Department of Energy's Program for Climate Model Diagnosis and Intercomparison provided coordinating support and led the development of software infrastructure in partnership with the Global Organisation for Earth System Science Portals. This study benefited from the ESPRI (*Ensemble de Services Pour la Recherche à l'IPSL*) computing and data center (https://mesocentre.ipsl.fr, last access: July 2022), which is supported by CNRS, Sorbonne Université, École Polytechnique and CNES and through national and international grants. We also thank the administrative and technical staff at Ecole Normale Supérieure/PSL. Hiroyuki Tsujino would like to thank Hideyuki Nakano and Shogo Urakawa for their kind support in carrying out the experiments at MRI. John P. Dunne and Charles Stock thank Jessica Luo for leading the internal NOAA review of the manuscript.

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
