# Peer review of "The representation of alkalinity and the carbonate pump from CMIP5 to CMIP6 ESMs and implications for the ocean carbon cycle"

_EGUsphere, 2022_

## Author Comment (AC2)

**Reviews**

We would like to thank the two reviewers of our manuscript for their thorough and careful review. We respond to their comments hereafter point by point and indicate what will be changed/updated in our revised manuscript.

**Referee #2 (Fortunat Joos)**

I congratulate the authors on this very comprehensive, clear, and nicely written paper. The summary of the CMIP5 and CMIP6 model results for alkalinity and CaCO3 export fluxes and the underlying analysis are excellent and instructive. The figures are well done and informative. I enjoyed reading the paper. I appreciate the great scientific and technical efforts that have made this publication possible.
I recommend publications, pending considering the following main comments.

→ We thank Referee #2 for his comments and suggestions for improving our manuscript.

**1)** The authors neglect water mass circulation/transport and water age in their presentation, despite tracer transport by circulation, mixing, and convection being key model components. The view of the biological carbon pumps and their impact on the vertical gradients in DIC and Alk is limited to the export and dissolution fluxes. This is misleading and, in my opinion, incorrect. The transport of remineralized Alk and DIC and other tracers from the deep ocean to the surface is equally important for the establishment of the vertical DIC and Alk gradients, i.e., the biological pump. Thus a major component of the marine biological cycle is simply omitted in the presentation, analysis, and discussion. This shortcoming must be corrected during the revision
Biases in the tracer gradients could be caused by biases in export fluxes and remineralization of CaCO3 and OM, but equally caused by biases in surface-to-deep water mass exchange and water mass age. A too sluggish (meridional) model circulation and deep water ventilation will lead to too old (radiocarbon) ages and too high concentrations and too high surface-to-deep gradients in Alk, DIC, P, N, AOU in comparison to observations even when export and remineralization/dissolution would be perfectly represented by the model. Similarly, too-rapid surface-to-deep water exchange would tend to diminish surface-to-deep gradients.
1a) The neglect of circulation is evident, for example, in Fig. 1 "Schematic illustration of the processes affecting alkalinity ..and key steps of the carbonate pump". The transport of high Alk waters from the depth to the surface is not shown in this figure. However, the surface ocean would become severely depleted in Alk and nutrients if there were no replacement of Alk from upward transport. Please add circulation/water transport to the scheme.
1b) I also do not trust the emergent constraint estimates for CaCO3 export for the same reason. For this estimate to be reliable, surface-to-deep water mass exchange/ventilation times need to be realistic. However, the authors do not discuss any biases in transport potentially affecting their estimate of CaCO3 export. For example, some models are known to have too old water mass ages and too high nutrients and AOU in the deep (e.g.,Frischknecht et al., GBC, 2022). The given estimate on CaCO3 export may be misleading. It requires a quantification of the influence of potential biases in circulation.
→ The reviewer's remark is highly relevant. In the revised manuscript, we will take into account the biases that the ocean circulation can contribute to.
1a) We have modified this scheme to add the circulation, but have also improved it in a more general way (see response to comment 7 below).
1b) We have assessed the potential impact of the ocean overturning circulation on our results considering AMOC and SMOC as output proxies of the global ocean circulation. We used the data provided by Heuzé et al. (2015) for the CMIP5 ensemble and Heuzé (2021) for CMIP6. Biases in ocean

circulation do not appear to be a significant driver of differences in export and alkalinity gradients across the CMIP ensembles. This will be discussed at length in the revised manuscript.

The inconsistent use of an age tracer ('agessc') across CMIP exercises limits its potential to clarify the impact of circulation on model biases in alkalinity (this will also be discussed in the revised manuscript). In our revision, we will also share a comparison between the Atlantic and Pacific oceans regarding the PIC export and sAlk vertical gradient, this provides some confidence in the relationship we share. However, the caveats the reviewer mentions will also be discussed. It was not our intention that this relationship be interpreted as a typical "emergent constraint"

**2)** Unit conversion errors related to the choice of density can lead to biases and such biased are potentially particularly large for the vertical gradients of Alk and DIC as shown in Fig. 11. The difference amounts to 32% of the vertical Alk gradient according to the authors (L661). The authors argue in section 4.2.2 that the in-situ and not the potential density of seawater should be used to convert gravimetric (mol/kg), as given by GLODAP, to volumetric (mol/m3) units, as used in most ESMs.

I remain confused regarding the recommendation to use in situ instead potential density for model-data comparison. Is the conversion between model (mol/m3) and observational units (mol/kg) not depending on whether the model is assuming incompressibility or not during tracer transport?

I also remain confused about whether GLODAP data and model data are compared correctly. The primitive equations are based on the assumption of incompressibility. Thus, tracer concentrations in mol/m3 do not change under increasing pressure. ESMs are, at least in my understanding, using the assumption of incompressibility when transporting tracers within the ocean. It seems to me that for these models potential density is the right choice to convert GLODAP data in mol/kg to model data in mol/m3 (whereas for models that do consider compressibility in-situ density would be the right choice.). Alternatively, model data could of course be converted using potential density to mol/kg. I have a suspicion that the authors converted GLODAP data to mol/m3 units using in-situ density, while model units (mol/m3) relate actually to potential density. If correct, this would affect most quantitative estimates given in this MS.

I ask all authors to check the appropriateness of the applied unit conversion and the GLODAP- model data comparison.

→ We have greatly considered the reviewer's point here. On reflection and in discussion with physical ocean modelers, the assumption of incompressibility for most models means that GLODAP unit conversion for model comparison is best performed with the reference constant density used in the models (as opposed to in situ or potential density). We note that the difference between using a constant or potential density is minimal. In the revised manuscript, we have chosen to convert observations to model units using a reference density of 1,026 kg/m3. This value is consistent with what the majority of the model groups use to initialize alkalinity and DIC (see Supplementary Table S1). It is also a value close to the average surface density. This change in the conversion of observations has little impact on our surface analysis but partially modifies our results at depth as will be discussed in the revised manuscript.

**Further comments**
3) The focus of the MS is on the surface and on global surface-to-deep gradients. This is very understandable given the high number of models analyzed. However, it is not so clear whether the improvement from CMIP5 to CMIP6 in dAlk also applies to individual basins and to the 3-d field. It would be nice if the authors would say a few words regarding 3-d fields.
(I realize that the MS contains already many elements, analyses, and figures and it is perfectly fine with me if the following suggestions on additional figures are ignored. Some ideas: perhaps complement Fig. 5 with similar figures showing profiles for different basins. It would be nice to see Fig D1 to be complemented with a figure showing the same quantities as plotted

for the surface along a section or zonally averaged (e.g., following Fig. D5). The Taylor diagram of Fig. 4 could be complemented with a diagram for the 3-d field).

→ During our analysis, we assessed multiple variables in 3D, in particular with profiles by basin and sections. Given the diversity of the representations, the analysis seemed too extensive for the manuscript. Nevertheless, we shared a set of additional figures via a zenodo link. We will put a reference to this link in the methods, in the "Background processing" section so that it is more accessible, as it was previously only cited at the end of the paper in the "Data availability" section: "We share additional figures (https://doi.org/10.5281/zenodo.7144330) to offer an ESM intercomparison (profiles, sections and maps) of the main three-dimensional variables processed in this study (Alk, sAlk, DIC, sDIC, NO3, PO4, CO3, T, S), both for CMIP5 and CMIP6, and in comparison with GLODAPv2 observations.". Given the changes that result from our response to comment 2), we will update/create a new zenodo link so that the observations in these additional figures are consistent with those in the revised manuscript.

4) I am wondering why the authors did not analyze CaCO3-derived Alk, TA*, and compare TA* model results with observation-based estimates of TA*, given their focus on PIC. Is this for reasons how the project developed or for scientific reasons? I am not suggesting that the authors should repeat or expand their analysis, but perhaps they could provide either their arguments against using TA* or say it if they want to encourage the use of TA* in follow-up studies.

→ We had the possibility to use TA*, dating from Feely et al. (2002) and reused by Koeve et al. (2014) for instance, as well as the method developed by Brewer at al. (1975) to which Sarmiento and Gruber (2006) added salinity normalization. The methods differ, in that one uses AOU (Apparent Oxygen Utilization) to distinguish alkalinity components (performed, remineralized and dissolved) and the other uses nitrate or phosphate to reconstruct potential alkalinity. Our choice of method here was influenced by Sarmiento and Gruber (2006), with little scientific justification (or downside). From a methodological point of view, it was also simpler as it would otherwise have been necessary to recalculate AOU for the models while we had the nitrate/phosphate fields directly available for the ESMs.

5) L90: As this MS deals also with the saturation horizon, it would be worthwhile to mention that the remineralization of POM has a dominating influence on $CO_3^{2-}$ in the thermocline, while CaCO3 dissolution and POM remineralization tend to have offsetting effects on $CO_3^{2-}$ in the deep ocean.

$CO_3^{2-}$ is roughly ALK-DIC. Remineralization of 1 mol of C from POM increases DIC by 1 mol and lowers ALK by about 0.15 mol (using a C:ALK ratio of 17:117) and ALK-DIC decreases by 1.15 mol. In contrast, the dissolution of 1 mol C from CaCO3 increases ALK-DIC (and thus $CO_3^{2-}$) by 1 mol. As the flux from POM/DOM remineralization is much larger than the CaCO3 dissolution flux in the thermocline, it has a dominating impact on $CO_3^{2-}$ in the thermocline. The effect of POM dissolution on $CO_3^{2-}$ can also not be neglected in the deep.

→ We agree with the reviewer, but think that this should not be part of the introduction, but rather the discussion where we assess the carbonate ion concentration and the saturation state. We will ensure that in the revised version of the paper we include elements regarding this in the discussion.

6) L97: burial of POM and CaCO3, balancing weathering input in the long run, should also be mentioned here.

→ We retrieved information regarding the global alkalinity budget for the different modeling groups (see Fig. C1). However, we are forced to discuss this possible balance between weathering input and PIC-POC burial rather than affirm it, as there are model discrepancies (see the last paragraph of Sect. 4.3.2 and the Supplementary Table S1).

7) Fig. 1: The scheme of Fig. 1 is misleading. Please show an arrow for circulation/physical transport. This is important. Upwelling indeed affects Alk at the surface and the displayed model- data differences in Alk could be due to biases in circulation and water mass age.

Fig. 1: The labeling "PIC export" is not very accurate. Suggest modifying to "PIC fluxes"

Fig. 1: Perhaps, show POM fluxes using a thin line or at least briefly mention their role for Alk in the caption.

→ Thank you for these comments. We have modified the scheme in order to be more consistent regarding the processes affecting alkalinity. We have added arrows for the circulation as well as the N-reactions. We have also modified the carbonate pump part and added the soft tissue pump in parallel to make both of them comparable.

8) L230: Battaglia et al used a probabilistic approach, while Sulpis et al used chemistry and age. Why not show both estimates in Fig. 1 and the corresponding SI figures? L235: Why focus on a single estimate for POC export and not on the range (e.g. given in Sarmiento and Gruber) or the estimate from Schlitzer, 2002,2004),Yao and Schlitzer, GMD, 2013.

→ Thank you for pointing out this confusion between Sulpis et al. (2021) and Battaglia et al. (2016), it has been changed accordingly.

We chose to consider only the latest export estimates for clarity and to have reference values for both PIC and POC export given the wide range of estimates in both cases. Indeed, the uncertainties in these exports are large as discussed in Sulpis et al. (2021) with respect to PIC, and in Schiltzer et al. (2013) with respect to POC. We have clarified the choice to use only the latest estimates in the method section, and we have added an overview of the estimated ranges for both PIC and POC exports (Sulpis et al., 2021; Schiltzer et al., 2013).

9) L268: (i) the soft tissue pump has no impact on Alk in BFM4 and MEDUSA-2.0; Perhaps these two models and CMCC-CMIP5 should be excluded from assessments of the saturation horizon?

→ It is true that this will have an influence on the calculation of both CMIP5 and CMIP6 ensemble means, or on their distributions. Nevertheless, we have decided in this study not to make a case-by-case approach which would necessitate a number of specificities to be mentioned as soon as the CMIP5/6 ensembles are considered from a statistical point of view.

10) L402: I am confused here. Has the inter-ESM sdv not increased from CMIP5 to CMIP6?

11) Fig. 6: I am a bit confused here regarding panels in b). The color bar is greyscaled, but in the chart, there are also colors such as green, orange, red, and blue. I guess darker color means higher values, but these are hard to associate with the grey scale. Maybe give the actual numbers above the bars? I guess the different bars correspond to different latitudinal bands, but how are these delineated? Maybe the colors relate to Fig. A1?

→ Exactly, the colors are referenced in Fig. A1, and enable us to distinguish the different basins. A few elements have been added to the legend in order to make things clearer and point towards fig. A1 as well to understand how the basins were delineated.

12) L485: Strictly speaking, you do not show that the effect of POM and salinity on the sAlk gradient is negligible as the results are given for the bias.

→ This has been corrected.

13) L487: Circulation and water mass age are missing. The increase at depth could also be related to an increase in water mass age allowing more time for Alk to accumulate. This should be mentioned. I would find the emergent constraint for PIC export more convincing when it would be demonstrated that the models do indeed capture the water mass age somewhat realistically, e.g. by analyzing radiocarbon. Is the relationship robust shown in Fig. 8a robust? Do you get the same global export when analyzing different basins separately? Do you get the same number when analyzing different depth levels? Do you also account for a plausible uncertainty in the observational estimate; this would broaden the inferred range.

→ The reviewer makes an interesting point regarding water mass age. The elements that we are about to implement to address this issue were stated in our response to comment 1).

This relationship between the PIC export and the sAlk vertical gradient evolves slightly depending on the basin considered with an R2 fluctuating between 0.43 for the Pacific and 0.64 for the Indian Ocean. In the reviewer responses we will share figures at basin scale, but we won't include them in the revised manuscript, given they require a complex and case-by-case analysis.

14) Section 4.2.2 I found this subsection a bit on the long side with the potential for shortening. I also missed a clear separation between spin-up and evaluation and suggest separating the discussion with respect to these two topics.

Spin up: In principle, it should not matter for the final steady state how the spatial gradient in ALK is initialized as long as the model is indeed run to a steady state. Thus, I have a hard time believing that the unit conversion affects the final steady state field. I would rather associate biases with an insufficient spin-up length and model deficiencies. Of course, if the model is initialized with the observed field and the spin-up period is kept short, the model-data agreement may be high, but potentially for the wrong reason.

→ We agree with the reviewer, particularly given comment 2) and our intended changes to observational unit conversion. This section will therefore be shortened and rewritten. In addition, we will add a section in the discussion entitled "Model data comparison" to discuss the potential biases in analyses combining observations and models.

---

## Author Response (AR1)

**Report on our manuscript revision**

Dear Editor,

We would like to thank you for your positive comment on our manuscript.

We followed most of the reviewers' comments in revising our manuscript, and we would like to thank both reviewers for their thorough and careful review that have led to improvements in the manuscript. In particular, we have:

- (1) changed text and figures to account for the modification of the density used for conversion of the GLODAPv2 observations
- (2) assessed the impact of water mass circulation on the Alk distribution
- (3) categorized model development priorities regarding the carbonate pump, highlighting the importance of the parameterization of CaCO3 production
- (4) modified figures (incl. rewriting of legends and improvement of axis labels)

In addition, we have modified the manuscript following other comments from the two reviewers. Please find below a detailed point-by-point response to all reviewers' comments.

Kind regards,

Alban Planchat, on behalf of all co-authors.

**Responses to Referee #1 (anonymous)**

→ We thank Referee #1 for his/her comments and suggestions for improving our manuscript.

In this manuscript, Planchat et al. compare the representation of ocean alkalinity within a set of 15 marine biogeochemical models, across two generations, from CMIP5 to CMIP6. Besides alkalinity they also compare seawater dissolved inorganic carbon (DIC), as well as sinking fluxes of CaCO3 and particulate organic carbon. They note an increased global export of CaCO3, closer to observations, and a strengthened vertical alkalinity gradient.

This is a long and dense manuscript, but remarkably clearly written and structured given the number of results presented. Overall, this is likely going to be a key paper that will be useful to anyone using or interpreting results from CMIP5/6 models, and will pave the way to the next generation of ocean biogeochemical models. I enjoyed reading and reviewing it and praise the authors for putting such an important piece of work together.

One main comment I have after reading this piece is that what has changed within models between both generations is clearly presented in the main text, and as a result, we are left wondering what exactly needs to be done for models to better reproduce observed alkalinity patterns. I reckon that there are many models included in this analysis and that all had unique improvements that were not necessarily well described in the original publications/technical notes, so it is very challenging to attribute the improvement to any process. Section 3.1 does a great job synthesizing the different ways to represent various processes across different model groups; it would be useful to do the same synthesis effort but focusing on changes between CMIP5 and CMIP6. Section 4.1 touches this issue, but is very short, and seems to be summarized by the fact that no major trend emerges with regards to the modelling schemes. Could the authors develop this section further? What could be the main processes behind the improved, stronger alkalinity gradient? Inclusion of aragonite or of better diagenesis modules? Could the authors express their opinions on which processes need to be implemented in priority in the next generation of models?

 $\rightarrow$  We now further discuss potential consequences of model developments in relation to the carbonate pump, the representation of alkalinity and its role in the carbon cycle in Sect. 4.4.3. Although it remains difficult to establish priorities for model development, the parameterization of CaCO3 production seems essential, since it can impact the surface carbon cycle on short timescales. The last paragraph of this section reads:

"In order to improve the representation of the carbonate pump and account for the impacts and feedbacks associated with acidification and climate change, certain model developments are desirable. The main priority is the parameterization of CaCO3 production, particularly a consensus on its dependence on the saturation state, temperature and organic matter production, as this should influence the surface carbon cycle. Further desirable developments include: i) the representation of aragonite, which could partly redistribute Alk in the sub-surface and is more responsive to climate perturbations, ii) the representation of benthic calcifiers, such as corals, that could respond on relatively short timescales, iii) the explicit representation of saturation state dependent PIC dissolution, and iv) the representation of dissolved and buried PIC fractions at the seafloor with or without the use of a diagenesis module."

**Minor comments:**

**Abstract:**

L31: avoid use of significant for non-statistical meaning  $\rightarrow$  This has been corrected.

**Introduction: L87: remove extra "Fig. 1"**

 $\rightarrow$  This has been corrected.

Methods: L170: define "piControl" and "Historical" experiments  $\rightarrow$  Now defined. L180-181: just to make sure I understand: export at 100m from the 3D fields is in theory the exact same thing than export at 100m from the 2D fields?

 $\rightarrow$  Exactly, it should be the same, but they were manually calculated from the 3D-fields.

L182-183: why using MIROC models then? Can you precise whether there is no export in those models, or those quantities exist but were not saved and/or made available to you?

 $\rightarrow$  We included MIROC CMIP5 ESMs, as export data were modeled although these outputs were no longer available. We have added this specification in the text to make it clear:

"Although modeled, export data were not available for MIROC-ESM and MIROC-ESM-CHEM (CMIP5)." L194: what are those constants?

 $\rightarrow$  Here we are referring to the seawater equilibrium constants (e.g., K1 and K2 for the CO2 system). We have added "seawater" before "equilibrium constant" to improve clarity. We have not mentioned specifically K1 and K2 however, since we think it can be a bit confusing for the reader without bringing much to the manuscript.

L196-197: are those two models the only ones that include exchanges at the seafloor? Discarding the lower layer would appear justified if the goal was to compare water-column processes amongst models and if only the lower layer was affected by non-water-column (i.e. seafloor). However, seafloor processes, e.g., dissolution or respiration, should also affect water-column chemistry far away from the seafloor. I would like further discussion regarding the role of seafloor processes in the current model intercomparison (see main comment).

 $\rightarrow$  We disregarded the scarce values given at 5500 m for MIROC ESMs because they were not plausible and likely affected by an error in model output processing before outputs were shared (Fig. R1). We have had confirmation from the MIROC development team that the last layer of the ocean is the bottom boundary layer, and we were asked to ignore this layer in our analysis.

Figure R1: Alk maps for MIROC-ESM-CHEM (CMIP5) at 5000 m (left panel) and 5500 m (right panel) in 2100 in the piControl experiment.

With regard to the role of sediment processes, we have expanded Sect. 3.1.4 "Sedimentation and Alk sources/sinks" to better distinguish the differences between the models. In addition, this is discussed in Sect. 4.2.1 and 4.4.2, highlighting the role of sediment processes in the closure of the Alk budget. It is difficult to further discuss seafloor processes within the context of an intercomparison as it becomes complicated when a diagenetic sub-module is considered for instance. Unfortunately, the model outputs shared on ESGF do not currently permit analysis of seafloor processes, particularly the burial of PIC and POC. This is a recommendation we make for future CMIP exercises in Sect. 4.2.3:

"Sharing of complete export fields would ideally be accompanied by three-dimensional fields of remineralization and dissolution ('remoc', 'dcalc' and 'darag') to facilitate analysis of processes such as the biological pump throughout the water column. Finally, sharing of vertically integrated calcite and aragonite production ('intpcalcite' and 'intparag') and POC and PIC burial ('froc' and 'fric') would also improve assessments of the influence of the biological pump on vertical DIC and Alk profiles (see Fig. 9a,b)."

**L203: define SSP and explain the difference with RCP**

 $\rightarrow$  This has been specified.

L230-231: I believe that the Sulpis and Battaglia references are mixed up in this sentence  $\rightarrow$  This has been changed accordingly.

L257: what is tau^5m? the concentration of a given tracer tau at 5 m-depth?

 $\rightarrow$  Yes, it is. We have clarified this in the manuscript.

Results:

Section 3.1.1.: An explicit integration of calcification in models would look like a series of equations used to compute calcification for individual groups, as a function of variables such as light, saturation state, etc. If I understand correctly the implicit integration used by all models skip the production step and computes the PIC export directly, as a function of the same variables (light, saturation, etc.). Because this implicit calcification scheme misses "gut dissolution", as explained in this section, models using it should all miss the shallow (in the couple hundred meters below the euphotic layer) but strong alkalinity production observed in the ocean, see Feely et al. (2004, DOI: 10.1126/science.1097329), Subhas et al. (2022, https://doi.org/10.1029/2022GB007388), Sulpis et al. (2021, https://doi.org/10.1038/s41561-021-00743-y).Could you show and/or discuss that somewhere?

 $\rightarrow$  The reviewer is correct. The fact that models do not represent explicit gut dissolution could explain why they only simulate one deep dissolution peak, compared to two in the observational-based studies, with a sub-surface peak at a few hundreds meters depth (Feely et al., 2004; Sulpis et al., 2021 and Subhas et al. 2022). The added paragraph in Sect. 4.4.3 "Potential changes in the carbonate pump" reads:

"Additionally, implicit PIC production avoids the representation of gross PIC production and zooplankton gut dissolution (e.g., Jansen and Wolf-Gladrow, 2001), which can potentially occur deeper than 100 m. Simulating gross PIC production and zooplankton gut dissolution may permit the representation of a sub-surface dissolution peak in addition to the deep dissolution peak seen in models with explicit saturation state dependent dissolution. Such a double peak in dissolution would be consistent with observations (Feely et al., 2004; Sulpis et al., 2021; Subhas et al., 2022). The representation of these two dissolution peaks may be important in the context of transient simulations as they may have different sensitivities to climate perturbations."

Section 3.1.4.: the term "sedimentation" sounds simply like physical accumulation or burial, whereas in this context it should include other sediment transport processes and chemical reactions. Perhaps replace with "diagenesis"?

 $\rightarrow$  The reviewer is correct that this is a useful distinction to make. Models like PISCESv2 only distinguish between a dissolved and a buried fraction, while others, like COBALTv2, consider diagenesis by using a sediment module. We have clarified this in Sect. 3.1.4:

"The fate of PIC reaching the seafloor is one of the determinants of the ocean Alk inventory and closure of the CaCO3 budget. There is a high diversity among models in their representation of sedimentation processes associated with calcium carbonate. For some models, all of the PIC reaching the seafloor is considered permanently buried and lost from the ocean (e.g., CMOC and OECO2). Other models dissolve all of the PIC reaching the seafloor closing the calcium carbonate cycle and avoiding its processing in the seabed (e.g., WOMBAT and NPZD-MRI). A final subset of models represents sediment processes. Some of these distinguish a dissolved and a buried PIC fraction (e.g., CanOE, BFM5.2 and PISCES), while others represent diagenesis with a sediment module (e.g., HAMOCC, BLINGv2 and COBALTv2)."

We have kept the same section title however because, when we refer to sedimentation in the manuscript, we are fundamentally interested in the potential buried fraction of PIC.

L445: please add a reference for an observational estimate of the rain ratio

The rain-ratio is here estimated from the PIC and POC export values of Sulpis et al. (2021) and De Vries and Weber (2017), respectively. This is now detailed in the methodology, in Sect. 2.3 "Data products": "The observational-based rain ratio amounts to 0.14, and was computed from integrated PIC and POC

export values from Sulpis et al. (2021) and DeVries and Weber (2017), respectively."

**Tables:**

Table 1 define "MBG"  $\rightarrow$  Now defined.

**Figures:**

Fig. 1 the arrow associated with (3) dissolution should be colored in green because, as for the arrow associated with sediment mobilization, it represents a flux impacting seawater alkalinity

 $\rightarrow$  This has been corrected. The schematic is now more consistent regarding the processes affecting Alk. We have added arrows for circulation and N-reactions. We have also modified the carbonate pump and added the soft tissue pump to make them comparable.

Figure 2 why are the 6th and 8th rows for the MPI models not colored instead of being colored in grey (N/A)?

 $\rightarrow$  This was because we were still waiting for the input from the MPI developer team regarding this. We have completed these few cells (as for Fig. C1 and Supplementary Table S1) using the information available in the reference papers corresponding to these ESMs (see Supplementary Table S1).

Figures 2 and 6: all CaCO3 in ACCESS-ESM1-5 is aragonite? Can you please elaborate on that?  $\rightarrow$  ACCESS-ESM1-5 shares CaCO3 as aragonite in its modeled outputs, although it is in fact a generic type of CaCO3 that is modeled. This is clarified in Sect. 3.1.1:

"Certain groups represent a generic biogenic CaCO3 (CSIRO with WOMBAT for CMIP6, MIROC with OECO1/2 for CMIP5/6 and MOHC with diat-HadOCC for CMIP5), but attribute it either to calcite or aragonite to conform to CMIP output requirements."

**Responses to Referee #2 (Fortunat Joos)**

I congratulate the authors on this very comprehensive, clear, and nicely written paper. The summary of the CMIP5 and CMIP6 model results for alkalinity and CaCO3 export fluxes and the underlying analysis are excellent and instructive. The figures are well done and informative. I enjoyed reading the paper. I appreciate the great scientific and technical efforts that have made this publication possible. I recommend publications, pending considering the following main comments.

**$\rightarrow$ We thank Referee #2 for his comments and suggestions for improving our manuscript.**

1) The authors neglect water mass circulation/transport and water age in their presentation, despite tracer transport by circulation, mixing, and convection being key model components. The view of the biological carbon pumps and their impact on the vertical gradients in DIC and Alk is limited to the export and dissolution fluxes. This is misleading and, in my opinion, incorrect. The transport of remineralized Alk and DIC and other tracers from the deep ocean to the surface is equally important for the establishment of the vertical DIC and Alk gradients, i.e., the biological pump. Thus a major component of the marine biological cycle is simply omitted in the presentation, analysis, and discussion. This shortcoming must be corrected during the revision

Biases in the tracer gradients could be caused by biases in export fluxes and remineralization of CaCO3 and OM, but equally caused by biases in surface-to-deep water mass exchange and water mass age. A too sluggish (meridional) model circulation and deep water ventilation will lead to too old (radiocarbon) ages and too high concentrations and too high surface-to-deep gradients in Alk, DIC, P, N, AOU in comparison to observations even when export and remineralization/dissolution would be perfectly represented by the model. Similarly, too-rapid surface-to-deep water exchange would tend to diminish surface-to-deep gradients.

 $\rightarrow$  The reviewer's remark is highly relevant. We now include assessment and discussion of the model differences that ocean circulation can contribute to."